**Technical Report**

# Retinoic acid induces human gastruloids with posterior embryo-like structures

Nobuhiko Hamazaki [1,2,3,4,5,11] ✉, Wei Yang [1,5,11], Connor A. Kubo [1,5], Chengxiang Qiu [1,5], Beth K. Martin [1,5], Riddhiman K. Garge [1,4], Samuel G. Regalado [1,5,6], Eva K. Nichols [1], Sriram Pendyala [1], Nicholas Bradley [1], Douglas M. Fowler [1,4,7], Choli Lee [1,5], Riza M. Daza [1,5], Sanjay Srivatsan [1,4,8] & Jay Shendure [1,3,4,5,9,10] ✉

Gastruloids are a powerful in vitro model of early human development. However, although elongated and composed of all three germ layers, human gastruloids do not morphologically resemble post-implantation human embryos. Here we show that an early pulse of retinoic acid (RA), together with later Matrigel, robustly induces human gastruloids with posterior embryo-like morphological structures, including a neural tube flanked by segmented somites and diverse cell types, including neural crest, neural progenitors, renal progenitors and myocytes. Through in silico staging based on single-cell RNA sequencing, we find that human RA-gastruloids progress further than other human or mouse embryo models, aligning to E9.5 mouse and CS11 cynomolgus monkey embryos. We leverage chemical and genetic perturbations of RA-gastruloids to confirm that WNT and BMP signalling regulate somite formation and neural tube length in the human context, while transcription factors TBX6 and PAX3 underpin presomitic mesoderm and neural crest, respectively. Looking forward, RA-gastruloids are a robust, scalable model for decoding early human embryogenesis.

The molecular, cellular and developmental biology of early human embryogenesis is of fundamental interest. However, studying in vivo post-implantation early human development, directly or by ex vivo culture[1–3], is challenging. In vitro embryo models, which derive from pluripotent stem (PS) cells[4–9], are a promising alternative for accessing early human development. For example, PS cell-derived gastruloids[10–14] bear derivatives of all three germ layers and specify an anteroposterior (A–P) axis[13].

The addition of Matrigel during mouse gastruloid induction results in morphological features even more characteristic of in vivo development, including a neural tube and segmented somites[10,15].

However, these structures are not observed in conventional human gastruloids. Although derivative protocols yield human embryo models with segmented somites[16–20], spinal cord neurons and skeletal muscle cells[21] or elongated multi-lineage structures with peripheral neurons[22], a human embryo model with balanced representation of the neural and somitic lineages remains elusive.

We investigated differences between mouse and human gastruloids, which led us to hypothesize that neuromesodermal progenitors (NMPs) in conventional human gastruloids are mesodermally biased. In seeking to correct this, we discovered that an early pulse of retinoic acid (RA), together with later Matrigel, induces human

[1]Department of Genome Sciences, University of Washington, Seattle, WA, USA. [2]Department of Obstetrics & Gynecology, University of Washington, Seattle, WA, USA. [3]Institute for Stem Cell & Regenerative Medicine, University of Washington, Seattle, WA, USA. [4]Brotman Baty Institute for Precision Medicine, Seattle, WA, USA. [5]Seattle Hub for Synthetic Biology, Seattle, WA, USA. [6]Medical Scientist Training Program, University of Washington, Seattle, WA, USA. [7]Department of Bioengineering, University of Washington, Seattle, WA, USA. [8]Fred Hutchinson Cancer Center, Seattle, WA, USA. [9]Howard Hughes Medical Institute, Seattle, WA, USA. [10]Allen Discovery Center for Cell Lineage Tracing, Seattle, WA, USA. [11]These authors contributed equally: Nobuhiko Hamazaki, Wei Yang. ✉e-mail: hamazaki@uw.edu; shendure@uw.edu

gastruloids with posterior embryo-like morphological structures, more advanced cell types and less inter-individual variation, than observed in conventional gastruloids. Here we extensively characterize these human RA-gastruloids, including leveraging single-cell RNA sequencing (scRNA-seq) data to computationally stage their progression relative to other mammalian embryo models and in vivo development. Finally, we perform chemical and genetic perturbations of human RA-gastruloids to showcase their potential as a model for advancing our understanding of early human embryogenesis.

## Results

### Comparative analysis of human versus mouse gastruloids

We sought to induce human gastruloids with more advanced morphological features than observed in conventional human gastruloids[11]. To this end, we initially attempted Matrigel supplementation, previously shown to induce mouse gastruloids with trunk-like structures (TLSs)[15]. However, although Matrigel supplementation substantially increased the extent and success rate of human gastruloid elongation, it did not alter their morphology (Extended Data Fig. 1 and Supplementary Table 1).

To investigate this difference, we performed scRNA-seq on conventional human gastruloids[23], without Matrigel supplementation, at 24, 48, 72 and 96 h after induction (Fig. 1a,b). After filtering low-quality cells and doublets, we performed dimensionality reduction, unsupervised clustering and cell type annotation on the transcriptional profiles of ~44,000 cells from four time points (Fig. 1c and Supplementary Table 2). In these data, we can follow the emergence of key cell populations over time (Fig. 1d and Supplementary Figs. 1–4). At 24 h after seeding, we observe a continuum of molecular profiles with heterogeneous expression of transcription factors with established roles in early development (Fig. 1d, Supplementary Fig. 1 and Supplementary Table 3). Distinct populations of $FOXA2^+$ cells were either $NOTO^+$ or $SOX17^+$ and presumably correspond to axial mesoderm and definitive endoderm equivalents, respectively (Supplementary Fig. 1). For annotation, integration with scRNA-seq data from Carnegie stage (CS7) human embryos[24] was informative, and suggest that early stage human gastruloids model the primitive streak, nascent mesoderm and emergent mesoderm (Supplementary Fig. 5). However, extra-embryonic, haematopoietic endothelial and primordial germ cell equivalents were missing, as were epiblast and non-neural ectoderm equivalents.

Through integration with scRNA-seq data from conventional[10] and TLS[15] mouse gastruloids (Supplementary Fig. 6), we annotated cell types in 48–96 h human gastruloids, including a continuum from NMPs ($TBXT^+$; $SOX2^+$; $NKX1-2^+$; $CDX2^+$) to presomitic mesoderm ($TBX6^+$) to a differentiation front ($MESP2^+$; $RIPPLY2^+$; $RIPPLY1^+$) giving rise to differentiated somites ($FST^+$; $PAX3^+$); and additionally, advanced ($OSR1^+$; $WT1^-$) and cardiac mesoderm ($HAND1^+$; $NKX2-5^+$; $TNNT2^+$) (Fig. 1d and Supplementary Figs. 2–4). Axial mesoderm or notochord-like cells ($NOTO^+$) were also detected at later time points. Through integration with mouse E6.5–E8.5 data[25], we could also track a gut/endoderm-like subpopulation through 96 h ($APELA^+$, $FOXA2^+$, $KRT8^+$, $SOX17^+$) (Supplementary Fig. 7a–h).

Bipotential NMPs are the source of both presomitic mesoderm (PSM) and posterior neural tubes[26–30]. To compare the differentiation potential of NMPs in mouse versus human gastruloids, we performed a focused analysis of NMPs, PSM and neural tube cells in scRNA-seq data from mouse gastruloids[10], mouse TLS[15] and human gastruloids[11]. Although NMPs and PSM were detected in all three models, neural tube cells ($IRX3^+$, $SOX1^+$, $PAX6^+$) were identifiable only in mouse gastruloids and mouse TLS (Fig. 1d,e and Supplementary Table 2). Based on this observation, we hypothesized that a bias in the differentiation potential of NMPs toward mesodermal fates might underlie the failure of human gastruloids to generate an elongated neural tube.

RA is a signalling molecule that can induce neural cell fates from NMPs, both in vivo and in vitro[30–37]. We compared the expression of

transcripts encoding enzymes involved in RA synthesis or degradation between gastruloid models (Extended Data Fig. 2a,b). Human gastruloids exhibit much lower expression of ALDH genes, which encode enzymes that convert retinal to RA (Extended Data Fig. 2a,b). The contrast is particularly stark for $ALDH1A2$, which encodes the enzyme that catalyses RA synthesis from retinaldehyde; its expression remains low in human gastruloids, but is acutely upregulated then downregulated at 96–120 h in mouse gastruloids and TLSs, suggesting a critical time window for receptivity to RA signalling in this model (Extended Data Fig. 2c). We also observed higher expression of CYP26 genes, which counteract RA signalling, in human gastruloids (Extended Data Fig. 2a–e). Conversely, human gastruloids exhibit higher expression of some WNT genes at 0 and 24 h (Extended Data Fig. 2b). Taken together, these observations suggest that if NMPs in conventional human gastruloids are biased toward mesodermal fates, it may be due to insufficient RA and/or excess WNT signalling.

### An early pulse of retinoic acid induces trunk-like structures

To test this hypothesis, we sought to restore the bipotential state of NMPs by supplementing human gastruloids with RA. In addition to Matrigel (which, on its own, failed to induce a neural tube or segmented somites; Extended Data Fig. 1 and Supplementary Table 1), we explored adding various concentrations of RA to the gastruloid induction medium (Supplementary Fig. 8a). Encouragingly, SOX2-mCit intensities increased at 24 or 96 h in a dose-dependent manner (Supplementary Fig. 8b,c). However, although gastruloid elongation was enhanced at RA concentrations ranging from 100 nM to 1 µM, we observed neither neural tube formation nor somite segmentation.

We speculated that continuous RA exposure might perturb the differentiation of other cell types between 24 and 48 h. We therefore attempted a similar experiment, but withdrew RA at 24 h and then added it back at 48 h together with 10% Matrigel (Extended Data Fig. 3a). Remarkably, this temporally discontinuous RA regimen induced the elongation of SOX2-mCit-positive neural tube-like structures as well as the formation of primitive segmentations of apparent somites (Extended Data Fig. 3b). Building on this finding, we optimized the number of cells used in the initial seeding of gastruloids (Extended Data Fig. 3c). With a larger seeding, discontinuous RA and Matrigel supplementation, gastruloids exhibited multiple, segmented somites, together with a neural tube-like structure along an A–P axis (Fig. 2a,b). These structures formed robustly under these conditions, with 89% of elongated gastruloids exhibiting both segmented somite and neural tube-like structures across five independent experiments (Fig. 2c). We hereafter refer to these as 'human RA-gastruloids'.

Notably, while the first pulse of RA (0–24 h), together with Matrigel starting at 48 h, was sufficient to induce structures resembling a neural tube flanked by somites, the second pulse of RA (48–120 h) was not (Fig. 2d), suggesting the early pulse of RA underlies the maintenance of bipotentiality in early NMPs. Consistent with the lower expression of ALDH genes in human gastruloids (Extended Data Fig. 2), neither retinol or retinal, which are metabolic precursors of RA, could substitute for RA in this role (Supplementary Fig. 9). In contrast, as shown by Yamanaka et al. (2022), retinal, retinol and RA are interchangeable in facilitating somite epithelialization[18].

We also assessed modulation of the concentration of CHIR99021 (hereafter CHIR; an agonist of WNT signalling) during pre-treatment of human RA-gastruloids. Lower CHIR resulted in elongated gut tube-like structures with expression of SOX17-tdTomato and FOXA2 alongside neural tube extension, albeit with greater disorganization of gastruloid structures (Extended Data Fig. 4). The CHIR dose-dependent appearance of gut tube-like structures suggests different WNT signalling requirements for endoderm and mesoderm differentiation from the primitive streak/mesendoderm-like state.

Immunostaining confirmed that the neural tube-like structures are $SOX2^+/SOX1^+$ (Fig. 2e,f), while the somite-like structures are

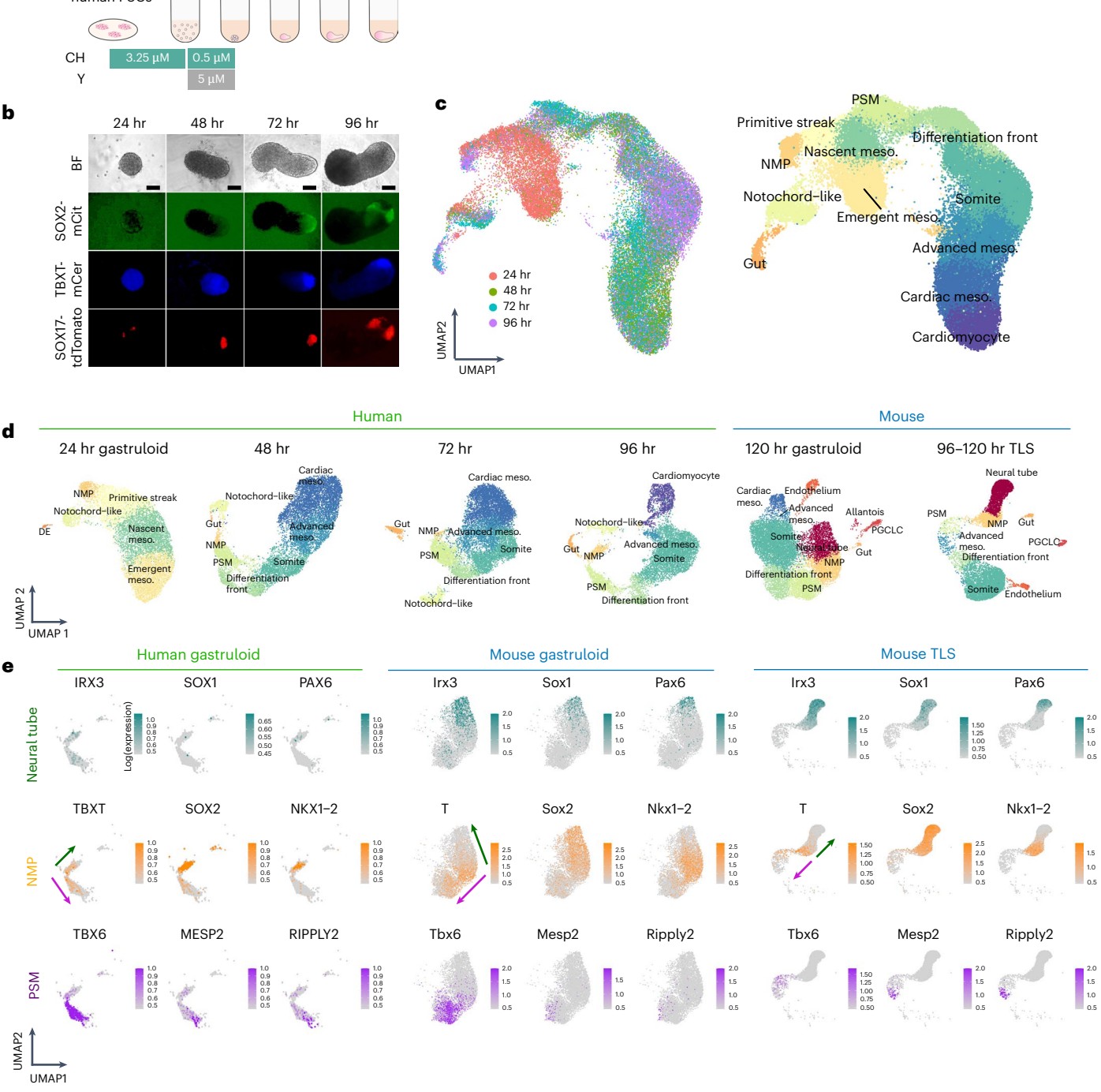

**Fig. 1 | Single-cell transcriptional profiling of a time-series of conventional human gastruloids. a**, Schematic of conventional human gastruloid protocol[11]. CH, CHIR99021; Y, Y-27632. **b**, Representative images of RUES2-GLR[44]. The experiments were repeated independently six times with similar results. SOX2-mCit, pluripotent and ectoderm marker; TBXT-mCer, mesoderm marker; SOX17-tdTomato, endoderm marker. Scale bar, 100 μm. **c**, Integrated UMAP of ~44,000 scRNA-seq profiles from four time points of human gastruloid development, coloured by time point (left) or cell type annotation (right). The 48 h to 96 h gastruloids were sequenced per time point. **d**, UMAP projection of scRNA-seq profiles from individual time points of human gastruloids (generated by this study based on published protocols[11]) or published data from mouse gastruloids[10] or mouse TLSs[15]. PGCLC, primordial germ cell-like cell; DE, definitive endoderm. **e**, Normalized expression of marker genes for neural tube (top row), NMPs (middle row) or PSM (bottom row) in UMAP projections of scRNA-seq profiles of extracted cell types (neural tube, NMP and PSM cells) from human gastruloids at 96 h, mouse gastruloids at 120 h (ref. [10]), or mouse TLS at 120 h (ref. [15]). Arrows represent putative differentiation of NMPs toward neural tube (green) and PSM (purple) fates, respectively. The key point is that NMP-like and PSM-like cells are detected in all three models, but neural tube-like cells are detected only in the two mouse models. UMAP, Uniform Manifold Approximation and Projection.

PAX3⁺ (ref. [38]). Both neural tube and somite-like structures exhibited asymmetrical accumulation of F-actin (stained by phalloidin) and CDH2 on their apical sides (Fig. 2f and Extended Data Fig. 5a,b), similar to in vivo embryonic somites and neural tubes[39–41].

We next quantified the morphological properties of human RA-gastruloids (Extended Data Fig. 5c,k). Overall, human RA-gastruloids elongate as a function of time and reach a length (1.5–2.0 mm; Extended Data Fig. 5d) similar to other human somite model systems

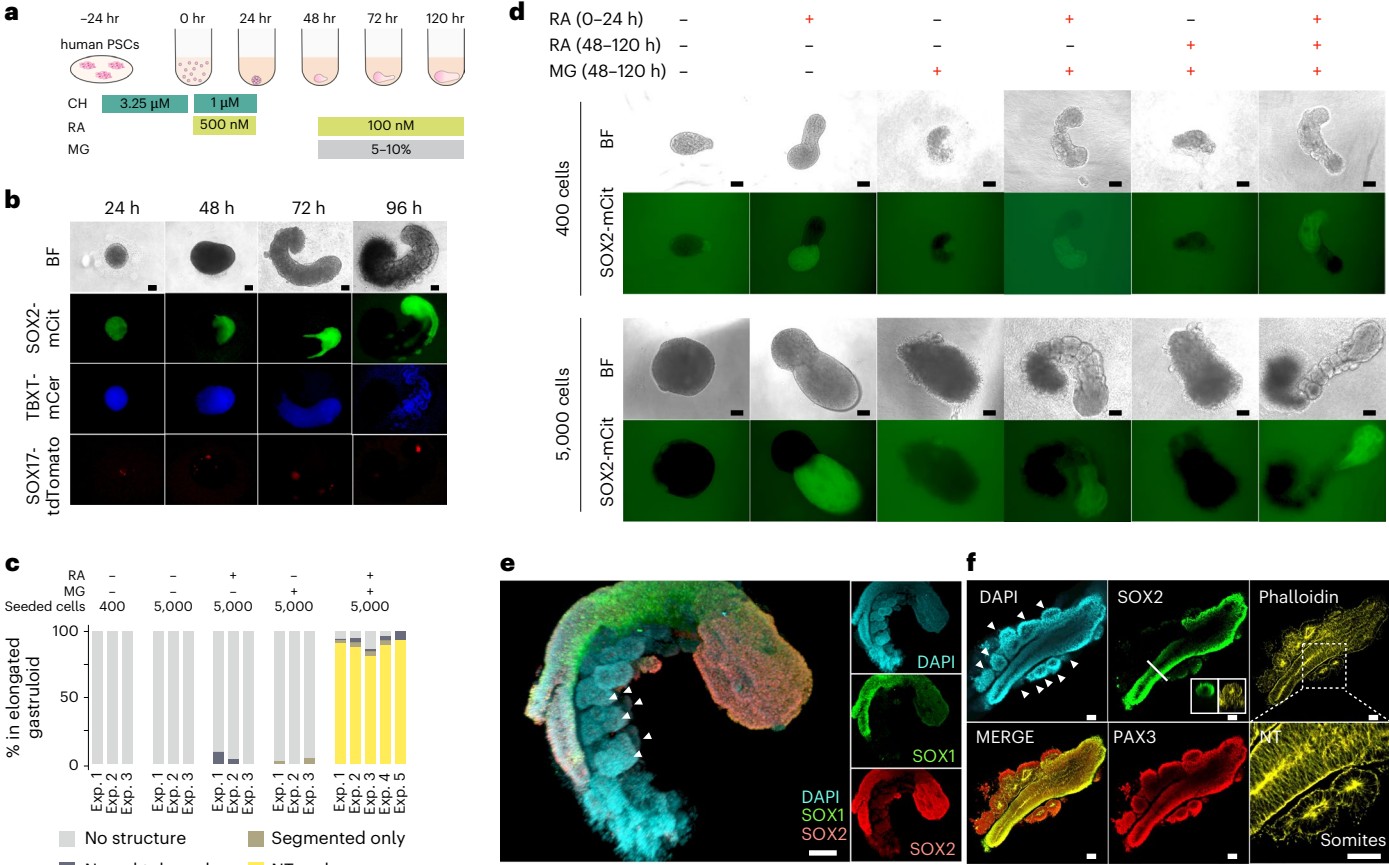

**Fig. 2 | Robust induction of human gastruloids with both a neural tube and segmented somites via a discontinuous regimen of retinoic acid. a**, Schematic of human RA-gastruloid protocol. MG, Matrigel; CH; CHIR99021. RA was applied for the first 24 h after induction, then withdrawn, then added back at 48 h along with 5–10% MG. **b**, Representative images of developing human RA-gastruloids. Scale bar, 100 µm. $n = 768$ (96 × 8 plates) human RA-gastruloids showed similar morphology (elongated gastruloid with flanking somites) and patterns of marker gene expression (asymmetric, elongated SOX2-mCit⁺ signal flanked by non-overlapping weak TBXT-mCer signal overlaying somites). **c**, Quantification of the frequency of NT elongation and somite segmentation under various experimental conditions. NT, neural tube; Seg, segmented somites. **d**, Representative images of gastruloids with or without RA/MG from 400 cells (top) or 5,000 cells (bottom). The concentration of RA at 0–24 h and 48–120 h was 500 nM and 100 nM, respectively. $n = 24$–48 per condition. The concentration of MG was 5%. Scale

bar, 100 µm. **e**, 3D projections of immunostained 120 h RA-gastruloids. The SOX1⁺, SOX2⁺ region corresponds to the NT-like structure, flanked by somite-like structures. Arrowheads indicate paired somites. $n = 12/13$ human RA-gastruloids showed similar morphology (elongated gastruloid with flanking somites) and patterns of marker gene expression (asymmetric, elongated, coincident SOX1 and SOX2 staining that did not extend to flanking somites). Scale bar, 100 µm. **f**, Confocal section of immunostained 120 h RA-gastruloid. Phalloidin staining shows the apical accumulation of F-actin in SOX2⁺, PAX3⁺ NT and PAX3⁺ somites. Slice images of the area indicated in the bold line are shown in the SOX2 staining image. The magnified region in the phalloidin staining image is indicated by a dotted square. Arrowheads indicate paired somites. Scale bar, 100 µm. $n = 9/11$ human RA-gastruloids showed similar morphology (elongated gastruloid with flanking somites) and patterns of marker gene expression (asymmetric, elongated SOX2 staining flanked by PAX3 staining of flanking somites).

(segmentoids[17], 1.2–1.6 mm; axioloids[18], 1.0–1.4 mm). Somite counts increased with time, reaching 8–9 pairs in human RA-gastruloids by 120 h (Extended Data Fig. 5j). Somite lengths, widths and areas are comparable (~100 µm) to the axioloid[18] and somitoid[16] models (Extended Data Fig. 5k). Of note, most somites in human RA-gastruloids are paired and epithelialized even without the second pulse of RA (Extended Data Fig. 5l–o). This contrasts with axioloids[18], which mostly depend on a late pulse of retinoid signalling for somite epithelialization. Possible explanations for this difference include residual effects from the first RA pulse, residual RA molecules from the first RA pulse, or additional signalling from cell types present in RA-gastruloids but not in axioloids. Upon live imaging, human RA-gastruloids exhibit periodic generation of segmented somite pairs in approximately 5-h intervals, similar to human embryos[42,43] and somitogenesis models[16,18] (Extended Data Fig. 6a–c).

Taken together, these results show that an early pulse of RA, together with Matrigel, robustly induces human gastruloids with trunk-like morphological structures, which is a neural tube flanked

by segmented somites. Notably, although the human RA-gastruloids characterized above were generated from RUES2-GLR[44] embryonic stem (ES) cell line, the same induction conditions robustly gave rise to these morphological structures when starting with another human ES (H9) cell line or a human induced PS (iPS) cell (WTC11) line (Supplementary Fig. 10).

**Transcriptional profiling of human RA-gastruloids**

We applied scRNA-seq to human RA-gastruloids. Clustering and annotation of 5,347 and 18,324 single-cell profiles identified 9 and 12 cell types at 96 and 120 h, respectively (Fig. 3a–d). In addition to cell types observed in conventional human gastruloids (NMPs, cardiac mesoderm, differentiation front, somites, gut), we identified cell types resembling neural tube (*PAX6⁺*; *SOX1⁺*), neural crest (*FOXD3⁺*; *SOX10⁺*), neural progenitors (*ONECUT1⁺*; *ONECUT2⁺*), intermediate mesoderm (IMM; *WT1⁺*; *OSR1⁺*) and renal epithelium (*LHX1⁺*; *PAX2⁺*) at 96 h, as well as myocytes (*NEB⁺*), cardiomyocytes (*TNNT2⁺*) and endothelium (*PLVAP⁺*) at 120 h (Fig. 3a–d, Supplementary Fig. 11 and Extended Data Fig. 7a–c).

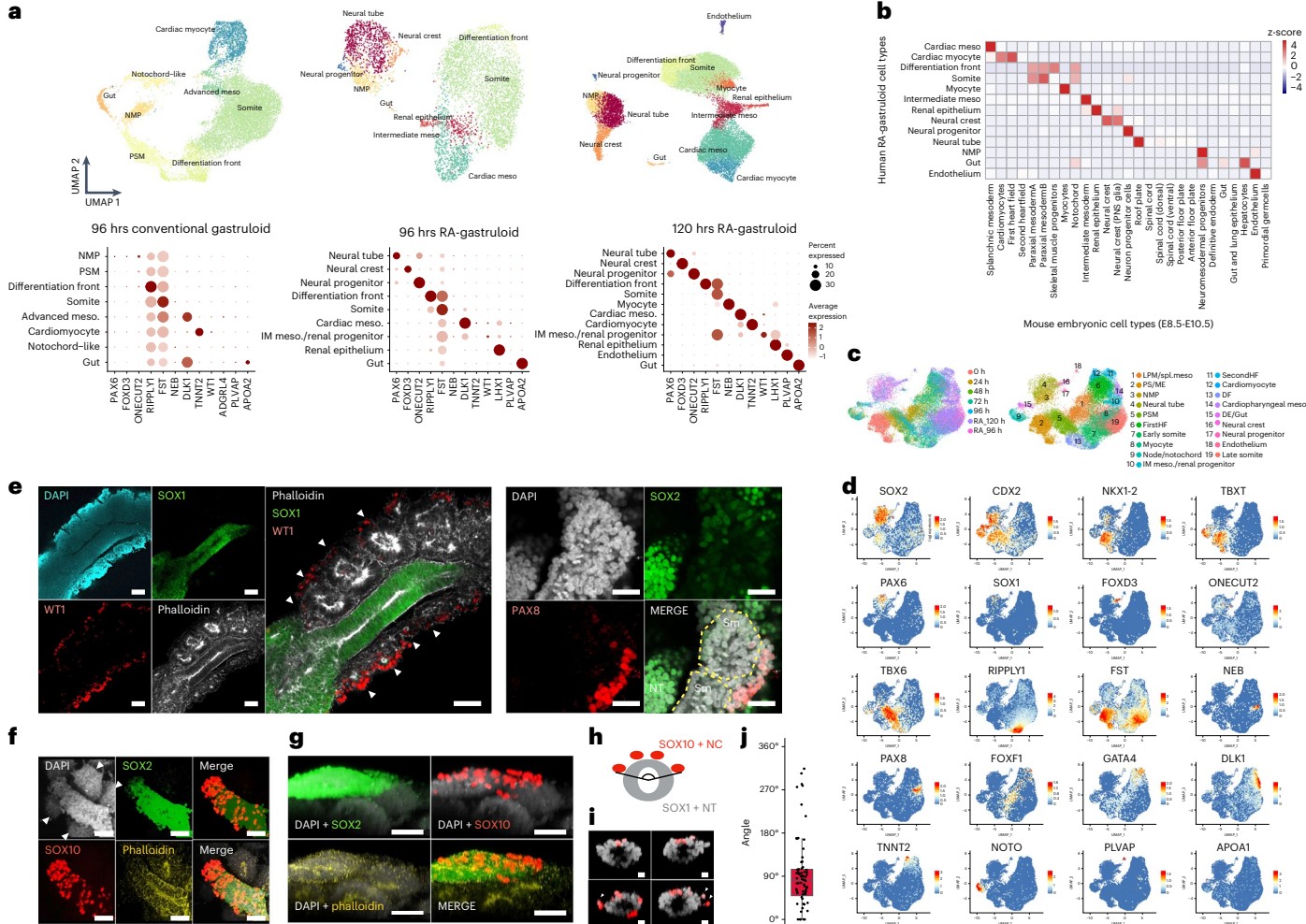

**Fig. 3 | Induction of neural crest, IMM and other advanced cell types in human RA-gastruloids. a**, Annotated UMAP of scRNA-seq profiles from conventional (96 h) or RA (96 or 120 h) human gastruloids (top). The 48 h to 96 h gastruloids were sequenced per time point. IM meso., intermediate mesoderm. Marker gene expression (bottom). **b**, Cell type mapping of 120 h human RA-gastruloids against mouse embryonic datasets[58,59] via non-negative least-squares regression. **c**, Integrated UMAP of scRNA-seq data from conventional (0–96 h) and RA (96–120 h) human gastruloids. LPM/Spl.meso, lateral plate mesoderm/splanchnic mesoderm; PS/ME, primitive streak/mesoendoderm; FirstHF, first heart field; IM meso/renal progenitor, intermediate mesoderm/renal progenitor; SecondHF, second heart field; DF, differentiation front; DE/Gut, definitive endoderm/gut. **d**, Marker gene expression. **e**, Immunostaining of IMM and renal epithelium in 120 h human RA-gastruloids. Anti-WT1 (red, IMM), anti-SOX1 (green, neural tube), phalloidin (white, F-actin) or DAPI (cyan, nuclear) staining (left). WT1 + IMM-like cells appear lateral to the phalloidin-stained somites. Scale bar, 100 µm. Anti-PAX8 (red, renal epithelium), anti-SOX2 (green, neural tube) or DAPI (cyan, nuclear) staining (right). Scale bar, 100 µm. Arrowheads indicate paired somites. n = 4/4 and n = 3/4 gastruloids showed similar patterns of marker gene expression, respectively (punctate WT1 or PAX8 staining at the lateral border of somites). Sm, somite. **f–j**, Immunostaining of neural crest-like cells in 120 h RA-gastruloids. **f**, 3D projection of somite and neural tube with anti-SOX2 (green, neural cells), anti-SOX10 (red, neural crest), phalloidin (yellow, F-actin) or DAPI (white, nuclear) staining. Arrowheads indicate paired somites. **g**, Lateral views of human RA-gastruloids. Scale bar, 50 µm. n = 11/12 gastruloids showed similar patterns of marker gene expression (punctate SOX10 staining asymmetrically localized on one surface of putative neural tube). **h**, Quantification of distribution of SOX10+ neural crest cells on neural tubes. Two straight lines were drawn from the centre point of a neural tube toward the outermost SOX10+ cells. NC, neural crest. **i**, Representative images of sliced neural tube images. Arrowhead indicates migrating SOX10+ cells. Scale bars, 10 µm. The experiments were repeated independently three times with similar results. **j**, Boxplot showing the distribution of the angle encompassing all SOX10+ cells observed on the surface of a given gastruloid. n = 71. DAPI, 4,6-diamidino-2-phenylindole.

Posterior HOX gene expression was enhanced at 120 h relative to 96 h (Extended Data Fig. 7d–e).

We were surprised to observe many of these additional cell types, for example IMM and renal epithelium, because they are absent not only from conventional human gastruloids, but also from mouse gastruloid and TLSs[10,11,15]. During in vivo development, IMM arises between the somites and lateral plate mesoderm along the mediolateral axis. To investigate the spatial distribution of the IMM-annotated cells, we immunostained human RA-gastruloids with anti-WT1 antibody or anti-PAX8. Consistent with expectation, both the WT1+ cells (annotated as IMM) and PAX8+ cells (annotated as renal epithelium) were located lateral to somite structures (Fig. 3e and Supplementary Fig. 11c,d).

We also identified neural crest-like cells (*SOX10*+ and *FOXD3*+), which are absent from conventional mammalian gastruloids, although observed in the EMLO model[22,45]. In mammalian development, multi-potent neural crest cells arise at the dorsal aspect of the neural tube and migrate throughout the embryo[46]. To visualize the spatial distribution of these neural crest-like cells, we immunostained human RA-gastruloids for SOX10. Remarkably, the neural crest-like cells were asymmetrically localized to one side of the neural tube, potentially the dorsal equivalent, suggesting that the spatial patterning of neural crest cells may be recapitulated in human RA-gastruloids (Fig. 3f–j).

To investigate the possibility of dorsal–ventral spatial patterning of the neuroectoderm of human RA-gastruloids more deeply,

we isolated and reanalysed scRNA-seq data from annotated neural tube, neural crest and neural progenitor cells (Extended Data Fig. 8a–d and Supplementary Table 4). A large proportion of neural tube cells expressed dorsal neural tube or roof plate markers, including *PAX3*. In contrast, ventral neural tube or floor plate markers were not expressed. These results suggest that human RA-gastruloids are dorsally biased, similar to mouse TLS and other embryo models[15,47–50]. We speculate that the incomplete establishment of the dorsal–ventral axis is due to the lack of a Sonic hedgehog (SHH)-secreting notochord[51].

Is any aspect of neural differentiation ongoing in human RA-gastruloids? We compared neural differentiation trajectories in human RA-gastruloids to those in cynomolgus monkey embryos at CS11 (ref. 47). We ordered the transcriptomes of neural tube and neural progenitor cells by pseudotime to obtain neural differentiation trajectories for each species. We then identified differentially expressed genes (DEGs) whose expression showed significant changes (false discovery rate (FDR) < 0.05) along a pseudotime axis corresponding to neural differentiation and compared their dynamics (Extended Data Fig. 8e–i and Methods). We observed that 86% of the DEGs in human RA-gastruloids showed conserved dynamics with monkey embryos (Extended Data Fig. 8f,g). Similarly, 83% of the neural differentiation DEGs in monkey embryos show conserved dynamics in human RA-gastruloids (Extended Data Fig. 8h,i). To the extent that differential dynamics are observed, they likely arise from the limitations of this model (for example dorsal bias, lack of anterior neural tube) and/or species differences. Overall, however, our results suggest that human RA-gastruloids may recapitulate at least some aspects of early neural differentiation.

We also performed a more detailed analysis of somites, by reanalysing scRNA-seq data from paraxial mesoderm derivatives (somites, differentiation front and myocytes) for spatial markers of somites' rostrocaudal (*UNCX*; *TBX18*) and dorsoventral (*PAX3*; *PAX1*) organization (Extended Data Fig. 9a,b). Although *PAX3* (dorsal) was strongly expressed in somites, *PAX1* (ventral) was not (Extended Data Fig. 9c,d and Supplementary Table 5), suggesting that like the neural tube, somites in human RA-gastruloids are dorsally biased. Additionally, as seen in both vertebrate embryos[52] and mouse gastruloid and TLS models[10,15], both *UNCX* (caudal) and *TBX18* (rostral) were expressed in a mutually exclusive manner (Extended Data Fig. 9e–h), suggesting that we are reconstituting the rostral–caudal axis of somites. Consistent with this, we identified subsets of somitic cells that may correspond to migratory muscle precursors (*PAX7*+; *MET*+), myotome (*MYF5*+; *MET*+), syndetome (*SCX*+), sclerotome (*PAX9*+, *NKX3-2*+), endotome (*KDR*+) and an unknown cell type (*DNAJC5B*+) (Extended Data Fig. 9d and Supplementary Table 5). It is noteworthy that we observe a subset resembling sclerotome, a ventral somite cell type, in the absence of SHH expression, as it has been reported that its differentiation is promoted by SHH signalling[53,54].

Overall, these scRNA-seq analyses confirm that we are inducing neural tube and segmented somite-like structures in human gastruloids through a discontinuous regime of RA. Although the dorsal–ventral axis is not fully established, human RA-gastruloids contain more advanced cell types than previously achieved in either human or mouse gastruloid models, suggestive of operational signalling gradients along both the dorsoventral (neural crest) and mediolateral (intermediate mesoderm) axes. Particularly in the 120 h human RA-gastruloids, we also observe progression toward even more differentiated cell types (for example neural progenitors, cardiomyocytes, myocytes and renal epithelium) and the appearance of endothelium.

## Computational staging of human and mouse embryo models

The original report of human gastruloids assessed, based on morphological features and Tomo-seq[23], that they model late CS8 to early CS9 in human development[11], which corresponds roughly to embryonic day (E) 7.5 to E8.0 of mouse development. Because we observed more advanced cell types, we hypothesized that human RA-gastruloids might model more advanced stages of human development. However, staging based on morphology, individual cell types, or marker gene expression patterns is somewhat ad hoc. We therefore sought to develop a more systematic framework[55,56] for benchmarking the progression of mammalian embryo models relative to in vivo development.

As a first step, we performed principal-component analysis (PCA) of pseudo-bulk RNA-seq profiles of human gastruloids and RA-gastruloids (96–120 h) along with two available datasets of human embryos at CS7 (ref. 24), CS12 and CS13 (ref. 57) (Fig. 4a; extra-embryonic cell types excluded). While the first principal component (PC)1 seemed to capture differences in scRNA-seq technologies, the second PC (PC2) correlated with developmental time (Fig. 4b–d). The genes most correlated with PC2 included pluripotency related genes (for example *POU5F1*), heterochronic genes (for example *LIN28A*), molecular markers of heart and neural lineages (for example *HES1*, *ZIC1* and *TNNT2*) and HOX genes (for example *HOXC9*, *HOXC6* and *HOXB5*) (Supplementary Fig. 12a,b). Gene Ontology enrichment found positively correlated genes enriched for biological processes such as anterior–posterior pattern specification, epithelial and mesenchyme development and embryonic organ development (Supplementary Fig. 12c,d).

To validate the apparent correlation of this human-derived PC2 with developmental progression, we sought to relate it to in vivo mouse development, where we have access to fine-scale temporal sampling of embryonic development. For mouse E7–E10.5, roughly equivalent to human CS7–CS13 (Supplementary Table 6), scRNA-seq data are available from several sources, staged in either 6 h (ref. 25), 24 h (ref. 58) or single somite[59] increments. We therefore generated pseudo-bulk RNA-seq profiles for staged mouse embryos, as well as mouse gastruloid and TLS models[10,15] (once again excluding extra-embryonic cell types) and then projected these onto the human-derived PC space (Fig. 4a,b). Despite differences with respect to species and technology, mouse embryos staged at 6 or 24 h intervals were ordered by PC2 in a stage-congruent manner (Fig. 4b–d). Human-derived PC2 was also informative at a finer temporal scale, as it was highly correlated with somite counts of E8.5 mouse embryos[59] (Spearman's correlation, 0.83; Fig. 4d). Overall, these results suggest that PC2 is capturing an aspect of developmental progression that generalizes from human to mouse, at least for the developmental window considered.

Although CS7 human embryos aligned with E7.0 mouse embryos, consistent with previous stage assessments, CS12–CS13 embryos were placed beyond E10.5 mouse embryos (Fig. 4e), suggesting that anatomical versus molecular alignments of mouse and human developmental stages may not be fully concordant (Supplementary Table 6). Conventional human gastruloids[11] at 72 h mapped to E7.75–E8 mouse embryos, corresponding to CS8–CS9, consistent with the original staging[11] (Fig. 4e). Both mouse gastruloids[10] and TLS[15] models at 120 h mapped to E8.5 mouse embryos (Fig. 4e). Finally, human RA-gastruloids at 120 h mapped to E9.5 mouse and CS11 macaque embryos (Fig. 4e).

We next sought to stage individual cell types within a given model. To test this concept, we took major cell types from mouse embryos (E8.5 to E10.5) and projected each onto the human-derived PC2 axis. Encouragingly, the computational staging of nearly all of these lineages was reasonably correlated with ground-truth developmental progression, despite the species difference (Supplementary Fig. 13a–d). We therefore extended this approach to various individual cell types from human gastruloids[15]. The results generally agreed with those obtained at the level of pseudobulked gastruloids, with cell types from conventional human gastruloids[11] at 96 h mapping to earlier than E8.5 and cell types from human RA-gastruloids at 120 h mostly mapping to around E9.5 (Supplementary Fig. 13e).

Overall, these results show how 'computational staging' based on scRNA-seq data can be used to benchmark the developmental progression of mammalian embryo models.

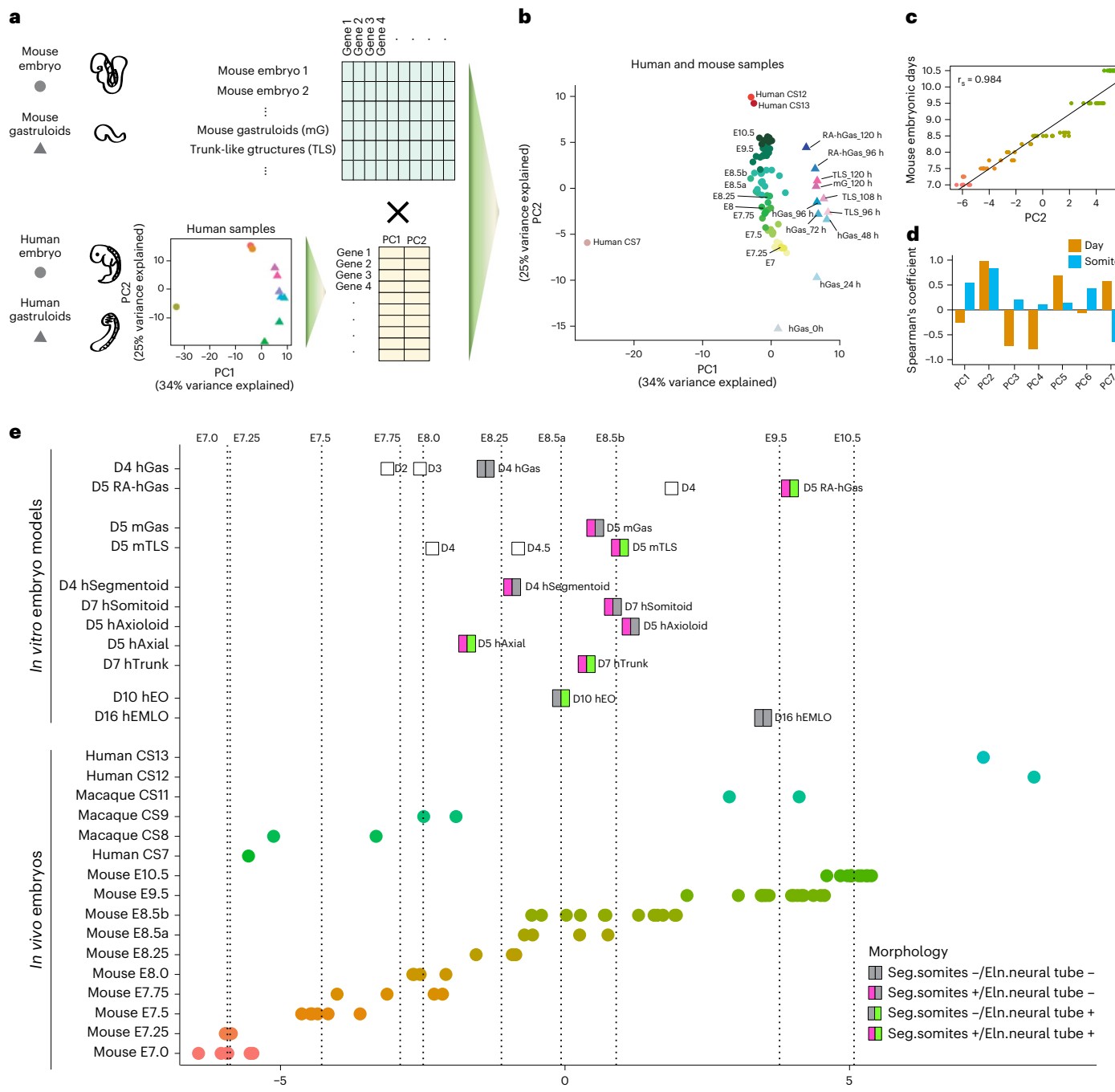

**Fig. 4 | Computational staging of human RA-gastruloids and other mammalian synthetic embryo models. a**, Schematic of strategy for computational staging. In brief, PCA on human samples defines a PC correlated with developmental progression (PC2). Projection of data from tightly staged mouse embryo data onto this human-defined PC enables staging of the relative progression of synthetic embryo models across species and systems. **b**, Projection of data from pooled mouse gastruloids (mG, 120 h), pooled mouse TLSs (96, 108 and 120 h) and individual[58,59] or pooled[80] mouse embryos (E7.0–E10.5) onto PC space defined by the analysis of human data (CS, Carnegie stage human embryos; hGas, conventional human gastruloids; RA-hGas, human RA-gastruloids). **c**, Scatter-plot and Spearman's correlation of mouse embryos' PC2 values (x axis) and their embryonic stage (y axis; E7.0–10.5). A fitted regression line from a linear model is plotted. $r_s$, Spearman's correlation. **d**, Spearman's correlation of mouse embryos' PC values for various human-defined PCs, focusing either on mouse embryo day (E7.0–10.5) or somite count (0–12 somites). **e**, Pseudo-bulk transcriptomes of pooled human embryo models

including RA-gastruloids (RA-hGas, 96, 120 h), conventional human gastruloids (hGas, 48, 72, 96 h)[11], human somitoids (hSomitoid, day 7)[16], human segmentoids (hSegmentoid, day 4)[17], human axioloids (hAxioloid, day 5)[18], human axial organoids (hAxial, day 5)[20], human trunk-like organoids (hTrunk, day 7)[19], EMLO gastruloids (hEMLO, day 16)[22], human EOs (hEO, day 10)[45]; pooled mouse embryo models including gastruloids (mGas, day 5)[10] and TLS (mTLS, day 5)[15]; individual human embryos at CS7, CS12 and CS13 (refs. 24,57); individual macaque embryos at CS8, CS9 and CS11 (ref. 47) and individual[58,59] or pooled[80] mouse embryos, projected onto the human-derived PC2. For mouse embryos, E8.5 and earlier samples are pooled embryos profiled with 10x Genomics scRNA-seq[80], whereas E8.5b and later samples are individual mouse embryos profiled with sci-RNA-seq3 (refs. 59,81). The dotted lines indicate the median PC2 values of mouse embryos at each embryonic day. Embryo models are coloured by the presence/absence of the morphologies of segmented somites and/or an elongated neural tube structure.

## Comparison of human RA-gastruloids to other embryo models

The number of human embryo models is increasing, with some more somitic[16–18] and others more neural[22,45] in character. While this paper was in review, two additional models were reported (axial[20] and trunk[19] organoids), which like RA-gastruloids exhibit more balanced somitic versus neural representation. We leveraged published and newly generated scRNA-seq data to compare these and other models with human RA-gastruloids with respect to developmental progression (Fig. 4e) and the cell types represented (Extended Data Fig. 10). Somitic embryo models, including d7 human somitoids, d5 axioloids and d4 segmentoids, mapped to E8.5, E8.5 and E8–E8.25 mouse embryos, respectively, with cell types limited to mainly somitic lineages. Although neural embryo models took notably longer to culture, d16 EMLO and d10 EO models mapped to E9.5 and E8.25–E8.5 mouse embryos, respectively, with cell types limited to mainly neural lineages. The recently reported human models, d5 axial organoids[20] and d7 trunk-like organoids[19], mapped to E8.0–E8.25 and E8.5 mouse embryos, respectively and lacked many of the more advanced cell types present in d5 human RA-gastruloids (for example renal epithelium, myocyte, gut). The d5 human RA-gastruloids mapped to E9.5 under the same framework and consistent with that, contained the greatest diversity of cell types (somites, neural tube, NMP, neural crest, neural progenitors, myocytes, cardiac myocytes, intermediate mesoderm, renal epithelium, epithelium and gut endoderm) (Fig. 4e and Extended Data Fig. 10).

Embryo models can also differ with respect to the efficiency and consistency with which they form. From a morphological perspective, human RA-gastruloids form embryo-like morphological structures with greater efficiency than other embryo models (89% for human RA-gastruloids (both neural tube and segmented somites) (Fig. 2c) versus 50% for mouse TLS[15] (segmented somites) versus 4% for Matrigel-embedded mouse gastruloids[10] (striped Uncx4.1 expression) versus 20–50% for EMLO[22] (elongation) versus 40% for human trunk-like organoids[19] (clustered somites)). To further evaluate consistency in terms of cell type compositional variance, we 'hashed' individual human RA-gastruloids before scRNA-seq with sci-Plex[60] (Fig. 5a–f). The resulting data shows that individual human RA-gastruloids exhibit less individual-to-individual variation than mouse gastruloids[61], particularly with respect to the balance of the somitic versus neural tube lineages (Fig. 5g–h). Looking forward, we anticipate that inter-individual consistency will be important for arrayed screens, wherein substantial technical variation in the rate of gastruloid formation and/or cell type composition would compromise statistical power and the interpretation of results.

Overall, these results show that human RA-gastruloids exhibit more advanced developmental progression (Fig. 4e) and greater cell type diversity (Extended Data Fig. 10) than contemporary embryo models. They also give rise to posterior embryo-like morphological structures more robustly (Fig. 2c) and exhibit less inter-individual variation with respect to cell type composition (Fig. 5).

## Perturbation of WNT and BMP signalling in RA-gastruloids

We next sought to evaluate human RA-gastruloids as a system for perturbing canonical signalling pathways. WNT signalling is known to play a key role in somite formation[62]. To explore the role of WNT signalling in RA-gastruloids, we reintroduced CHIR at 48 h (Supplementary Fig. 14a). Consistent with mouse TLS and human somitoids[15,16], this resulted in an excess of somite-like structures along the entire anterior–posterior axis (Supplementary Fig. 14b) and gastruloids nearly twice as long (Supplementary Fig. 14c). The result supports a role for WNT signalling in somite segmentation and tail elongation[63,64] that is broadly conserved in vertebrate embryogenesis, through to humans.

BMP signalling plays important roles in lineage segregation during early development. To perturb BMP signalling in RA-gastruloids, we added either LDN193189 (LDN; a BMP inhibitor) or BMP4 at 48 h

onwards (Fig. 6a). With LDN, the neural tube-like structure was consistently longer, while with BMP4, it was shorter and posteriorly confined (Fig. 6b,c). Of note, early LDN treatment (0–24 h) had a greater effect in terms of boosting neural tube length (Supplementary Fig. 15).

To evaluate the role of BMP signalling more broadly, we performed scRNA-seq on LDN-treated RA-gastruloids. We detected vastly more NMPs and neural tube cells than in the untreated condition (Fig. 6d–f and Supplementary Fig. 16a). These observations support previous reports that BMPs may restrict the area within which NMPs can arise to the tailbud region[27,28,65,66]. With LDN-mediated BMP inhibition, this restriction is removed, potentially allowing for expansion of the NMP pool. On the other hand, LDN treatment also resulted in a paucity of other advanced cell types, including IMM, renal epithelium, cardiac cells, neural crest cells and myocytes (Fig. 6d–f and Supplementary Fig. 16a), suggesting a positive role for BMP signalling in their induction. Consistent with this, these same cell types expressed ID genes (direct targets of BMP-SMAD signalling) in untreated human RA-gastruloids (Supplementary Fig. 16b).

To investigate these effects further, we separately examined the consequences of BMP inhibition on neural or somitic lineages. In neural lineages, we observed increases in not only neural tube and NMP markers, but also tailbud markers (FGF17, WNT5a)[67,68], supporting the possibility that LDN-mediated BMP inhibition may promote the generation of tailbud NMPs (Fig. 6g–i). However, in addition to a decrease in neural crest marker, we also observed marked decreases in PAX3 (dorsal) and PAX7 (dorsal) expression, consistent with the established role of BMP in dorsalizing neural cells[66], as LDN may be counteracting the dorsal bias of untreated human RA-gastruloids. A similar phenomenon is observed in the somitic lineages, where we observe a modest decrease in PAX3 (dorsal) expression and a marked increase in PAX1 (ventral) expression (Fig. 6j–l).

In the somitic lineages, marker gene reductions also confirm the marked depletion of myocytes upon BMP inhibition (Fig. 6j–l). We also observe a decrease in recently characterized markers of the endotome (KDR1, EBF2 and ETV2)[69,70] (Fig. 6m). In normal development, the endotome is thought to migrate to the dorsal aorta and specify haematopoietic stem cells, but its developmental regulation is poorly understood[71]. Notably, MEOX1, an essential homeobox transcription factor for the specification of the endotome[69], remains unaffected by BMP inhibition (Fig. 6k,l). This suggests that BMP signalling may be required for the regulation of endotome development downstream of MEOX1; alternatively, these may be secondary effects of other alterations to somites upon BMP inhibition.

## Perturbation of key transcription factors in RA-gastruloids

We next sought to evaluate human RA-gastruloids as a system for genetic perturbations. For proof-of-concept, we chose the transcription factors TBX6 and PAX3, as their roles in early development are well characterized in mouse models[72,73]. To knock out these genes, we introduced CRISPR-Cas9 RNA–protein complexes (RNPs) to human PS (hPS) cells and subsequently induced RA-gastruloids (Fig. 7a and Supplementary Fig. 17).

In mice, Tbx6 is an established master regulator of NMP fate, promoting somitic differentiation and suppressing neuronal lineages via repression of a Sox2 enhancer[73,74]. Relative to non-targeting-control (NTC)-RNP RA-gastruloids, TBX6-KO RA-gastruloids exhibited a shorter body length (Fig. 7b and Supplementary Fig. 18). Upon performing scRNA-seq, we observed a gross excess of SOX2+ neural cells (including neural crest) in TBX6-KO RA-gastruloids (Fig. 7c–e and Supplementary Fig. 19). In contrast, presomitic mesoderm was almost entirely absent. We speculate that this is a secondary effect of the loss of presomitic mesoderm, which may contribute to the appropriate posterior environment for definitive endoderm in the tailbud.

To further characterize the regulatory programmes underlying the shift of NMPs toward neural fates in TBX6-KO RA-gastruloids, we

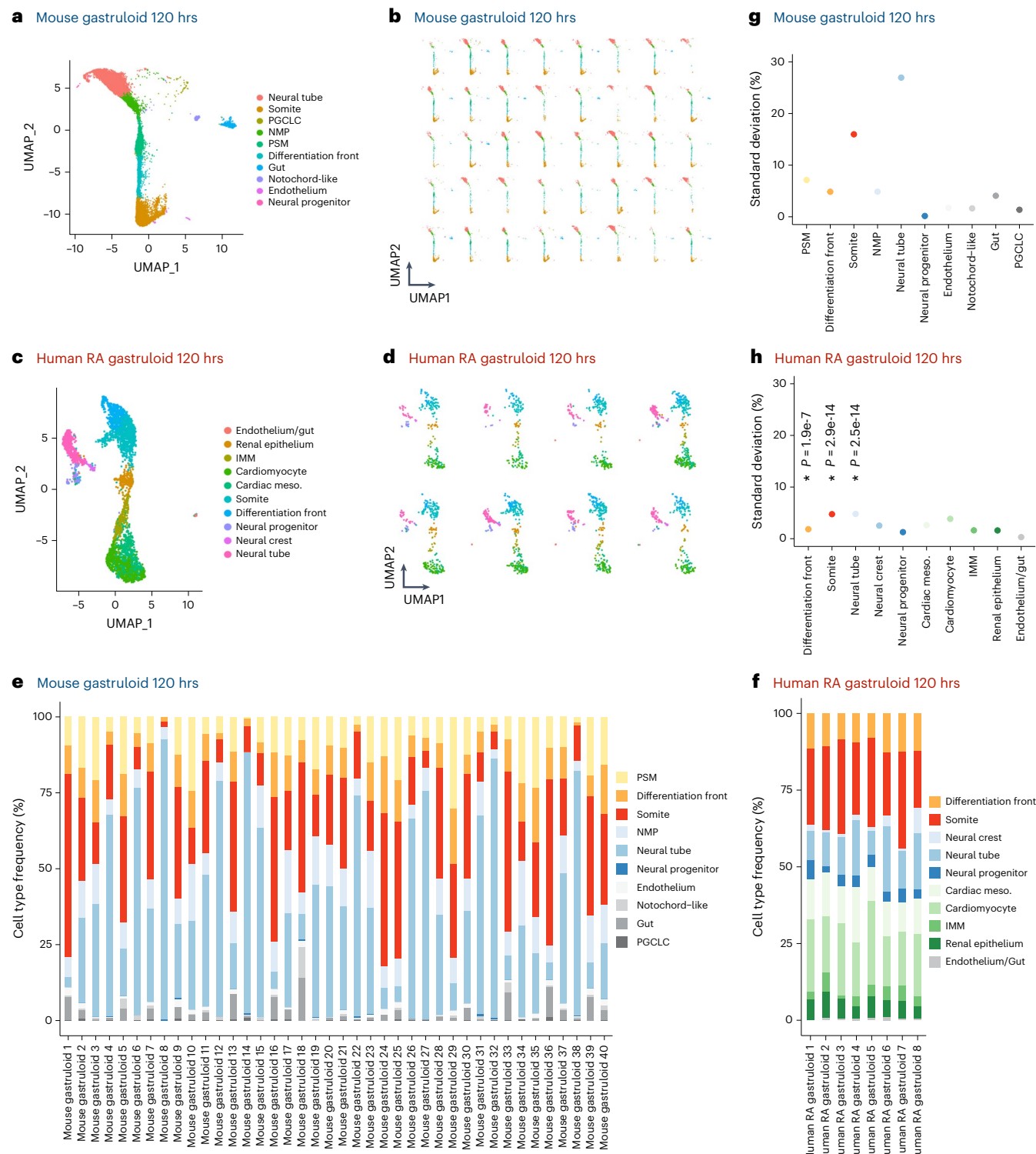

**Fig. 5 | Inter-individual variation in cell type composition in mouse gastruloids versus human RA-gastruloids. a,b**, UMAP visualization of cell types observed in mouse gastruloids at 120 h, based on data from GSE212050 (ref. 61). In **a**, all individuals are represented, whereas in **b**, the same UMAP projection shows the contribution of individual gastruloids. **c,d**, Same as **a,b**, but based on individual human RA-gastruloids with scRNA-seq data obtained by sci-Plex[60]. **e,f**, Frequency with which individual cell types are observed in individual mouse gastruloids at 120 h (**e**) or individual human RA-gastruloids at 120 h (**f**). **g,h**, The s.d. of cell type proportions in individual mouse gastruloids at 120 h (**g**) or individual human RA-gastruloids at 120 h (**h**). In brief, to account for differences in the total number of cells per individual gastruloid, we randomly sampled 100 cells from each individual gastruloid, ten times per individual, to

generate pseudo-replicates. For three cell types abundantly present in both models (differentiation front, somites, neural tube), we performed an analysis of variance (ANOVA) to test for significant differences in s.d. values between human and mouse samples. NPCs were excluded from the analysis because although present in both models, on average less than one NPC per individual mouse gastruloid was detected, precluding variance analysis. If the ANOVA was significant ($P < 0.05$), a post hoc Tukey's honest significant difference (HSD) test was conducted to further evaluate pairwise differences between the species. For all three cell types compared, variation across individuals was significantly lower in human RA-gastruloids than mouse gastruloids (differentiation front, $P = 1.89 \times 10^{-7}$; somites, $P = 2.9 \times 10^{-14}$; neural tube, $P = 2.5 \times 10^{-14}$).

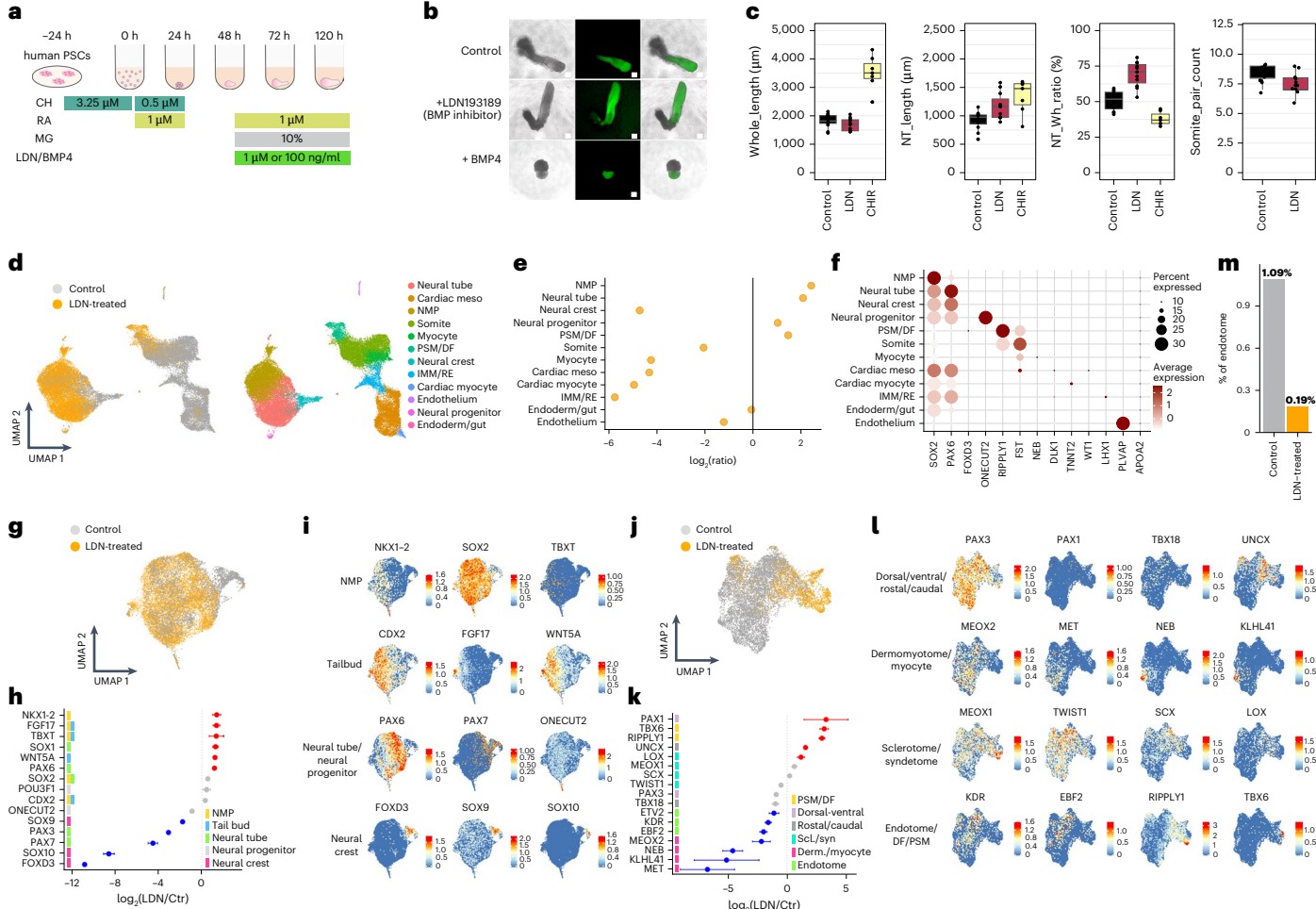

**Fig. 6 | Effects of perturbing BMP signalling on human RA-gastruloids.**
**a**, Schematic of perturbation of BMP signalling in human RA-gastruloids. CHIR, CHIR99021; LDN/BMP4, LDN193189 or BMP4. **b**, Representative images of untreated control, LDN-treated and BMP4-treated RA-gastruloids. The experiments were repeated independently three times with similar results. Scale bar, 100 μm. **c**, Morphometrics. From left to right: full length (μm) of 120 h RA-gastruloids, neural tube (NT) length (μm) measured with SOX2-mCit signal, neural tube/full-length ratio and somite pair counts. $n = 6$ and 11 for untreated and LDN treatment, respectively. **d**, UMAP visualization of co-embedded scRNA-seq data from untreated (grey) and LDN-treated (yellow) 120 h human RA-gastruloids (left). Same UMAP labelled by cell type annotation (right). **e**, Cell type composition changes upon LDN treatment of human RA-gastruloids. **f**, Marker gene expression in each cell type. **g–i**, The effects of BMP inhibition on neural lineages (NMP, neural tube, neural crest and neural progenitor cells).

UMAP visualization of co-embedded scRNA-seq data (neural lineages only) from untreated (grey) and LDN-treated (yellow) 120 h human RA-gastruloids (**g**). Changes in marker gene expression with LDN treatment in neural lineages (**h**). Colour bars at left indicate the cell type(s) for which each gene is a marker. Dots indicate the ratio of average expression in LDN-treated versus control gastruloids. Data are presented as mean ± s.e.m. across pseudo-replicates ($n = 3$). Dots corresponding to increases or decreases larger than twofold are coloured red and blue, respectively. Same UMAP projection as **g** (**i**). Gene expression of three marker genes for each neural cell type are shown in each row. **j–l**, Same as **g–i**, but restricting instead to somitic lineages with pseudo-replicates ($n = 3$). Data are presented as mean ± s.e.m. **m**, Effects of LDN treatment on the proportion of endotome cells. Endotome cells are defined as the subset of somitic cells that are both $KDR2^+$ and $EBF2^+$. The bar chart shows the proportion of endotome cells, out of all somitic cells, in two experimental conditions.

examined DEGs in neural lineages (Supplementary Fig. 20a–d,f,h). As expected, many genes involved in neural development were upregulated. Perhaps less expectedly, additional transcription factors normally expressed in the posterior tailbud, including CDX2 and various HOX genes (*HOXA7*, *HOXA9*, *HOXB9*, *HOXC6* and *HOXC8*), were among the most downregulated genes in the neural lineages of TBX6-KO RA-gastruloids, relative to the same lineages in controls. Whether the expansion of neural lineages in TBX6-KO RA-gastruloids is contributed to by transdifferentiation from somitic lineages as seen in mouse models[15,27] is an outstanding question, potentially addressable with chimeric WT/KO RA-gastruloids, lineage tracing[75] or by a combined knockout-reporter assay[15].

Turning to the PAX3-KO RA-gastruloid, although morphologically similar to controls, clear differences were observed upon scRNA-seq profiling. This included an increased proportion of neural cells but a decreased proportion of neural crest (Fig. 7c–g and Supplementary Fig. 19). This is consistent with a previous finding reporting dysregulation of neural crest development in *Pax3*-deficient or Splotch mutant mice[76,77]. Again focusing on neural lineages, we examined DEGs in PAX3-KO RA-gastruloids (Supplementary Fig. 20e,g,i). Consistent with the loss of neural crest, sharply downregulated genes included neural crest markers (for example *SOX9*) as well as drivers of the epithelial-to-mesenchymal transition (for example *SNAI2*). Upregulated genes included *PAX7*, *PAX6* and *SFRP2*, normally expressed in dorsal progenitor domains 3–6 of the embryonic caudal neural tube[78,79], suggesting that the dorsal bias of human RA-gastruloids may be mitigated by knockout of PAX3. Altogether, these observations are consistent with pleiotropic roles for PAX3 in early human development, for example including in neural crest cell development, EMT and dorsal–ventral patterning.

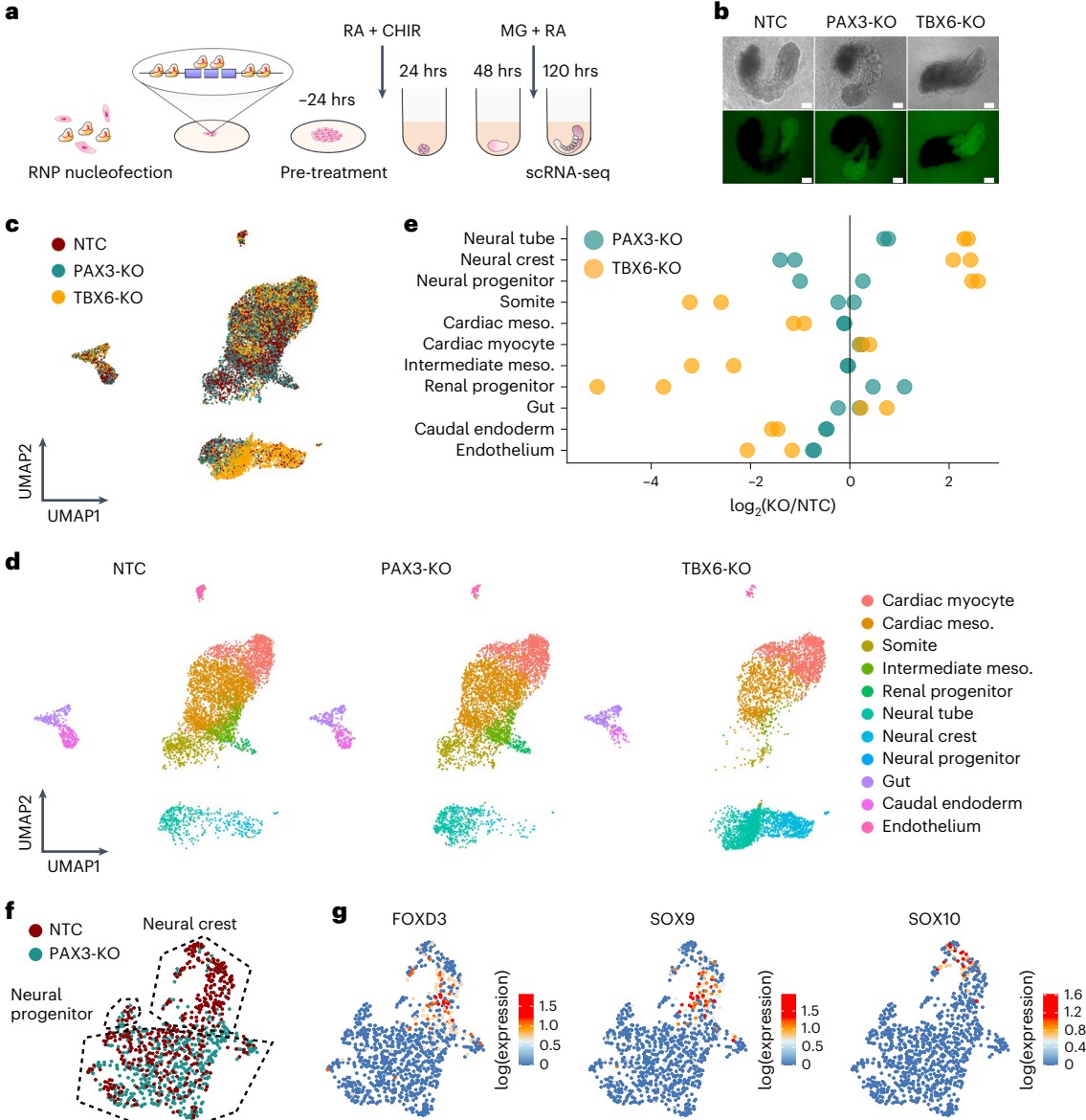

**Fig. 7 | Genetic perturbation of transcription factors in human RA-gastruloids. a**, Schematic of transcription factor knockouts in human RA-gastruloids using CRISPR/Cas9 RNPs. Six Cas9-gRNA RNPs were nucleofected into PS cells, inducing indels or full deletion of exons of *PAX3* or *TBX6*. Nucleofected PS cells were subjected to RA-gastruloid induction protocol. RA-gastruloids were collected at 120 h after cell aggregation. **b**, Representative images of non-targeting-control (NTC), PAX3-KO, TBX6-KO RA-gastruloids. *n* = 96 gastruloids for each condition. Scale bar, 100 μm. **c**,**d**, UMAP of scRNA-seq data from NTC, PAX3-KO and TBX6-KO RA-gastruloids, labelled by genotype (**c**) or cell type (**d**). **e**, Cell type composition changes upon knockout of each transcription factor. log$_2$ (KO/NTC) fold changes for each cell type are shown as dots. Vertical black line corresponds to no change in the proportion of the cell type between KO and NTCs. **f**, Annotated UMAP embedding of neural cell types only (neural tube, neural crest and neural progenitor cells) from PAX3-KO RA-gastruloid and controls. **g**, Same embedding as **e**, but showing gene expression (log-scaled) of neural crest marker genes.

Overall, we conclude that RA-gastruloids may serve as a versatile model for studying early human post-implantation development, via both chemical and genetic perturbations.

## Discussion

Here we show that early but limited RA supplementation, together with Matrigel, robustly induces human gastruloids with embryo-like morphological structures, including a neural tube and segmented somites. The early RA pulse seems to balance human NMPs, such that they will later adopt both mesodermal and neural fates. Human RA-gastruloids also progress to a more advanced developmental stage than conventional gastruloids and consistent with that, contain more advanced cell types, including neural crest, neural progenitors, renal progenitors and myocytes.

Why do conventional human gastruloids, which possess both NMPs and presomitic mesoderm (Fig. 1), fail to form segmented somites even in Matrigel, whereas Matrigel alone is sufficient to support somitogenesis in mouse gastruloids[1,2]? We speculate that there is a minimum threshold on the size of the axial progenitor pool at the onset of Matrigel-induced somitogenesis. Consistent with this hypothesis, the number of seeded cells is a critical factor in human RA-gastruloid morphogenesis (Fig. 2d and Extended Data Fig. 3c). To assess this further, we examined the cell type composition of conventional human gastruloids versus axioloids, somitoids and segmentoids, based on scRNA-seq data[5–7]. At 48 h, axioloid, somitoid and segmentoid models were entirely composed of PSMs (91%, 92% and 82%, respectively) and NMPs (9%, 8% and 18%, respectively), a trend conserved in mouse

gastruloid/TLS[1,8] (Supplementary Fig. 21a–c). However, conventional human gastruloids had far fewer PSMs (~9%) or NMPs (~2%) and a high proportion of cardiac mesoderm-like cells (38%). The latter expressed several BMPs (Supplementary Fig. 21d), which could negatively impact axial progenitor maintenance, including of NMPs[9,10], as also observed in the LDN-treated RA-gastruloids (Fig. 6e).

Additional indirect support for our hypothesis can be found in the means by which various groups achieve in vitro human somitogenesis by adding BMP and/or NODAL inhibitors to the pre-treatment medium, such as SB431542 and DMH1 (somitoid[6]), SB431542 (axioloid[5]) or LDN183189 (segmentoid[7]) (Supplementary Table 7). As BMP and NODAL signals promote differentiation toward extra-embryonic lineages and endoderm[11,12], suppressing these signals during pre-treatment may protect the axial progenitor pool from depletion or exhaustion before the onset of somitogenesis. However, the early inhibition of these signalling pathways might also restrict progenitors' future potential. In contrast, we found an early RA pulse actively induces bipotent NMPs[13] during pre-treatment. By avoiding BMP or NODAL inhibitors, we might have inadvertently facilitated the retention of progenitors' potential to later diversify into more advanced cell types, for example lateral plate mesoderm[14], intermediate mesoderm[15], neural crest[16] and myocytes[17].

Overall, our results suggest that initial/early conditions, including the number of seeded stem cells as well as their signalling state, strongly shape the self-organization potential of gastruloids and thus what structures and cell types will ultimately develop even at much later time points[18]. Of note, it remains possible that the dependence on seeding size is due to the impact of gastruloid size on morphogen concentrations and/or gradations, in addition to (or rather than) the absolute number of cells.

While we found 'computational staging' to be a powerful approach for benchmarking mammalian embryo models against one another as well as in vivo development, there are limitations. First, this approach may be restricted to the specific developmental window from which the human PC2 was constructed (roughly corresponding to E7–E10.5 of mouse development). It is possible that this window could be extended, both earlier and later, with an improved reference dataset. Second, as we used human gastruloids to 'fill' temporal windows for which human embryo data were not available while constructing PC space, cell types absent from gastruloids might be problematic to computationally stage. For example, the expression of the anterior-most HOX genes are weaker than posterior HOX genes in gastruloids (Extended Data Fig. 7d), although the contribution of HOX genes to PC staging seems minimal (Supplementary Fig. 12e,f). Altogether, we conclude that computational staging offers a robust and standardized framework for assessing the relative developmental progression of mammalian embryo models, including across cell types and species. Although scRNA-seq does not capture the spatial patterning of embryos/embryo models, we can imagine future versions of computational staging that incorporate spatial transcriptomic information.

There are tremendous possibilities on the horizon for leveraging embryo models to decode early human embryogenesis and to model certain developmental disorders (such as neural tube defects). Our results show that compared with alternative embryo models, human RA-gastruloids exhibit more advanced developmental progression (Fig. 4e), greater cell type diversity (Extended Data Fig. 10), more robust formation of embryo-like morphological structures (Fig. 2c, Extended Data Figs. 5 and 6 and Supplementary Fig. 10) and less inter-individual variation (Fig. 5). As we consider how best to scale reverse genetic approaches to study the phenotypic consequences of thousands of specific coding and regulatory mutations on early human development, these attributes of human RA-gastruloids are attractive.

## Online content

Any methods, additional references, Nature Portfolio reporting summaries, source data, extended data, supplementary information,

acknowledgements, peer review information; details of author contributions and competing interests; and statements of data and code availability are available at https://doi.org/10.1038/s41556-024-01487-8.

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

# Methods

## Ethics statement

Our research on the induction and molecular analysis of human conventional and RA-gastruloids was reviewed and approved by the Embryonic Stem Cell Research Oversight of the University of Washington (E0047-001) and is in compliance with the principles laid out by the International Society for Stem Cell Research Guidelines for Stem Cell Research and Clinical Applications of Stem Cells[82]. No new experiments involving human embryos were performed in this study, and all natural human embryo data analysed were obtained from publicly available datasets.

## Statistics and reproducibility

For box plots in this study, upper whisker, upper box edge, bar and lower box edge represent 1.5 × interquartile range, third quartile (Q3), median and first quartile (Q1), respectively. No statistical methods were used to pre-determine sample sizes but our sample sizes are similar to those reported in previous publications[11,15]. No experimental data were excluded from the analyses. Sequencing data exclusion criteria is outlined in the Methods, including filtering out the substandard data in single-cell measurements, following the general practice in the field. Human conventional gastruloids and RA-gastruloids used in experiments were randomly selected from each time point before sample preparation. The investigators were not blinded to allocation during experiments and outcome assessment. Data distribution was assumed to be normal but this was not formally tested.

## Human pluripotent stem cell culture

RUES2-GLR human ES (hES) cells were gifted by A. Brivanlou (Rockefeller University). The H9 hES cells were obtained from WiCell. WTC11 human iPS (hiPS) cells were gifted by B. Conklin (Gladstone Institutes). RUES2-GLR hES cells were maintained in StemFlex medium (Thermo, A3349401) on Geltrex (Thermo, A1413201). H9 hES cells and hiPS cells were maintained in Essential 8 Flex medium (Thermo, A2858501). hPS cells were routinely passaged using StemPro Accutase (Thermo, A1110501) to new Geltrex-coated wells as recommended by the manufacturer. For the first 24 h after passaging, hPS cells were cultured in the medium with 10 μM Y-27632 (Sellek, S1049) to prevent apoptosis[83].

## Conventional human gastruloid induction

For conventional human gastruloid induction, we carefully followed the method described in previous reports[11,84]. Then, 4 × 10⁴ hES cells were seeded on 0.5 μg cm⁻² Vitronectin-coated 12-well plate (Gibco, A14700) in Nutristem hPS cell XF medium (Biological Industries, 05-100-1 A) with 10 μM Y-27632. On day 1, the medium was replaced with Nutristem containing 5 μM Y-27632. On day 2, the medium was replaced with Nutristem. On day 3, the medium was replaced with Nutristem containing 3.25 μM CHIR99021 (CHIR, Millipore, SML1046). As CHIR is relatively unstable and can exhibit batch-to-batch inconsistency, we aliquoted it into 10 μl volumes, kept at −20 °C for up to 3 months and used freshly thawed CHIR for each gastruloid induction. After 24 h, cells were detached by StemPro Accutase and dissociated into single cells. Then, 400 cells were seeded to 96 wells with 40 μl Essential 6 medium (Thermo, A1516401) containing 0.5 μM CHIR and 5 μM Y-27632. After 24 h, 150 μl Essential 6 medium was added to each well. After 48 h, 150 μl of the medium was removed with a multi-channel pipette and 150 μl Essential 6 medium was added. This medium change was repeated at 72 h. Human conventional gastruloids were collected or discarded by 96 h.

## Human RA-gastruloid induction

Detailed step-by-step protocols can be found at https://doi.org/10.17504/protocols.io.261ge5epog47/v1. Then, 2 × 10⁴ hES cells were seeded on 0.5 μg cm⁻² Vitronectin-coated 12-well plate (Gibco, A14700) in Nutristem hPS cell XF medium (Biological Industries, 05-100-1A) with 10 μM Y-27632. On day 1, the medium was replaced with Nutristem containing 5 μM Y-27632. On day 2, the medium was replaced with Nutristem. On day 3, the medium was replaced with Nutristem containing 3.25 μM CHIR (Millipore, SML1046). On day 4, cells were detached by StemPro Accutase and dissociated into single cells. Then, 4,000–5,000 cells were seeded into 96 wells with 50 μl Essential 6 medium (Thermo, A1516401) containing 1 μM CHIR, 500 nM RA (Millipore Sigma, R2625) and 5 μM Y-27632. At 24 h, 150 μl of Essential 6 medium was added to each well. After 48 h, 150 μl of the medium was removed with a multi-channel pipette and 150 μl of Essential 6 medium containing 5% Matrigel and 100 nM RA was added and maintained in a 37 °C, 5% CO₂ incubator until 120 h. As with conventional gastruloids, we used freshly thawed CHIR for every RA-gastruloid induction.

For RA-gastruloid induction from the H9 cell line, 4 × 10⁴ H9 hES cells were seeded on 0.5 μg cm⁻² Vitronectin-coated 12-well plate in Essential 8 medium with 10 μM Y-27632. On day 1, the medium was replaced with Essential 8 containing 5 μM Y-27632. On day 2, the medium was replaced with Essential 8 containing 5 μM CHIR. On day 3, the medium was replaced with Essential 8 containing 500 nM RA, 5 μM CHIR and 10 μM SB431542 (Selleck chemicals, S1067). On day 4, cells were detached by StemPro Accutase and dissociated into single cells. Then, 4,000–5,000 cells were seeded into 96 wells with 50 μl NDiff 227 medium (Takara, Y40002) containing 1 μM CHIR and 5 μM Y-27632. At 24 h, 150 μl of NDiff 227 medium was added to each well. After 48 h, 150 μl of the medium was removed with a multi-channel pipette and 150 μl NDiff 227 medium containing 5% Matrigel and 100 nM RA was added and maintained in a 37 °C, 5% CO₂ incubator until 120 h. Human RA-gastruloids were collected or discarded by 120 h.

For RA-gastruloid induction from the WTC11 cell line, 2 × 10⁴ WTC11 hiPS cells were seeded on 0.5 μg cm⁻² Vitronectin-coated 12-well plate in Essential 8 medium with 10 μM Y-27632. On day 1, the medium was replaced with Essential 8 containing 5 μM Y-27632. On day 2, the medium was replaced with Essential 8 containing 5 μM CHIR. On day 3, the medium was replaced with Essential 8 containing 500 nM RA, 5 μM CHIR and 10 μM SB431542 (Selleck chemicals, S1067). On day 4, cells were detached by StemPro Accutase and dissociated into single cells. Then, 4,000–5,000 cells were seeded into 96 wells with 50 μl NDiff 227 medium containing 1 μM CHIR and 5 μM Y-27632. After 24 h, 150 μl NDiff 227 medium was added to each well. After 48 h, 150 μl of the medium was removed with a multi-channel pipette and 150 μl of NDiff 227 medium containing 5% Matrigel and 100 nM RA was added and maintained in a 37 °C, 5% CO₂ incubator until 120 h.

## Chemical perturbation

To perturb signalling pathways in human RA-gastruloids, we added 1 μM LDN193189 (STEMCELL Technologies, 72147), 100 ng ml⁻¹ BMP4 (R&D, 314-BP-010) or 3 μM CHIR to the medium at 48 h along with 5% Matrigel and 100 nM RA.

## Genetic perturbation with CRISPR/Cas9 RNPs

RNP complexes were prepared according to the manufacturer's procedures. In brief, equal molar amounts of crRNA and tracrRNA (IDT, 1072532, listed in Supplementary Table 8) were mixed and heated at 95 °C for 5 min in a thermal cycler and kept at room temperature for 10–20 min for hybridization. AltR-Cas9 protein (IDT, 1081058) was added to the crRNA and tracrRNA mixture to assemble Cas9 RNPs. RUES2-GLR hES cells were trypsinized with StemPro Accutase and the reaction was quenched with StemFlex medium supplemented with 10 mM Y-276322. Then, 2 × 10⁵ cells were transferred to a new tube and centrifuged at 250g for 5 min. Cells were resuspended in 20 μl nucleofection buffer (16.4 μl Nucleofector Solution + 3.6 μl Supplement) provided in P3 Primary Cell 4D-Nucleofector X kit S (Lonza, V4XP-3032). After the addition of 3 μl RNP and 0.5 μl of AltR-Cas9 Electroporation Enhancer (IDT, 1075915), cells were transferred to 16-well Nucleocuvette Strips and nucleofected with the CA-137 programme. The nucleofected cells were transferred to a 12-well plate that contained Nutristem with 10 mM Y-27632 and after 24 h, the medium

was replaced with Nutristem without Y-27632. Cells were maintained until they reached 50–70% confluence. Then, RNP-introduced cells were transferred onto 0.5 µg cm$^{-2}$ Vitronectin-coated 12-well plates. We then proceeded to the RA-gastruloid induction steps described above and collected at 120 h for scRNA-seq analysis.

## Immunostaining of gastruloids

For whole-mount immunostaining, gastruloids were fixed in 4% paraformaldehyde overnight at 4 °C, washed with PBST (0.2% Tween 20), soaked in blocking buffer (PBS containing 0.1% BSA and 0.3% Triton X-100) overnight at 4 °C and then incubated with primary antibodies diluted with blocking buffer overnight at 4 °C. The samples were washed with washing buffer (PBS containing 0.3% Triton X-100), incubated with secondary antibodies and DAPI overnight at 4 °C and washed and mounted in Fluoro-KEEPER antifade reagent (Nacalai). For phalloidin staining, Alexa Fluor 647 phalloidin (A22287; Thermo) was added to the secondary antibody at a dilution of 1:400. All samples were analysed with an LSM710 (Zeiss) or LEICA SP8X (Leica) confocal microscope with negative control samples where the primary antibodies were not added. The antibodies used in this study are listed in Supplementary Table 9.

## In situ hybridization chain reaction

HCR and following measurements of UNCX4.1 and TBX18 were performed as described previously[10,16,18,85]. RA-gastruloid was fixed with 4% PFA at 4 °C overnight. After the wash with PBST (PBS + 0.1% Tween 20), RA-gastruloid are dehydrated with MeOH and kept in −20 °C for longer storage. Probes are designed by Molecular Instruments on TBX18 (accession NM_001080508.3, hairpin B1), UNCX4.1 (accession NM_001080461.3, hairpin B2), HOXA1 (accession NM_005522.5, hairpin B1), HOXA5 (accession NM_019102.4, hairpin B2) and HOXA10 (accession NM_018951.4, hairpin B3). Hairpin B1 was labelled with Alexa 488, B2 was labelled with Alexa 546 and B3 was labelled with Alexa 647. Images were taken with the Nikon A1R system. For the quantification of the TBX18/UNCX4.1 staining, Fiji with ImageJ v.2.14.0 was used. Intensities of each channel were quantified along the longitudinal axis of RA-gastruloids with the segmented line tool, with a 50-line width. Subsequently, a locally estimated scatterplot smoothing (LOESS) function was applied to smooth the intensity data with the parameter $f = 0.075$ in R. Intensity data were normalized to their respective means to facilitate direct comparisons. Differences between TBX18 and UNCX4.1 intensities were then calculated, plotted and used for the peak detection using findpeaks function from the pracma package with a minimum peak distance of 60. Plots were made with the ggplot package.

## Morphometric quantification

Imaging of both conventional and RA-gastruloids was carried out using a DMi8 inverted microscope (Leica). Subsequent image analysis was performed with the ImageJ software package, specifically the Fiji distribution. Before quantification, images underwent level and contrast adjustments to enhance the visibility of structural features. For each experimental condition and time point, at least three gastruloids ($n \geq 3$) were subjected to analysis. Data from the morphometric quantification were visualized via custom R code and are available on our GitHub site (https://github.com/shendurelab/Human-RA-Gastruloid/).

## Neural tube quantification

For neural tube analysis, the segmented line tool in Fiji was used to measure the length of the SOX2-positive area. The centre of this area was also determined with the same tool. Neural tube width was quantified at three distinct positions along the length of the SOX2-positive area: 10%, 50% and 90%, from the posterior to the anterior end, using the straight-line tool. The widths obtained at these positions were averaged to calculate the average neural tube width for each gastruloid. The length-to-width ($L{:}W$) ratio of the neural tube was calculated by dividing the length by the average width.

## Whole body quantification

For whole-body morphometric analysis, the segmented line tool was employed to measure the total length of the area captured in the bright-field. Whole-body width was measured at three standardized positions (10%, 50% and 90%) along the length from posterior to anterior ends and these measurements were averaged to calculate the average whole body width. The $L{:}W$ ratio for the whole body was determined similarly to the neural tube $L{:}W$ ratio.

## Somite quantification

For the morphometric analysis of somites, the length, width and area were measured on one side of the somites based on their order from the posterior end. Length and width were quantified using the straight-line tool and the area was measured with the polygon selection tool. To assess the frequency of somite pairing in RA-gastruloids, first, we checked whether there are two rows of somite structures within a single RA-gastruloid. Then, we defined putative somite pairs based on their positional order (for example the third somite from the posterior end on both sides) along the posterior-to-anterior axis. Next, we checked whether members of each putative somite pair have similar sizes (area). Somites with areas exhibiting 70% to 130% similarity were classified as 'paired somites'. This comparison was made for three randomly chosen putative somite pairs within each gastruloid. A gastruloid was subsequently designated as 'paired gastruloid' if at least two out of three putative somite pairs were classified as 'paired somites'.

## Quantification of epithelialization

The 120 h RA-gastruloids were immunostained with phalloidin with anti-SOX1 antibody and anti-PAX3 antibody (listed in Supplementary Table 9). We defined a 'gastruloid with epithelialized somite or neural tube' as ones in which at least one epithelialized somite or neural tube that exhibited accumulation of phalloidin staining at the apical side of its SOX1 or PAX3-positive cells. $n = 11$ and 10 for gastruloids with a second RA pulse and without a second RA pulse, respectively.

## Live imaging of human RA-gastruloids

After supplementation with Matrigel at 48 h, RA-gastruloids were promptly placed under a microscope (DMi8, Leica). They were subsequently stored in a humidified $CO_2$-chamber for an additional 48 h. Images were taken at consistent 30-min intervals. To achieve a comprehensive representation of the whole RA-gastruloid structure, tiled images were later combined and converted to movies using a custom macro. The custom macro is available on GitHub (https://github.com/shendurelab/Human-RA-Gastruloid/). Based on the movie, segmentations were manually determined based on the emergence of the boundaries of somites at the posterior region.

## Single-cell RNA-seq of gastruloids

Gastruloids ($n = 48$–96) were collected from 96 wells using a micropipette, transferred to PBS (−) to wash out the medium and incubated with 0.05% trypsin-EDTA for 8 min at 37 °C. After quenching the reaction by the addition of 1:10 volume of FBS, gastruloids were dissociated into single cells by repeated pipetting. For the 0 h sample, pre-treated PS cells were washed with PBS (−) once and dissociated into single cells with StemPro Accutase. Dissociated single-cell suspensions were loaded in Chromium Next GEM Chip G Single Cell kit (Chromium, PN-1000120) on a 10x Chromium controller according to the manufacturer's instructions. Single-cell RNA-seq libraries were generated with Chromium Next GEM Single Cell 3′ GEM, Library & Gel Bead kit v.3.1 (Chromium, PN-1000121) and Dual Index kit TT Set A, 96 reactions (Chromium, PN-1000215). Library concentrations were determined by Qubit (Invitrogen) and the libraries were visualized by TapeStation (Agilent). All libraries were sequenced on NextSeq 2000 (Illumina) (read 1, 28 cycles; read 2, 85 cycles; index 1, 10 cycles; index 2, 10 cycles).

## Processing of sequencing reads

Base calls were converted to fastq format using the Cell Ranger v.6.0.0 (ref. 86) mkfastq function. The sequencing reads are then demultiplexed based on i5 and i7 barcodes, mapped to hg38 reference genome and assigned to GRCh38 (GENCODE v.32/Ensembl 98) genes with Cell Ranger v.6.0.0 count function with default settings.

## scRNA-seq analysis

The filtered feature barcode matrices generated by Cell Ranger were loaded into R and converted to Seurat objects with Seurat v.4.1.1 (ref. 87). The thresholds for unique molecular identifier (UMI) counts and feature counts were determined by the valley in the bimodal distribution of UMI counts and feature counts, respectively. Cells showing upper outlier values in the violin plot of UMI counts and feature counts are also excluded to remove potential doublets. To further ensure doublets were removed, Scrublet was used to calculate a doublet score for each cell. Cells with doublet scores higher than the simulated threshold value by Scrublet are potential doublets with two different cell types and were excluded from the dataset. Parameters for filtration of low-quality cells and doublets are described in Supplementary Table 10. As we observed strong cell cycle effects in our dataset after initial dimensionality reduction, we decided to remove cell cycle-related genes[88] (specifically, genes with a prefix *HIST*, *MT*, *TOP*, *CDK*, *CCN*, *CDC*, *CCDC* and *MKI*; genes containing *MALAT*, *AUR* and genes with suffix *NUSAP*, *SMC*, *CENP*, *UBE*, *SGO*, *ASPM*, *PLK*, *KPN*, *RP*, *PTTG*, *SNHG*, *CK*, *BUB*, *KIF*, *KCNQ*, *SMO*, *HMG*, *S100*, *LINC*, *ATP*, *IGFBP*, *HSP*, *FOS* and *JUN*). After quality control, each dataset was log-normalized, and heterogeneity associated with cell cycle stage and mitochondrial contamination was regressed out with SCTransform. The top 3,000 variable genes identified by SCTransform were then used for dimensionality reduction by PCA. The first 30 PCs were used for dimensionality reduction by UMAP. Cells were clustered by graph-based clustering algorithms. In particular, 30 PCs were used to compute the 20-nearest neighbours of each cell to construct the shared nearest neighbour (SNN) graph. Then clusters were identified with the Louvain algorithm with a resolution of 0.8. SNN construction and clustering are implemented through the FindNeighbors and FindClusters functions in the Seurat package with default settings for all other parameters. Samples with two batches (for example 72 h conventional gastruloids, 120 h RA-gastruloids) were integrated with Harmony with the top 2,000 highly variable genes. Time-course scRNA-seq datasets were integrated with Harmony as implemented in Seurat. In brief, datasets from each time point were merged together. The merged dataset was log-normalized and the top 2,000 variable genes were selected for downstream integration. Then, the merged dataset was *z*-transformed and heterogeneity associated with cell cycle stage and mitochondrial contamination were regressed out. Dimensionality reduction by PCA was implemented to generate a low-dimensional embedding of cells in the merged dataset. The low-dimensional embedding was then input into Harmony to correct for batch effects among samples collected from different experiments.

To compare conventional human gastruloid cell types with mouse gastruloids[10]/mouse TLS[15] cell types, a lift-over list of mouse and human genes was downloaded from BioMart-Ensembl (Ensembl Genes 102). The gene features in the mouse and human datasets were converted to a list of human and mouse orthologous pairs. The relationship of multiple homologues to one gene in the other species were retained in this case. After feature conversion, human and mouse gastruloid datasets were integrated with Seurat V3 integration algorithm[89]. Specifically, human and mouse datasets were log-normalized and *z*-transformed separately. The top 3,000 variable genes in both datasets were selected for integration. Thirty canonical correlation analysis (CCA) PCs were used to search for integration anchors. The integrated dataset was then subjected to dimensionality reduction by PCA. The top 30 PCs were used for UMAP visualization with min.dist of 0.3. We used default settings for all the other parameters. We used the same strategy for integration of conventional human gastruloids and mouse embryos (E6.5–E8.5)[25], except that we used reciprocal PCA for dimensionality reduction (reduction method = rpca) for FindIntegrationAnchors to reduce the computing time.

To compare cell types between conventional human gastruloids versus human CS7 embryos[24], we first downsampled each cell type in conventional human gastruloids to 63 cells per cell type because the large discrepancy of cell numbers between gastruloid (8,000–19,000 cells in each dataset) versus embryo (1,195 cells) datasets. We noticed that the large discrepancy of cell numbers could lead to overfitting of cell types between two datasets because dimensionality reduction will be largely driven by the datasets with many more cells. After downsampling, the gastruloid and embryo datasets were merged. The merged dataset was log-normalized and the top 3,000 variable genes selected for downstream integration. Then, the merged dataset was scaled and heterogeneity associated with cell cycle stage and the mitochondrial contamination were regressed out. Dimensionality reduction by PCA was implemented to generate a low-dimensional embedding of cells in the merged dataset. The low-dimensional embedding was then inputted into Harmony to correct for batch effects among samples. Top 30 PCs from Harmony were used for UMAP visualization with min.dist of 0.3.

## Cell type assignment with non-negative least-squares

Cell types in gastruloids were compared with cell types from corresponding lineages in mouse embryos (E8.5–E10.5). Orthologous genes were identified in both datasets and converted to orthologous pairs as described above. Non-negative least-squares regression was applied to predict gene expression in target cell type ($T_a$) in dataset A based on the gene expression of all cell types ($M_b$) in dataset B, $T_a = \beta_{0a} + \beta_{1a}M_b$, based on the union of the 200 most highly expressed genes and 200 most highly specific genes in the target cell type. Datasets A and B were then switched for prediction; that is, predicting the gene expression of target cell type ($T_b$) in dataset B from the gene expression of all cell types ($M_a$) in dataset A, $T_b = \beta_{0b} + \beta_{1b}M_a$. Finally, for each cell type a in dataset A and each cell type b in dataset B, two correlation coefficients, $\beta = 2(\beta_{ab} + 0.001)(\beta_{ba} + 0.001)$, were combined to obtain a statistic for which high values reflect reciprocal, specific predictivity. The $\beta$ scores were then subjected to a *z*-score scaling across all selected cell types.

## Comparative analysis of neural differentiation trajectories

Cynomolgus monkey scRNA-seq data was downloaded from the Gene Expression Omnibus (GEO) (GSE193007). Spinal cord cells from CS11 in the first batch were selected for later comparison. To accommodate the fact that the neural tube in human RA-gastruloids are dorsally biased, as reported in other gastruloid models (for example TLS[15]), we excluded neural tube cells of monkey embryos with ventral markers (*PHOX2B*+, *FOX2A*+ and *TFF3*+) from this analysis. Highly variable genes for trajectory analysis were calculated following methods in the original study. After log-normalization, we selected 2,000 highly variably expressed genes by the FindVariableFeatures function in Seurat with 'mean.var. plot'. Spinal cord cells were then sub-clustered with default settings of Seurat and annotated using marker genes from the original study. The raw count matrix was then imported into Monocle2 for pseudotime analysis. The raw count matrix was normalized by size factors. The 2,000 highly variable genes were selected for dimensionality reduction with DDRtree. The cells were then ordered by orderCells in Monocle2 and each assigned a pseudotime value. By examining distribution of cell types along the resulting trajectory, only the trajectory that leads to neural progenitor cells were selected for DEG analysis. The DEG analysis was performed with the VGAM package by modelling gene expression as a smooth, non-linear function of pseudotime. Genes with *q*-value less than 0.05 were defined as DEGs along the neural differentiation trajectory of cynomolgus monkeys.

Neural differentiation trajectories were inferred for human RA-gastruloids in the first batch with the same methods as cynomolgus

monkeys using the 2,000 highly variable genes calculated from the scRNA-seq analysis step. DEGs of neural differentiation trajectories in human RA-gastruloids were computed in the same way as for cynomolgus monkey embryos.

The dynamics of DEGs in cynomolgus monkey embryos embryo and human RA-gastruloids were compared between species. In particular, DEGs from one species were clustered by hierarchical clustering (ward.D2) into two groups (upregulated and downregulated) according to their dynamics in cynomolgus monkey embryos and human RA-gastruloids. Based on their group assignments in both species, DEGs were then classified into six categories: up–up, down–down, up–down, down–up, up–unexpressed and down–unexpressed. 'Unexpressed' means that transcripts of this DEG were not detected in the other species.

### Staging of in vitro embryo models

We downloaded the following data; 10x Genomics scRNA-seq data from CS12–16 human embryos (GSE157329)[57], Smart-seq2 scRNA-seq data from a CS7 human embryo (http://www.human-gastrula.net)[24], 10x Genomics scRNA-seq data from CS8–11 cynomolgus monkey embryo (GSE193007)[47], 10x Genomics scRNA-seq data from mouse embryos E6.5–8.5 (E-MTAB-6967)[80], sci-RNA-seq data from mouse embryos E8.5–13.5 (GSE186069 and GSE186068)[58,59], 10x Genomics scRNA-seq data from mouse gastruloids (GSE123187)[10], TLS (GSE141175)[15], somitoids (ERX7494538)[16], axioloids (GSE199576)[18], segmentoids (GSE195467)[17], EMLO (GSE166603)[22] and EO (GSE155383)[45]. As human gastruloids are composed of only embryonic lineages, we excluded cells derived from extra-embryonic tissues from these analyses. Extra-embryonic tissues were excluded from all embryo datasets. As described above, cell cycle and mitochondrial genes were removed. As CS7 data, which were generated with the Smart-seq2 technology[24], had notable variations in library depth between single cells and did not have the UMI counts, additional normalization with Seurat V4.1 (ref. [87]) NormalizeData function was performed. After this normalization, the read distribution for each cell approximated a Poisson distribution with a similar library size range as UMI counts from other datasets.

Where possible, all embryo datasets were pseudobulked by embryo. Specifically, all UMI counts connected to a particular embryo sample (or embryo sample, in the case of pools) were added together for each gene. Human and mouse gastruloids were pseudobulked by sample (as these derive from multiple embryos). Specifically, all UMI counts connected to a particular gastruloid sample were added together for each gene.

As we are mainly focusing on human early embryonic development, highly variable genes from human embryo developmental datasets[24,57] were used for downstream analyses. For CS12–16 embryos, highly variable genes were calculated based on Poisson distribution[90] and a total of 1,361 highly variable genes were obtained. We then performed an intersection of CS12–16 embryos' highly variable genes with features in CS7 embryos, human and mouse gastruloid samples. The intersection resulted in a total of 447 human genes for dimensionality reduction with PCA. The pseudo-bulk transcriptomes of human embryos CS12–13, human gastruloids (0–96 h) and human RA-gastruloids (96–120 h) were used to construct a PC space with these 447 human genes. Specifically, the pseudo-bulk matrix was normalized by sequencing depth, log-transformed and z-transformed by Seurat V4.1 (ref. [87]). PCA was performed on the normalized matrix. To accomplish cross-species comparison, we then converted the 447 human genes to human and mouse orthologous pairs as described above. The raw count pseudo-bulk matrix of mouse embryos, cynomolgus macaque embryos, human embryo models and mouse embryo models were combined with human raw count matrix. The resulting matrix was normalized as described above and projected onto the human-derived PC space by multiplying the loadings of the human-defined PCs to the normalized matrix. We then computed the Spearman's correlation between ground-truth developmental time and PC values of mouse embryo samples for the top seven PCs. Based on the highest correlation of mouse developmental time (somite counts and embryonic days) and PC values, PC2 was selected to represent developmental time. The normalized expression of genes in each sample were subjected to Pearson's correlation with the PC2 embeddings of the sample to obtain top genes highly correlated or anti-correlated with PC2.

### Lineage-specific staging of in vitro models

All embryo datasets were separated into cell types based on annotations from the original papers. Where possible, each cell type were pseudobulked by embryo sample (or embryo samples, in the case of pools). Specifically, all UMI counts connected to a particular cell type in a particular embryo sample were aggregated for each gene. Human and mouse gastruloids were also separated by annotated cell types. Each cell type is pseudobulked by gastruloid samples (as these derive from multiple embryos). Specifically, all UMI counts connected to a particular cell type in a particular gastruloid sample were aggregated for each gene.

We then projected mouse embryo cell types at E8.5–10.5 onto the PC space constructed from human whole embryo samples and human gastruloids, described above. Human or mouse gastruloid cell types were projected to the same PC space to evaluate their progression relative to related cell types.

Evaluation of inter-variation of cell type composition of human RA-gastruloids and mouse gastruloids based on single-cell RNA-seq from individual gastruloids

scRNA-seq data for individual 120 h human RA-gastruloid were obtained by sci-Plex scRNA-seq method[60]. The 120 h mouse gastruloid data were downloaded from the publicly available dataset GSE212050. To account for differences in the total number of cells per individual gastruloid, we performed random sampling. Specifically, 100 cells were randomly sampled from each individual gastruloid in both the human and mouse datasets. This sampling was performed independently ten times per gastruloid to generate pseudo-replicates. For each cell type, within each replicate and species, the s.d. of the frequencies was calculated. An ANOVA was subsequently performed for each cell type to test for significant differences in s.d. values between human and mouse samples. If the ANOVA was significant ($P < 0.05$), a post hoc Tukey's HSD test was conducted to further evaluate pairwise differences between the species. All statistical analyses were executed using a native stats package in R.

### Reporting summary

Further information on research design is available in the Nature Portfolio Reporting Summary linked to this article.

## Data availability

Raw sequencing data and processed single-cell datasets that support the findings in this paper are available in the GEO under accession GSE208369. The hg38 reference genome was used for mapping human-derived sequencing reads. Reanalyses of previously published datasets were also deposited to GEO, again under accession GSE208369. These published datasets were originally downloaded from the following sources: 10x Genomics scRNA-seq data from CS12–16 human embryos (GSE157329)[57], Smart-seq2 scRNA-seq data from a CS7 human embryo (http://www.human-gastrula.net)[24], 10x Genomics scRNA-seq data from CS8–11 cynomolgus monkey embryos (GSE193007)[47], 10x Genomics scRNA-seq data from mouse embryos E6.5–8.5 (E-MTAB-6967)[80], sci-RNA-seq3 data from mouse embryos E8.5–13.5 (GSE186069 and GSE186068)[58,59], 10x Genomics scRNA-seq data from mouse gastruloids (GSE123187)[10], TLS (GSE141175)[15], somitoids (ERX7494538)[16], axioloids (GSE199576)[18], segmentoids (GSE195467)[17], EMLO (GSE166603)[22] and EO (GSE155383)[45]. All other data supporting the findings of this study are available from the

corresponding author on reasonable request Source data are provided with this paper.

## Code availability

Code for analysing the single-cell datasets and performing morphometric measures is available at https://github.com/shendurelab/Human-RA-Gastruloid/.

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

## Acknowledgements

We thank all members of the Shendure Laboratory and D. Kimelman, as well as members of the Allen Discovery Center for Cell Lineage Tracing, for helpful feedback and discussions; A. Boulgakov for helping with cell culture experiments; A.M. Arias, N. Moris, A. Alemany and S.C. van den Brink for helpful discussions; A.H. Brivanlou for providing RUES2-GLR ES cells; the Tom & Sue Ellison Stemcell Core of the Institute for Stem Cell and Regenerative Medicine and the KECK microscopy centre of the University of Washington. This work was supported by a grant from the Paul G. Allen Frontiers Group (Allen Discovery Center for Cell Lineage Tracing to J.S.), Alex's Lemonade Stand Foundation (to J.S.) and the National Human Genome Research Institute (HG010632 and HG011586 to J.S.), as well as philanthropic support from the Brotman Baty Institute for Precision Medicine. J.S. is an Investigator of the Howard Hughes Medical Institute.

## Author contributions

N.H. and J.S. designed the research. W.Y., C.K. and N.H performed experiments. W.Y. and N.H. performed computational analysis. C.K., C.Q. and S.S. assisted with data analysis. S.P., N.B. and D.F. assisted with live imaging. B.K.M., R.K.G., S.G.R., C.L., R.M.D. and S.S. assisted with the interpretation of results. E.N. assisted with the imaging of gastruloids. J.S., W.Y. and N.H. wrote the paper with input from all authors.

## Competing interests

J.S. is a scientific advisory board member, consultant and/or co-founder of Prime Medicine, Cajal Neuroscience, Guardant Health, Maze Therapeutics, Camp4 Therapeutics, Phase Genomics, Adaptive Biotechnologies, Scale Biosciences, Sixth Street Capital, Pacific Biosciences and Somite Therapeutics. The other authors declare no competing interests.

## Additional information

**Extended data** is available for this paper at https://doi.org/10.1038/s41556-024-01487-8.

**Correspondence and requests for materials** should be addressed to Nobuhiko Hamazaki or Jay Shendure.

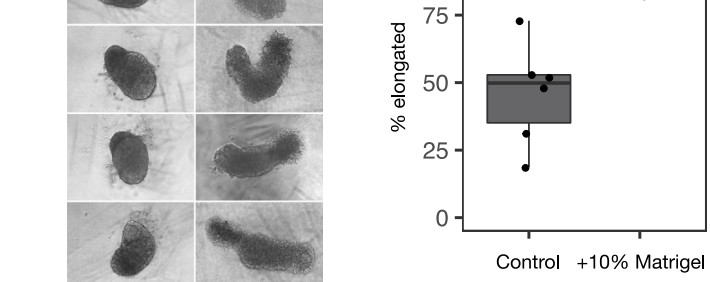

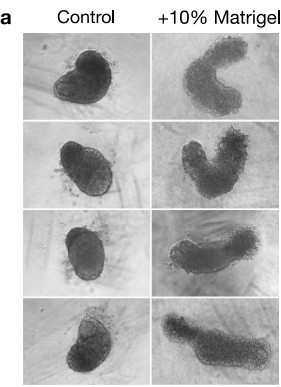

**Extended Data Fig. 1 | Effects of Matrigel on human gastruloid morphology.**
**a**, Representative images of human gastruloids without (left) vs. with (right) 10%
Matrigel. The addition of 10% Matrigel enhances the extent of human gastruloid
elongation. **b**, Boxplot showing the proportion of elongated gastruloids

observed in the absence (left) vs. presence (right) of 10% Matrigel across a
total of ten experiments. Raw counts are provided in Supplementary Table 1.
The addition of 10% Matrigel enhances the robustness of human gastruloid
elongation.

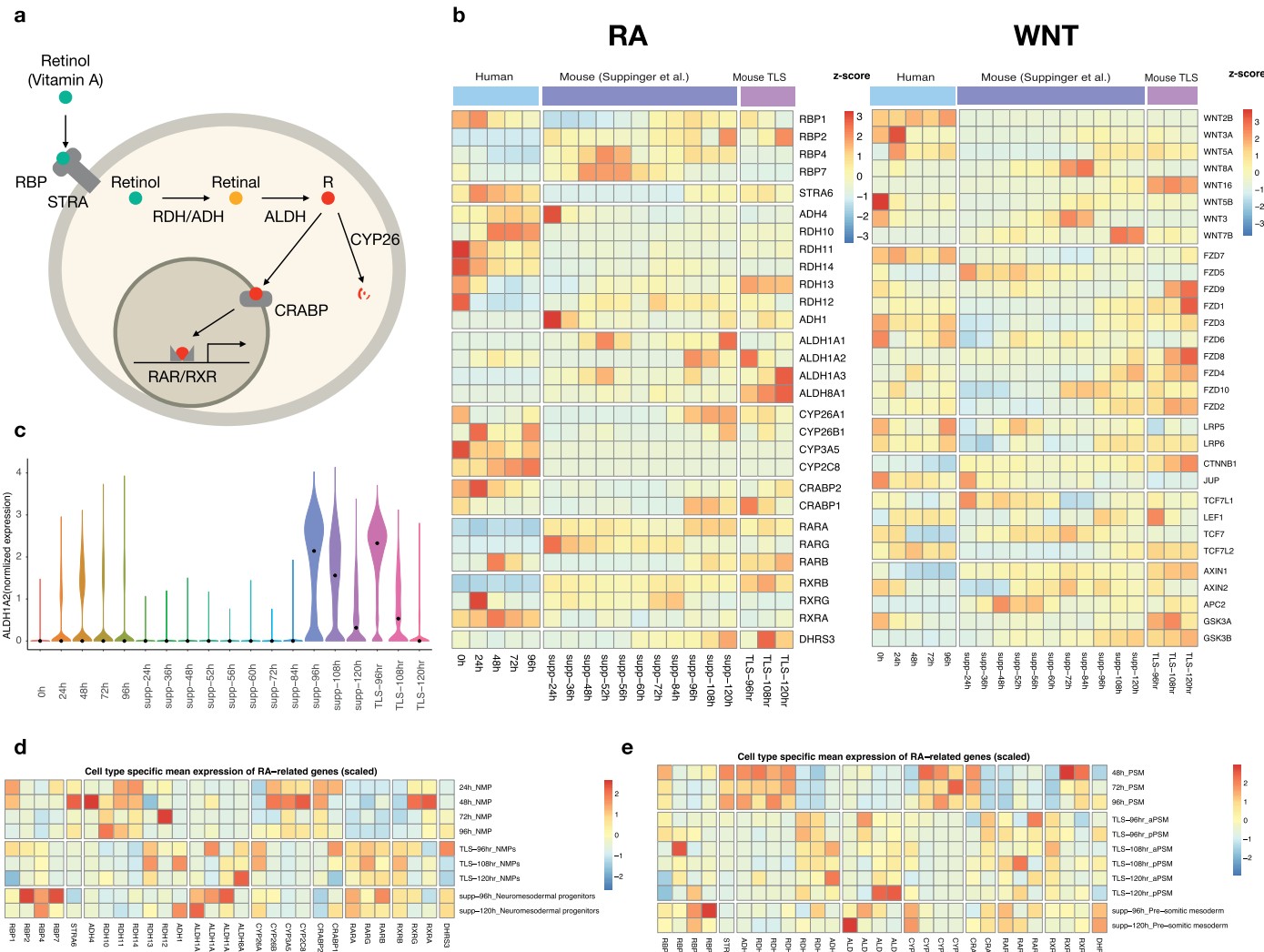

**Extended Data Fig. 2 | Differential expression of genes related to RA/Retinol/Retinal pathways in mouse vs. human gastruloids. a**, Schematic of RA metabolism pathways. **b**, Heatmaps showing expression levels of genes involved in RA (left panel) and WNT (right panel) signalling pathways in conventional human gastruloids at 0–96 h from Moris et al.[11], mouse gastruloids at 24–120 h from Suppinger et al.[91] and mouse TLS 96–120 h from Veenvliet et al.[15] Gene expression values were scaled by z-score across samples. **c**, Normalized

expression of *ALDH1A2* in conventional human gastruloids at 0–96 h, mouse gastruloids at 24–120 h[91], and mouse TLS at 96–120 h[15]. **d-e**, Heatmaps showing expression levels of genes involved in RA signalling pathways in NMPs (**d**) and PSMs (**e**) of conventional human gastruloids at 0–96 h, mouse TLS at 96–120 h[91] and mouse gastruloids at 96–120 h[15]. Gene expression levels are shown for timepoints where NMPs and PSMs were detected in a given model.

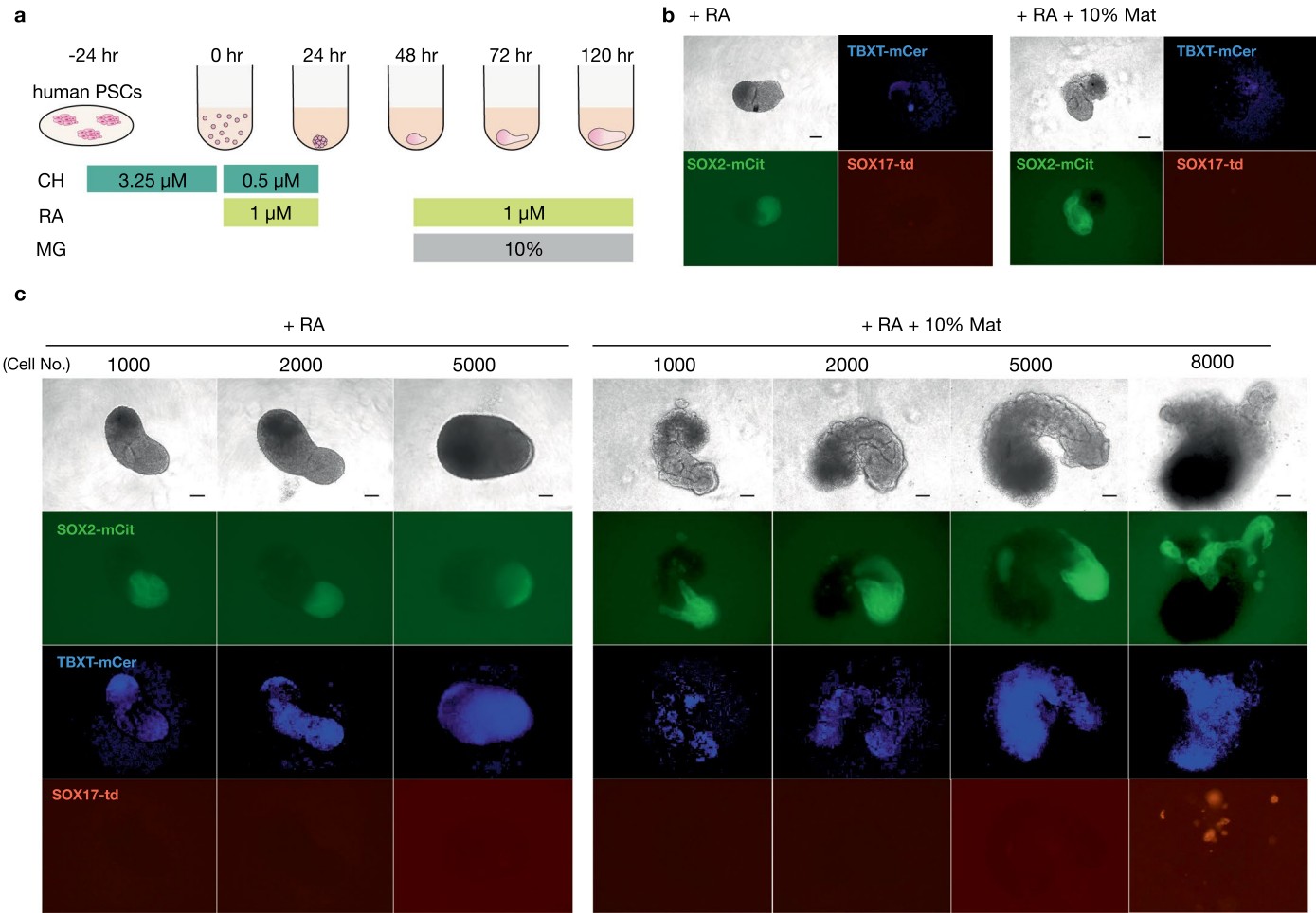

**Extended Data Fig. 3 | The number of seeded cells impacts human gastruloid formation in the context of a discontinuous regimen of retinoic acid. a**, Schematic of discontinuous regimen of RA and Matrigel treatment while inducing human gastruloids. RA, retinoic acid; MG, 10% Matrigel. **b**, Representative images of 96 h human gastruloids induced from 400 cells.

1 μM RA (0–24 h and 48–96 h) (left) or 1 μM RA (0–24 h and 48–96 h) + 10% Matrigel (48–96 h) (right) were added to the medium. Scale bars, 100 μm, N = 32. **c**, Representative images of 96 h human RA-gastruloids while varying the number of cells used for initial seeding. Scale bars, 100 μm, N = 48.

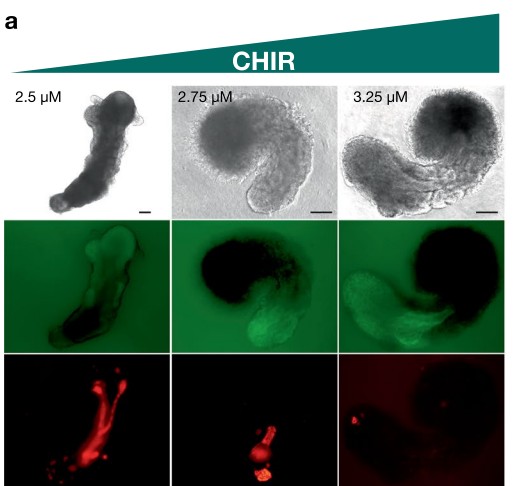

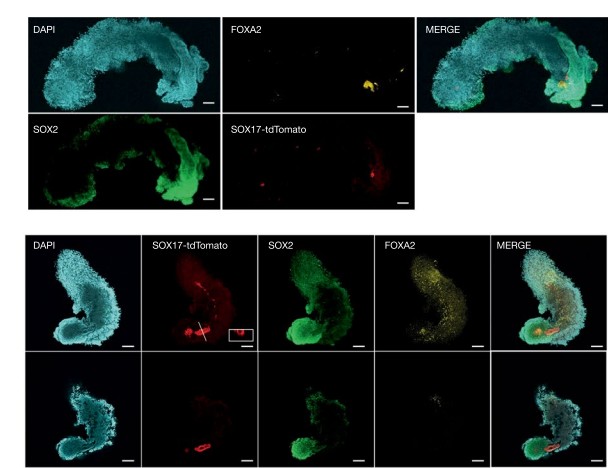

**Extended Data Fig. 4 | Lower CHIR concentrations facilitate formation of elongated gut tube-like structures in human RA-gastruloids. a**, Effects of CHIR concentration at the pre-treatment stage on SOX17-tdTomato positive cell accumulation and elongation. Scale bars, 200 μm N = 32, 38, and 48, respectively. **b**, Immunostaining of 3.25 μM CHIR-treated 120 h human RA-gastruloids with anti-SOX2, anti-SOX17-tdTomato, and anti-FOXA2 antibodies. Scale, 100 μm,

N = 8 out of 12 showed a similar staining pattern. **c**, Immunostaining of 2.75 μM CHIR-treated 120 h human RA-gastruloid with anti-SOX2, anti-SOX17-tdTomato, and anti-FOXA2 antibodies. (Top) Max projection of z-stack image. (Bottom) A slice of z-stack. Scale bars, 100 μm. N = 5 out of 7 gastruloids showed a similar staining pattern.

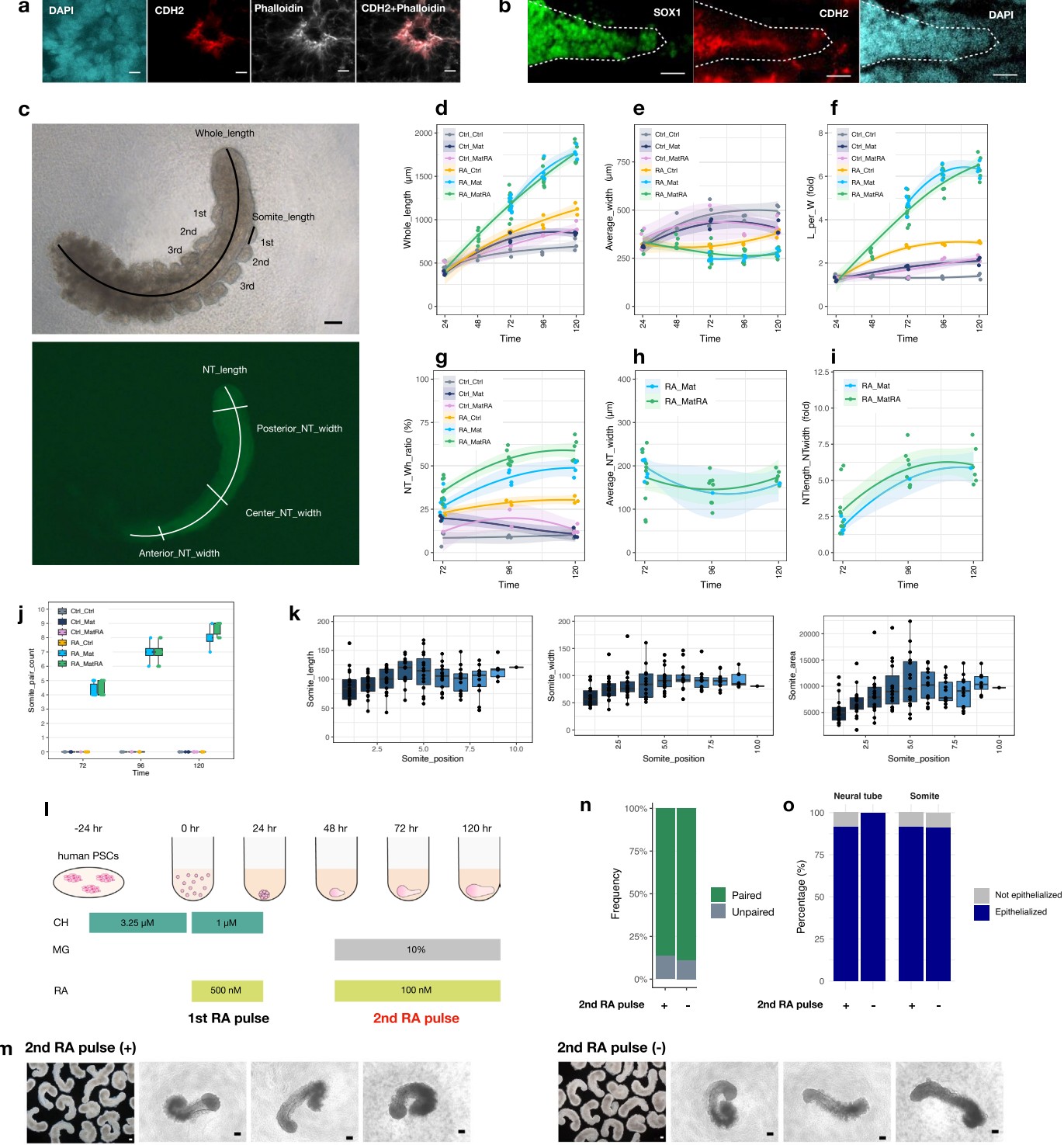

**Extended Data Fig. 5 | See next page for caption.**

**Extended Data Fig. 5 | Morphological properties of human conventional and RA-gastruloids. a**, Immunostaining of N-cadherin (CDH2) and phalloidin in somites in an RA-gastruloid. Phalloidin-stained F-actin and CDH2 were co-localized and highly concentrated at the apical surface of somites. Scale bar, 10 μm. **b**, Immunostaining of N-cadherin (CDH2) and SOX1 in the neural tube in an RA-gastruloid. Scale bar, 100 μm. **c**, (Top) Bright-field of a human RA-gastruloid. The whole length was measured as the length of a line along the centre of the body. Each somite length was measured from the posterior end. (Bottom) SOX2-mCit view of the top picture. Neural tube length (NT_length) was measured as the continuous SOX2+ area. The width of neural tubes was measured and averaged over several positions (10%, 50%, 90% along the full length of the structure). Scale bar, 100 μm. **d-l**, Morphometric measurements of gastruloids which originated from 5,000 cells, as a function of time. Ctrl, no treatment controt; RA, Retinoic acid; Mat, 5% Matrigel; MatRA, Matrigel + RA. Left and right part of each text label indicates the conditions at 0–24 h and at 48–120 h, respectively. For example, Ctrl_Ctrl indicates no treatment for both 0–24 h and 48–120 h. N = ≥3 for each time point and condition. **d**, Whole length (μm) of gastruloids. **e**, Average width (μm) of gastruloids. **f**, Ratio (%) of whole length to average width. **g**, Ratio (%) of length of neural area to the whole length. **h**, Average neural tube width (μm).

**i**, Ratio (%) of neural tube length-to-width. **j**, Number of somites observed as a function of time. **k**, Length, width, and area of somites as a function of position. N = 16 RA-gastruloids. **l**, Schematic of RA-gastruloid induction protocol, highlighting the first vs. second RA pulse. **m**, Bright-field images of human RA-gastruloids induced with (left column) vs. without (right column) inclusion of the second RA pulse. Scale bars = 100 μm. **n**, Frequency of paired somites in RA-gastruloids with vs. without inclusion of the second RA pulse. Somites with areas within 30% of one another were classified as "paired somites". This comparison was made for 3 randomly chosen putative somite pairs within each gastruloid. A gastruloid was subsequently designated as "paired gastruloid" if at least 2 out of 3 putative somite pairs were classified as "paired somites". N = 13/14 (92.9%) and N = 11/12 (91.7%) for RA-gastruloids with vs. without inclusion of the second RA pulse, respectively. **o**, Frequency of neural tube (left) and somite (right) epithelialization with vs. without inclusion of the second RA pulse, respectively. Epithelization was defined by the accumulation of phalloidin staining at the apical side of the structures upon immunostaining. The percentages indicate the frequency of gastruloids with epithelialized somite or neural tube. N = 11 and N = 10 for RA-gastruloids with vs. without inclusion of the second RA pulse, respectively.

**a**

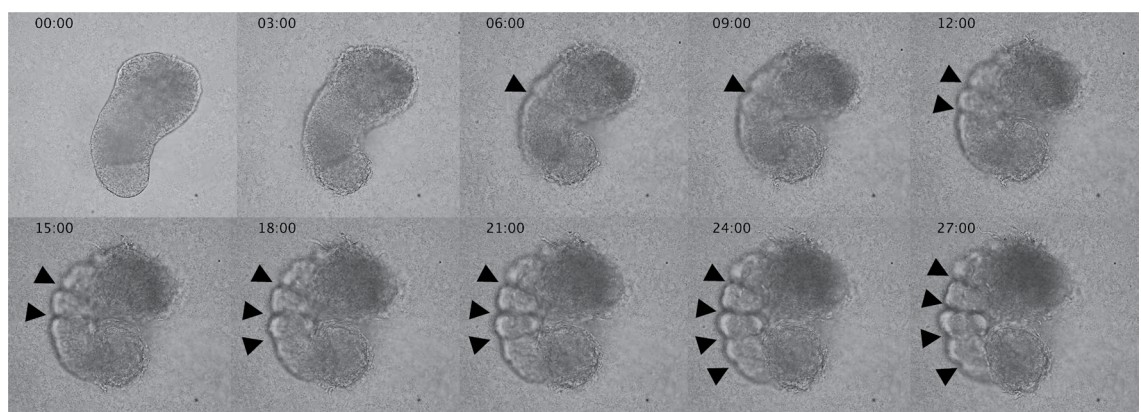

**b**

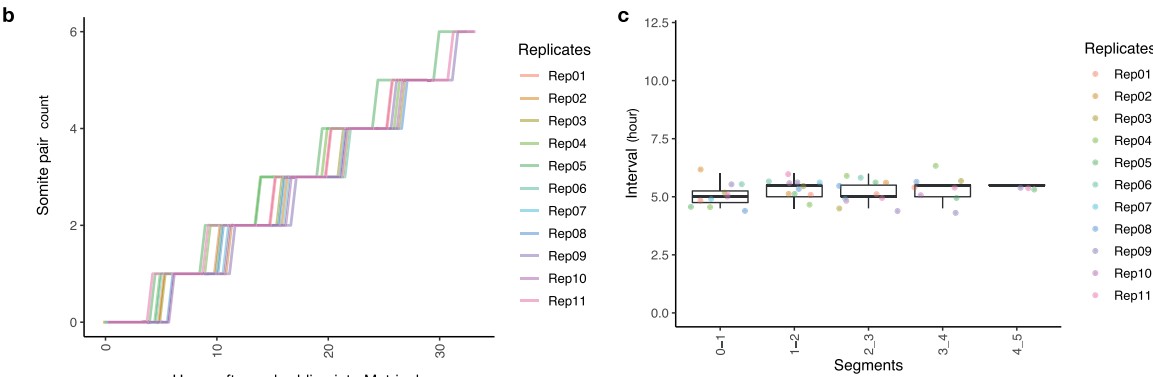

**Extended Data Fig. 6 | Live imaging of human RA-gastruloids. a**, Snapshots of the elongating human RA-gastruloids. After RA supplementation at 0–24 h, these gastruloids were subjected to live imaging after the addition of 5% Matrigel and RA at 48 h from the induction. Arrowheads indicate the emergence of a segmentation. The time after embedding gastruloid into Matrigel and RA is shown on the top left. **b**, Number of somite pairs observed in RA-gastruloid in the live imaging. The x-axis indicates the hours from embedding of the RA-gastruloid into Matrigel. **c**, Boxplot of the time interval between successive segmentations.

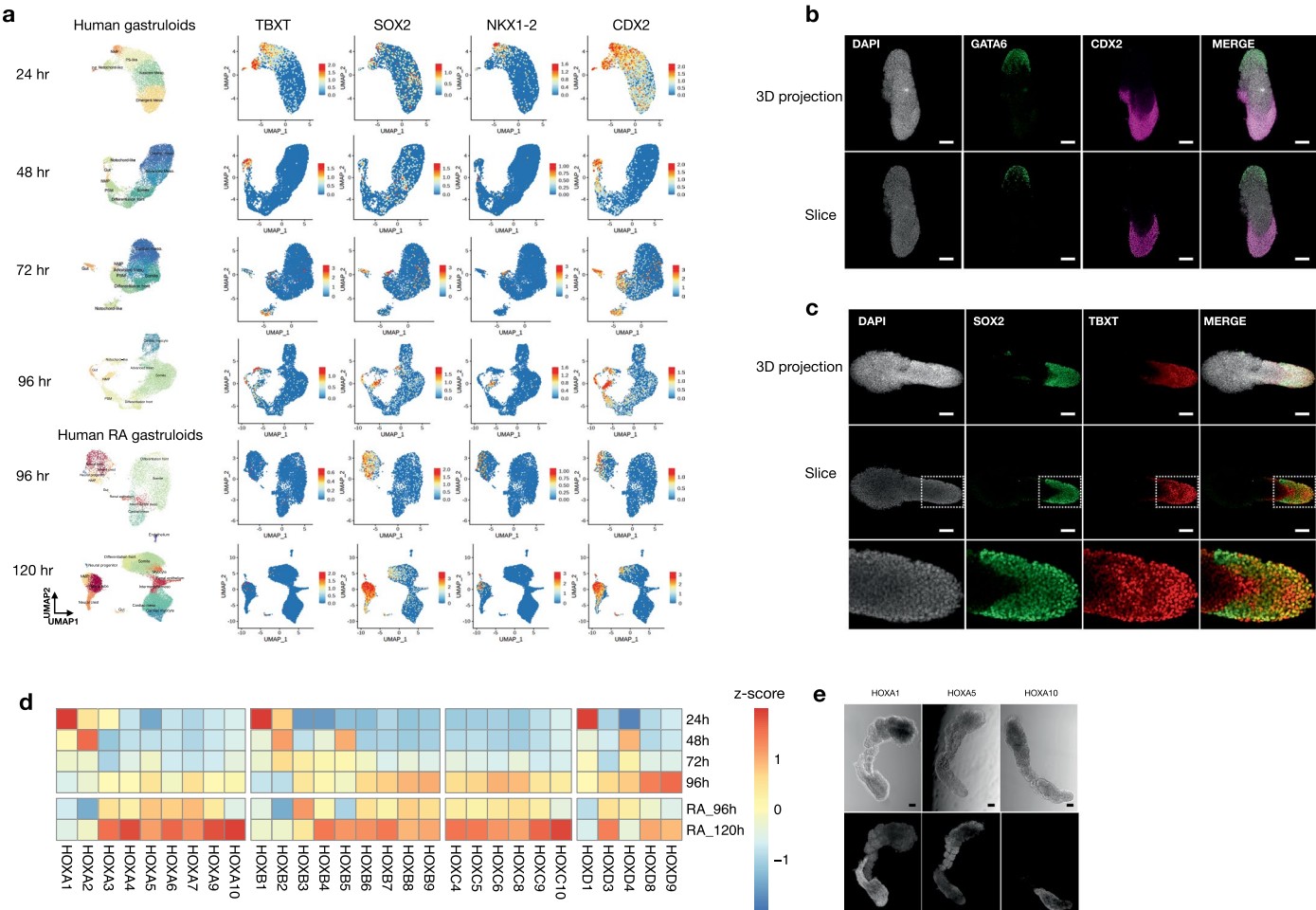

**Extended Data Fig. 7 | Evaluation of NMPs and anterior–posterior patterning in human RA-gastruloids. a**, UMAP visualization with cell types and normalized expression patterns of NMP marker genes reported in literature[37] for conventional human gastruloids at 24, 48, 72, or 96 h, or human RA-gastruloids at 96 or 120 h. **b**, Immunostaining of GATA6 (anterior marker) and CDX2 (posterior marker) in 48 h human RA-gastruloid. 3D max projection (Top) and sliced (Bottom) view. Scale bar, 100 μm. **c**, Immunostaining of SOX2 and TBXT in 48 h human RA-gastruloid. 3D max projection (Top) and sliced (Bottom) view. The dotted square in the middle row is zoomed in the bottom row. Scale bar, 100 μm. **d**, Heatmap showing the relative expression of HOX genes in conventional and RA-gastruloids at various timepoints. Mean expression levels of whole cells of each sample were normalized by z-score across samples. **e**, Representative images of HCR of HOX genes in human RA-gastruloids at 120 h. (Top) Bright-field and (Bottom) HCR imaging of *HOXA1*, *HOXA5* and *HOXA10*. Scale bar, 100 μm. N = 18, 17, 21 gastruloids for *HOXA1*, *HOXA5* and *HOXA10*, respectively.

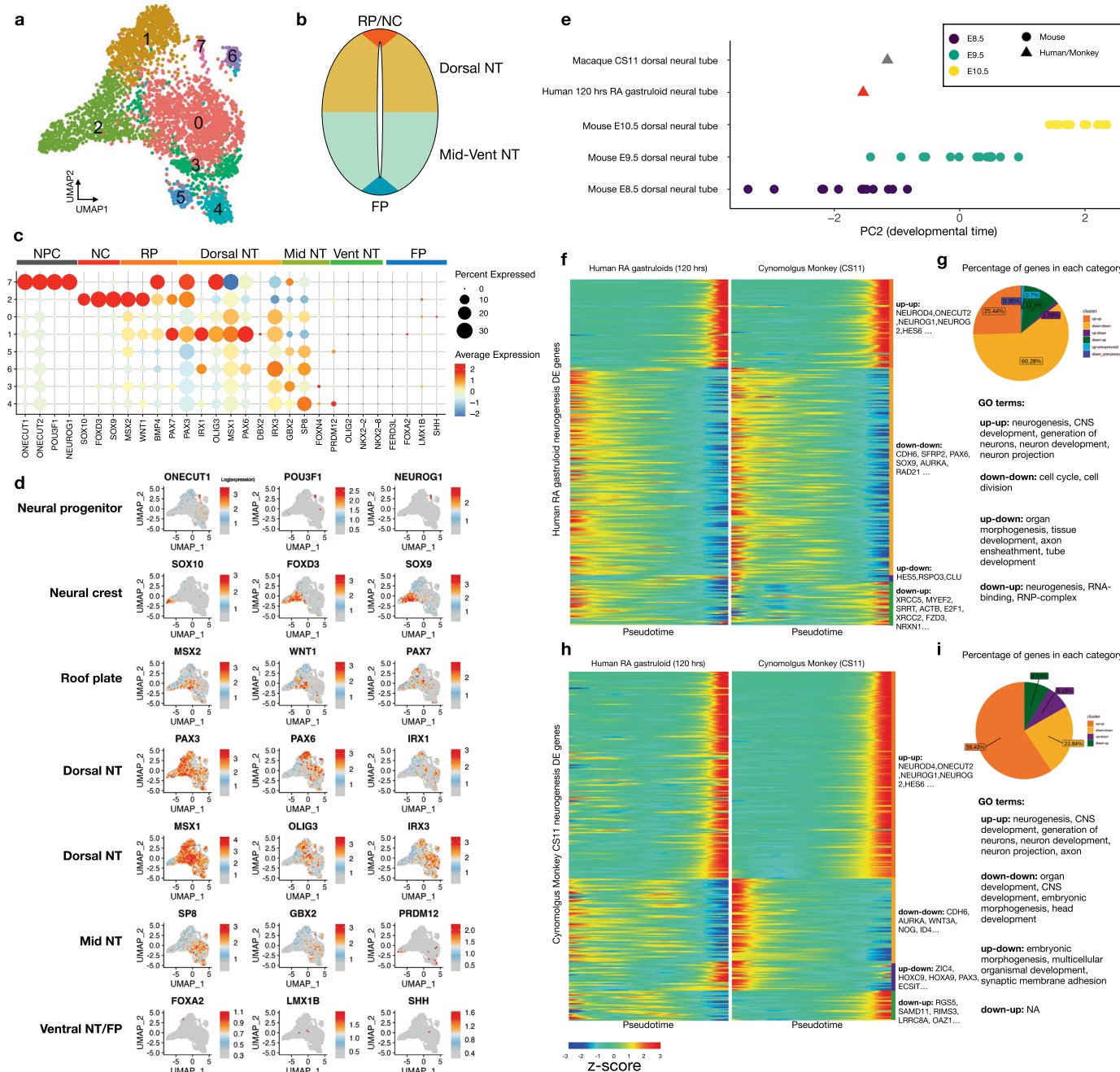

**Extended Data Fig. 8 | Evaluation of dorsal-ventral markers and neural differentiation in human RA-gastruloids. a**, Clustering and UMAP visualization of neural-related cells from scRNA-seq data of 120 hr human RA-gastruloids. **b**, Schematic of dorsal-ventral axis of neural tube, with labels corresponding to subsets of marker genes shown in panel (**c**). **c**, Bubble plot of marker gene expression patterns in each of the clusters shown in panel (**a**). **d**, Marker gene expression, projected onto UMAP shown in panel (**a**). NPC, neural progenitor; NC, neural crest cells; RP, roof plate; NT, neural tube; FP, floor plate. **e**, Staging alignment of pseudo-bulk profiles of neural tube cells from CS11 cynomolgus monkey embryos, neural tube cells from 120 h human RA-gastruloids and dorsal neural tube cells from E8.5–E10.5 mouse embryos, leveraging human-defined PC2 (see Fig. 4, Supplementary Fig. 12, Supplementary Fig. 13 and corresponding sections of main text for more details). **f**, Developmentally differentially expressed genes (DEGs) along the neural differentiation trajectory of human

RA-gastruloids were computed. The heatmaps show side-by-side comparison of the scaled expression level of these DEGs along neural differentiation trajectories in 120 h human RA-gastruloids (left) or CS11 cynomolgus monkey embryos (right). Genes are grouped into concordant (up-up: upregulated in both species, down-down: downregulated in both species) and discordant (up-down: upregulated in human and downregulated in monkey, down-up: downregulated in human and upregulated in monkey) categories with example genes shown for each category. Genes that are not detected in monkeys are not shown in the heatmaps. **g**, The percentage of DEGs in each category and the GO terms (http://www.gsea-msigdb.org/gsea) associated with each category shown in panel (**f**). **h**, Same as panel (**g**), except that the DEGs shown are from the neural differentiation trajectory of cynomolgus monkey. **i**, The percentage of DEGs in each category and the GO terms associated with each category shown in panel (**h**).

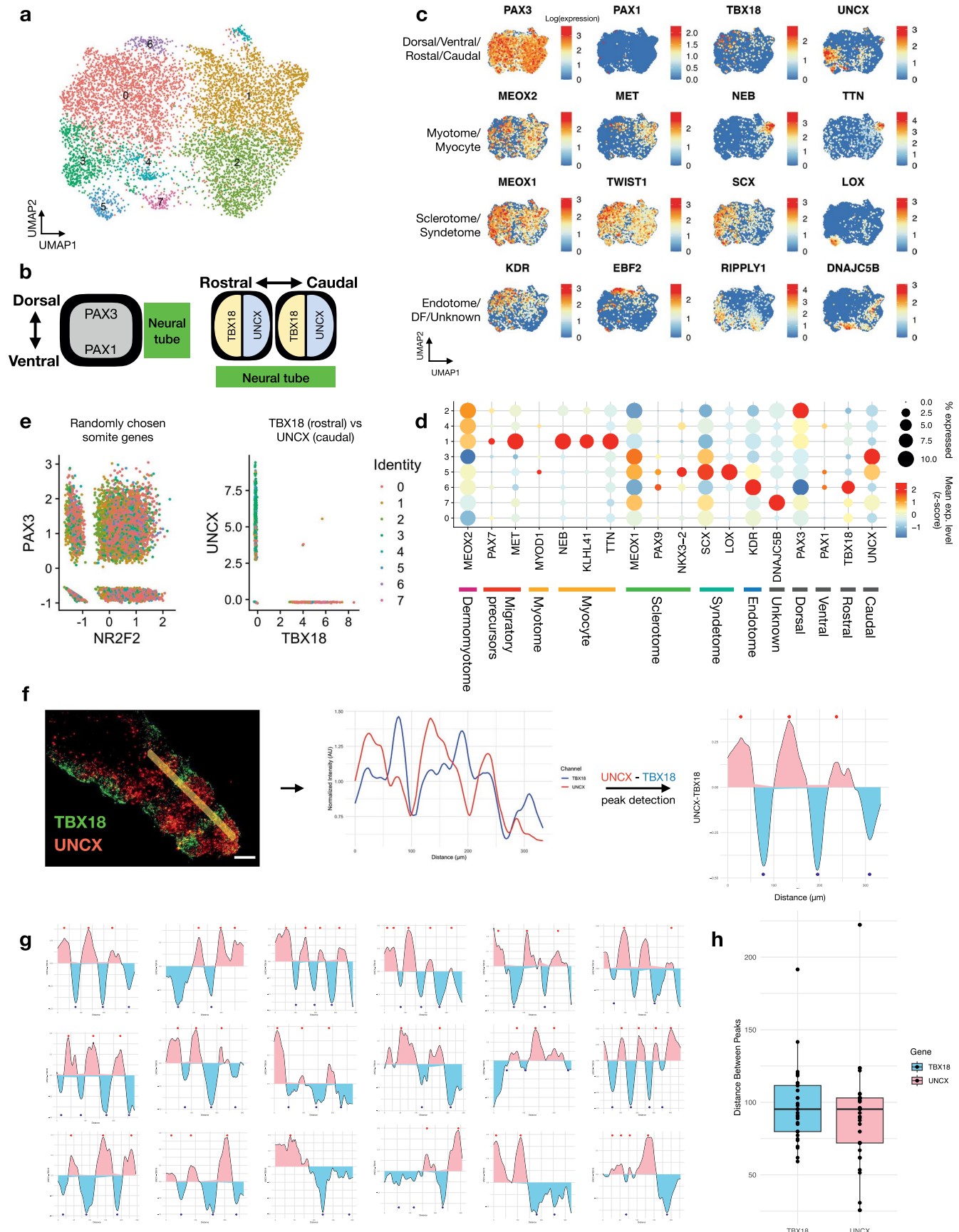

**Extended Data Fig. 9 | See next page for caption.**

**Extended Data Fig. 9 | Markers of spatial patterning and differentiation in the somites of human RA-gastruloids. a**, Clustering and UMAP visualization of somite-related cell types (somite, differentiation front, myocyte) from scRNA-seq data of 120 h human RA-gastruloids. **b**, Schematic of dorsal-ventral and rostral-caudal axes of somites, with expected marker genes noted. **c**, Same UMAP as panel (**a**) showing expression of selected marker genes for various subtypes of cells within somites. DF, differentiation front. **d**, Bubble plot showing expression patterns of selected marker genes for clusters shown in panel (**a**). **e**, Scatter-plot showing scaled expression levels of randomly chosen pairs of genes (left) or *TBX18* vs. *UNCX* (right). The expression of *TBX18* and *UNCX*, markers of the rostral-caudal axis of somites, are mutually exclusive. **f**, (left)

Representative image of HCR of *UNCX* and *TBX18* in 120 h human RA-gastruloids, and quantification of the signal intensity along with the A-P axis. Scale bar, 50 µm. (middle) Signal intensities of TBX18 (blue) and UNCX (red) were measured on the yellow line of the left panel. Data was normalized with the mean values of each signal and processed with LOESS smoothing. (right). Line plot showing the difference between TBX18 and UNCX values. Red and blue dots indicate the peaks for UNCX-high (red) and TBX18-high (blue), respectively, detected by a findpeaks function of pracma R package. **g**, Similar line plots for 18 RA-gastruloids. **h**, Boxplot showing the distribution of the distances between successive TBX18-high (blue) or UNCX-high (red) peaks.

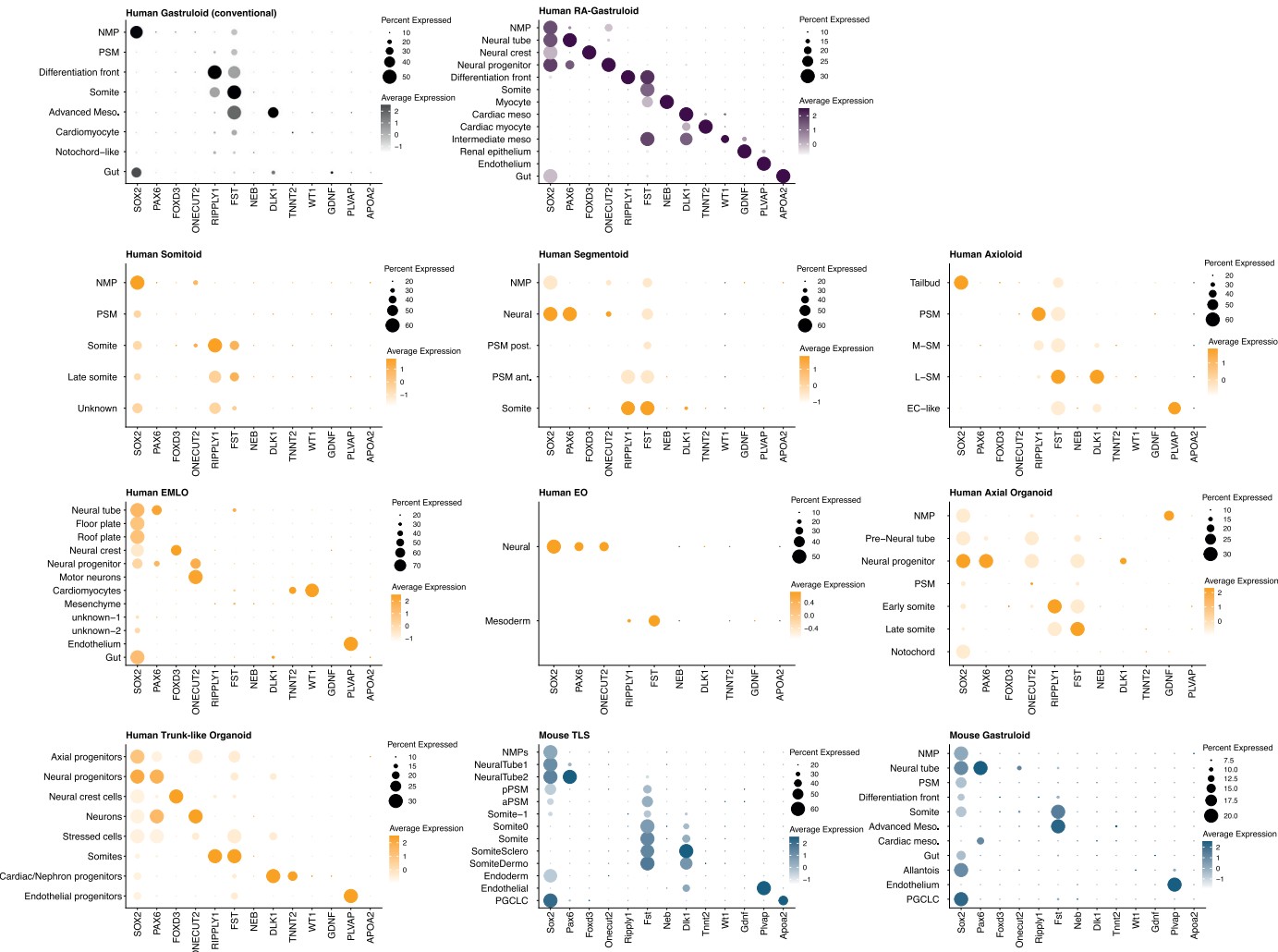

**Extended Data Fig. 10 | Cell types identified in various human and mouse embryo models.** Marker gene expression of annotated cell types in various embryo models are shown as bubble plots. The colours indicate the classification of embryo models; conventional human gastruloids (grey)[11], human RA-gastruloids (purple), other embryo models[16–20,22,45] (yellow), and mouse gastruloids (blue)[10,15]. The same set of marker genes (columns) are used in all panels.

Jay A Shendure

# Reporting Summary

## Statistics

For all statistical analyses, confirm that the following items are present in the figure legend, table legend, main text, or Methods section.

| n/a | Confirmed | |
|---|---|---|
| ☐ | ☒ | The exact sample size (*n*) for each experimental group/condition, given as a discrete number and unit of measurement |
| ☐ | ☒ | A statement on whether measurements were taken from distinct samples or whether the same sample was measured repeatedly |
| ☐ | ☒ | The statistical test(s) used AND whether they are one- or two-sided<br>*Only common tests should be described solely by name; describe more complex techniques in the Methods section.* |
| ☐ | ☐ | A description of all covariates tested |
| ☐ | ☒ | A description of any assumptions or corrections, such as tests of normality and adjustment for multiple comparisons |
| ☐ | ☒ | A full description of the statistical parameters including central tendency (e.g. means) or other basic estimates (e.g. regression coefficient) AND variation (e.g. standard deviation) or associated estimates of uncertainty (e.g. confidence intervals) |
| ☐ | ☒ | For null hypothesis testing, the test statistic (e.g. *F*, *t*, *r*) with confidence intervals, effect sizes, degrees of freedom and *P* value noted<br>*Give P values as exact values whenever suitable.* |
| ☒ | ☐ | For Bayesian analysis, information on the choice of priors and Markov chain Monte Carlo settings |
| ☒ | ☐ | For hierarchical and complex designs, identification of the appropriate level for tests and full reporting of outcomes |
| ☐ | ☒ | Estimates of effect sizes (e.g. Cohen's *d*, Pearson's *r*), indicating how they were calculated |

*Our web collection on statistics for biologists contains articles on many of the points above.*

## Software and code

Policy information about availability of computer code

| Data collection | Base calls were converted to fastq format using cellranger v6.0.0 mkfastq function. The sequencing reads are then demultiplexed based on i5 and i7 barcodes, mapped to hg38 reference genome and assigned to GRCh38 (GENCODE v32/Ensembl 98) genes by cellranger v6.0.0 count function with default setting. |
|---|---|
| Data analysis | Custom codes were available on https://github.com/shendurelab/Human-RA-Gastruloid.git.<br>The following common, freely available data analysis software packages were used in this project:<br>cellranger v6.0.0 (https://support.10xgenomics.com/single-cell-gene-expression/software/pipelines/6.1/release-notes)<br>Scrublet v0.2.3 (https://github.com/swolock/scrublet.git)<br>monocle 2.4.0 (https://github.com/cole-trapnell-lab/monocle-release)<br>data.table_1.15.4 (https://cran.r-project.org/web/packages/data.table/index.html)<br>harmony_1.2.0 (https://cran.r-project.org/web/packages/harmony/index.html)<br>Rcpp_1.0.12 (https://cran.r-project.org/web/packages/Rcpp/index.html)<br>reticulate_1.37.0 (https://cran.r-project.org/web/packages/reticulate/index.html)<br>lubridate_1.9.3 (https://cran.r-project.org/web/packages/lubridate/index.html)<br>forcats_1.0.0 (https://cran.r-project.org/web/packages/forcats/index.html)<br>stringr_1.5.1 (https://cran.r-project.org/web/packages/stringr/index.html)<br>purrr_1.0.2 (https://cran.r-project.org/web/packages/purrr/index.html)<br>readr_2.1.5 (https://cran.r-project.org/web/packages/readr/index.html)<br>tidyr_1.3.1 (https://cran.r-project.org/web/packages/tidyr/index.html)<br>tibble_3.2.1 (https://cran.r-project.org/web/packages/tibble/index.html)<br>tidyverse_2.0.0 (https://cran.r-project.org/web/packages/tidyverse/index.html) |

```
dplyr_1.1.4  (https://cran.r-project.org/web/packages/dplyr/index.html)
SeuratObject_4.1.4 (https://rdocumentation.org/packages/SeuratObject/versions/4.1.4)
Seurat_4.4.0 (https://github.com/satijalab/seurat/releases/tag/v4.4.0)
readr_2.1.5  (https://github.com/cran/readr/releases/tag/2.1.5)
stringr_1.5.1  (https://cran.r-project.org/web/packages/stringr/index.html)
dplyr_1.1.4  (https://github.com/cran/dplyr/releases/tag/1.1.4)
ggplot2_3.5.1 (https://cran.r-project.org/web/packages/ggplot2/index.html)
Matrix_1.6-5 (https://cran.r-project.org/src/contrib/Archive/Matrix/Matrix_1.6-5.tar.gz)
```

For manuscripts utilizing custom algorithms or software that are central to the research but not yet described in published literature, software must be made available to editors and reviewers. We strongly encourage code deposition in a community repository (e.g. GitHub). See the Nature Portfolio guidelines for submitting code & software for further information.

## Data

Policy information about availability of data

All manuscripts must include a data availability statement. This statement should provide the following information, where applicable:
- Accession codes, unique identifiers, or web links for publicly available datasets
- A description of any restrictions on data availability
- For clinical datasets or third party data, please ensure that the statement adheres to our policy

The data generated in this study can be downloaded in raw and processed forms from the National Center for Biotechnology Information (NCBI) Gene Expression Omnibus (GEO) under accession numbers GSE208369.

## Research involving human participants, their data, or biological material

Policy information about studies with human participants or human data. See also policy information about sex, gender (identity/presentation), and sexual orientation and race, ethnicity and racism.

| Reporting on sex and gender | N/A |
|---|---|
| Reporting on race, ethnicity, or other socially relevant groupings | N/A |
| Population characteristics | N/A |
| Recruitment | N/A |
| Ethics oversight | N/A |

Note that full information on the approval of the study protocol must also be provided in the manuscript.

# Field-specific reporting

Please select the one below that is the best fit for your research. If you are not sure, read the appropriate sections before making your selection.

☒ Life sciences          ☐ Behavioural & social sciences          ☐ Ecological, evolutionary & environmental sciences

For a reference copy of the document with all sections, see nature.com/documents/nr-reporting-summary-flat.pdf

# Life sciences study design

All studies must disclose on these points even when the disclosure is negative.

| Sample size | No statistical methods were used to pre-determine sample sizes but our sample sizes are similar to those reported in previous publications(Veenvliet et al. 2020; Moris et al. 2020, see the references in the main text) |
|---|---|
| Data exclusions | Sequencing data exclusion criteria is outlined in th method section, including filtering out the substandard data in single-cell measurements, following the general practice in the field. |
| Replication | Single-cell RNA-seq experiments were done in a single replicate experiment, or two biological replicates for knockout gastruloid Single-cell RNA-seq experiments. All the other experiments were repeated in at least three times. |
| Randomization | Human conventional gastruloids and RA-gastruloids used in experiments were randomly selected from each timepoint before sample preparation. |
| Blinding | The Investigators were not blinded to allocation during experiments and outcome assessment. |

# Reporting for specific materials, systems and methods

We require information from authors about some types of materials, experimental systems and methods used in many studies. Here, indicate whether each material, system or method listed is relevant to your study. If you are not sure if a list item applies to your research, read the appropriate section before selecting a response.

## Materials & experimental systems

| n/a | Involved in the study |
|---|---|
| ☐ | ☒ Antibodies |
| ☐ | ☒ Eukaryotic cell lines |
| ☒ | ☐ Palaeontology and archaeology |
| ☒ | ☐ Animals and other organisms |
| ☒ | ☐ Clinical data |
| ☒ | ☐ Dual use research of concern |
| ☒ | ☐ Plants |

## Methods

| n/a | Involved in the study |
|---|---|
| ☒ | ☐ ChIP-seq |
| ☒ | ☐ Flow cytometry |
| ☒ | ☐ MRI-based neuroimaging |

## Antibodies

**Antibodies used**

All antibodies are commercially available.
Primary antibody
Goat anti-SOX1 R&D AF3369-SP
Goat anti-SOX2 R&D AF2018
Rabbit anti-SOX2 Millipore AB5603
Rabbit anti-PAX3 ThermoFisher 38-1801
Rabbit anti-PAX3 Thermo 701147
Rabbit anti-PAX8 Proteintech 10336-1-AP
Rabbit anti-WT1 Abcam ab89901
Rabbit Anti-POU3F1 Abcam ab272925
Rabbit Anti-TBX6 Abcam ab38883
Goat Anti-GATA6 R&D AF1700
Rabbit Anti-CDX2 Thermo MA5-14494
Rabbit Anti-TBXT abcam ab209665
Rabbit Anti-SOX10 abcam ab227680

Secondary antibody
Donkey anti-Goat IgG (H+L) Highly Cross-Adsorbed Secondary Antibody, Alexa Fluor Plus 488 ThermoFisher A32814
Donkey anti-Rabbit IgG (H+L) Highly Cross-Adsorbed Secondary Antibody, Alexa Fluor Plus 555 ThermoFisher A32794

**Validation**

Goat anti-SOX1 R&D AF3369-SP
Validation was performed by the manufacturer using iBJ6 human iPS cells and iBJ6 human iPS cells differentiated into neuroprogenitor cells (https://www.rndsystems.com/products/human-mouse-rat-sox1-antibody_af3369).

Goat anti-SOX2 R&D AF2018
Validation was performed by the manufacturer using D3 mouse embryonic stem cell line, NTera-2 human testicular embryonic carcinoma cell line, F9 mouse teratocarcinoma stem cells, and rat cortical stem cells (https://www.rndsystems.com/products/human-mouse-rat-sox2-antibody_af2018).

Rabbit anti-SOX2 Millipore AB5603
Validation was performed by the manufacturer using Mouse or human embryonic stem cells, and mouse embryonic germ cells (https://www.emdmillipore.com/US/en/product/Anti-Sox2-Antibody,MM_NF-AB5603?ReferrerURL=https%3A%2F%2Fwww.google.com%2F).

Rabbit anti-PAX3 ThermoFisher 38-1801
Validation was performed by the manufacturer using HEK-293, B16-F10, and Neuro-2a cells (https://thermofisher.com/antibody/product/PAX3-Antibody-Polyclonal/38-1801).

Rabbit anti-PAX3 Thermo 701147
Validation was performed by the manufacturer using U2OS, A431, A375, HEK-293, THP1, Mouse Testis and Mouse Cerebellum (https://www.thermofisher.com/antibody/product/PAX3-Antibody-clone-16H22L10-Recombinant-Monoclonal/701147)

Rabbit anti-PAX8 Proteintech 10336-1-AP
Validation was performed by the manufacturer using xxx cells (link).

Rabbit anti-WT1 Abcam ab89901
Validation was performed by the manufacturer using HEK-293T, and K-562 cell lines (https://www.abcam.com/wt1-antibody-ab89901.html).
Rabbit Anti-POU3F1 Abcam ab272925
Validation was performed by the manufacturer using A549 (Human lung carcinoma cells) and mouse fetal brain tissue extracts

(https://www.abcam.com/products/primary-antibodies/oct6-antibody-ab272925.html).

Rabbit Anti-TBX6 Abcam ab38883
Validation was performed by the manufacturer using mouse testis tissue lysate, mouse lung tissue lysate, human testis tissue lysate, and human lung tissue lysate (https://www.abcam.com/products/primary-antibodies/tbx6-antibody-ab38883.html).

Goat Anti-GATA6 R&D AF1700
Validation was performed by the manufacturer using KATO-III human gastric carcinoma cell line and PC-3 human prostate cancer cell line (https://www.rndsystems.com/products/human-gata-6-antibody_af1700).

Rabbit Anti-CDX2 Thermo MA5-14494
Validation was performed by the manufacturer using CaCO2, SW480, HCT116, HeLa and MCF7 (https://www.thermofisher.com/antibody/product/CDX2-Antibody-clone-EPR2764Y-Monoclonal/MA5-14494).

Rabbit Anti-TBXT abcam ab209665
Validation was performed by the manufacturer using Human chordoma tissue, Mouse E14.5 embryo tissue, and Rat E14.5 embryo Tissue cells (https://www.abcam.com/products/primary-antibodies/brachyury--bry-antibody-epr18113-ab209665.html).

Rabbit Anti-SOX10 abcam ab227680
Validation was performed by the manufacturer using Human melanoma tissue; mouse and rat breast tissues (https://www.abcam.com/products/primary-antibodies/sox10-antibody-sp267-ab227680.html).

Donkey anti-Goat IgG (H+L) Highly Cross-Adsorbed Secondary Antibody, Alexa Fluor Plus 488 ThermoFisher A32814
Validation was performed by the manufacturer using LNCaP (positive model) and HeLa (negative model) cells (https://www.thermofisher.com/antibody/product/Donkey-anti-Goat-IgG-H-L-Highly-Cross-Adsorbed-Secondary-Antibody-Polyclonal/A32814).

Donkey anti-Rabbit IgG (H+L) Highly Cross-Adsorbed Secondary Antibody, Alexa Fluor Plus 555 ThermoFisher A32794
Validation was performed by the manufacturer using A549 cells (https://www.thermofisher.com/antibody/product/Donkey-anti-Rabbit-IgG-H-L-Highly-Cross-Adsorbed-Secondary-Antibody-Polyclonal/A32794).

## Eukaryotic cell lines

Policy information about cell lines and Sex and Gender in Research

| | |
|---|---|
| Cell line source(s) | RUES2-GLR were provided by Dr. Ali H. Brivanlou (The Rockefeller University). H9 hESCs were obtained from WiCell. WTC11 hiPSCs were gifted by Dr. Bruce Conklin (Gladstone Institutes). |
| Authentication | Activities of three knocked-in reporter genes (SOX2-mCit, TBXT-mCer, and SOX17-tdTom) were validated in every experiment with the fluorescence microscope observation. The RUES2-GLR line was authenticated with scRNA-seq. |
| Mycoplasma contamination | This cell lines are not ested for Mycoplasma contamination. |
| Commonly misidentified lines (See ICLAC register) | No commonly misidentified cell lines were used. |

## Plants

| | |
|---|---|
| Seed stocks | *Report on the source of all seed stocks or other plant material used. If applicable, state the seed stock centre and catalogue number. If plant specimens were collected from the field, describe the collection location, date and sampling procedures.* |
| Novel plant genotypes | *Describe the methods by which all novel plant genotypes were produced. This includes those generated by transgenic approaches, gene editing, chemical/radiation-based mutagenesis and hybridization. For transgenic lines, describe the transformation method, the number of independent lines analyzed and the generation upon which experiments were performed. For gene-edited lines, describe the editor used, the endogenous sequence targeted for editing, the targeting guide RNA sequence (if applicable) and how the editor was applied.* |
| Authentication | *Describe any authentication procedures for each seed stock used or novel genotype generated. Describe any experiments used to assess the effect of a mutation and, where applicable, how potential secondary effects (e.g. second site T-DNA insertions, mosiacism, off-target gene editing) were examined.* |

