## [Peer Review File · Nature Cell Biology]

Peer Review Information

Journal: Nature Cell Biology

Manuscript Title: Retinoic acid induces human gastruloids with posterior embryo-like structures

Corresponding author name(s): Professor Jay Shendure

Editorial Notes:

Reviewer Comments & Decisions:

Decision Letter, initial version:
--

*Please delete the link to your author homepage if you wish to forward this email to co-authors.

Dear Jay,

Once again, please accept our sincere apologies for the delay here. Your manuscript, "Induction and *in silico* staging of human gastruloids with neural tube, segmented somites & advanced cell types", has now been seen by all the original referees. As previously communicated, reviewer #2 continued to raise persisting concerns, which is why we asked for your response to them via e-mail. We subsequently asked from reviewer #1 to provide their feedback into some of the persisting concerns raised by reviewer #2 and your response to them (you can find this in the comments by reviewer #1 in the section 'ADDITIONAL COMMENTS/VIEW ON REMAINING ISSUES BY REVIEWER #2'). To explain, the possibility of seeking extra feedback is part of our standard editorial process of potentially discussing comments with referees, in our efforts to provide as balanced and fair a peer-review process as possible. As you will see from all the referee comments (attached below) there are still some points that need to be addressed. Although we are also very interested in this study, we believe that these concerns should be addressed before we can consider publication in Nature Cell Biology.

Nature Cell Biology editors discuss the referee reports in detail within the editorial team, including the chief editor, to identify key referee points that should be addressed with priority, and requests that are overruled as being beyond the scope of the current study. To guide the scope of the revisions, I have listed these points below. We are committed to providing a fair and constructive peer-review process, so please feel free to contact me if you would like to discuss any of the referee comments further.

In particular, while revising your manuscript, please follow the guidelines below:

(A) You should address the 3 remaining points by referee #1:

Referee #1:

"1) In the part where the authors discuss their interpretation that having a sufficient number of NMPs, PSMs could be a prerequisite for the subsequent formation of somites upon MG addition, I would recommend to refrain from using the terms "stem cells" and "stem cell pool" and instead use "axial progenitor cells" or "axial progenitor pool" to make it more clear that they do not refer to pluripotent stem cells.

2) Could the authors please add summary statistics for the RC somite patterning measured in all 18 RA-gastruloids (Fig S20D)? Given that the authors state that 14/18 RA-gastruloids displayed similar (RC) patterning, there really is no reason to not provide the summary statistics instead of just 3 examples with individual quantifications.

3) It would be helpful if the authors could add a short note to the Methods where they identify the criteria for "similar morphology" for all occasions where they refer to reproducibility by this terminology."

(B) All sequencing data should be deposited and become publicly accessible, but we leave it to your discretion to decide whether you wish to also provide the data in the processed form (i.e., including cell type annotation, etc), like referee #3 suggests.

(C) In terms of the remaining comments of referee #2, which referee #1 has commented on, we require that you please address the following point by referee #1: "Point #2: In my opinion, the neural tube staining issue is sufficiently addressed by the SOX1 stainings in the current Fig 2 & 3. One technical point of concern regarding the immunostainings is the staining in Figure 3e. Why is there such a DAPI signal gradient? Especially since the WT1 staining partially follows this gradient. The authors could co-stain with eg FOXC1 (pan-somite) and WT1 to resolve".

(D) We leave it to your discretion to decide what to do regarding point #7 by referee #2 about the work by Faustino Martins et al.

(E) We require that you address point #3 by referee #2, the way you have proposed to address it in the unofficial response to reviewer #2 you sent us via e-mail.

(F) As previously communicated, we do not expect you to address point #8 by referee #2, as we find this would be insightful, but beyond the scope of the study at this point.

(G) Finally please pay close attention to our guidelines on statistical and methodological reporting (listed below) as failure to do so may delay the reconsideration of the revised manuscript. In particular please provide:

We therefore invite you to take these points into account when revising the manuscript. In addition, when preparing the revision please:

- ensure that it conforms to our format instructions and publication policies (see below and <https://www.nature.com/nature/for-authors>).
- provide a point-by-point rebuttal to the full referee reports verbatim, as provided at the end of this letter.
- provide the completed Reporting Summary (found here <https://www.nature.com/documents/nr-reporting-summary.pdf>). This is essential for reconsideration of the manuscript and will be available to editors and referees in the event of peer review. For more information see <http://www.nature.com/authors/policies/availability.html> or contact me.

When submitting the revised version of your manuscript, please pay close attention to our [href="https://www.nature.com/nature-portfolio/editorial-policies/image-integrity">Digital Image Integrity Guidelines](https://www.nature.com/nature-portfolio/editorial-policies/image-integrity). and to the following points below:

Nature Cell Biology is committed to improving transparency in authorship. As part of our efforts in this direction, we are now requesting that all authors identified as 'corresponding author' on published papers create and link their Open Researcher and Contributor Identifier (ORCID) with their account on the Manuscript Tracking System (MTS), prior to acceptance. ORCID helps the scientific community achieve unambiguous attribution of all scholarly contributions. You can create and link your ORCID from the home page of the MTS by clicking on 'Modify my Springer Nature account'. For more information please visit www.springernature.com/orcid.

This journal strongly supports public availability of data. Please place the data used in your paper into a public data repository, or alternatively, present the data as Supplementary Information. If data can only be shared on request, please explain why in your Data Availability Statement, and also in the correspondence with your editor. Please note that for some data types, deposition in a public repository is mandatory - more information on our data deposition policies and available repositories appears below.

[Redacted]

We would like to receive the revision within four weeks. If submitted within this time period, reconsideration of the revised manuscript will not be affected by related studies published elsewhere, or accepted for publication in Nature Cell Biology in the meantime. We would be happy to consider a revision even after this timeframe, but in that case we will consider the published literature at the time of resubmission when assessing the file.

We hope that you will find our referees' comments, and editorial guidance helpful. Please do not hesitate to contact me if there is anything you would like to discuss.

Best wishes,

Stelios

Stylianos Lefkopoulos, PhD
He/him/his
Senior Editor, Nature Cell Biology
Springer Nature
Heidelberger Platz 3, 14197 Berlin, Germany

E-mail: stylianos.lefkopoulos@springernature.com
Twitter: @s_lefkopoulos
LinkedIn: [linkedin.com/in/stylianos-lefkopoulos-81b007a0](https://www.linkedin.com/in/stylianos-lefkopoulos-81b007a0)

Reviewers' Comments:

Reviewer #1:

Remarks to the Author:

The authors should be commended for doing a stellar job in re-revising their manuscript, and I'd happily recommend publication in NCB. Just three final comments:

1) In the part where the authors discuss their interpretation that having a sufficient number of NMPs, PSMs could be a prerequisite for the subsequent formation of somites upon MG addition, I would recommend to refrain from using the terms "stem cells" and "stem cell pool" and instead use "axial progenitor cells" or "axial progenitor pool" to make it more clear that they do not refer to pluripotent stem cells.

2) Could the authors please add summary statistics for the RC somite patterning measured in all 18 RA-gastruloids (Fig S20D)? Given that the authors state that 14/18 RA-gastruloids displayed similar (RC) patterning, there really is no reason to not provide the summary statistics instead of just 3 examples with individual quantifications.

3) It would be helpful if the authors could add a short note to the Methods where they identify the criteria for "similar morphology" for all occasions where they refer to reproducibility by this terminology.

Also, the authors asked for some advice how to distinguish the ectopic NTs from the central NT in future work. My suggestions:

i) the genome engineering strategy employed for mouse Tbx6-KO TLS in Veenvliet et al., 2020. This allows for imaging-based distinction between ectopic and central NTs.

ii) Sampath Kumar et al., Nature Gen 2023 (<https://www.nature.com/articles/s41588-023-01435-6>) performed spatial transcriptomics on WT vs Tbx6-KO embryos & identified distinct molecular profiles for ectopic and central NTs (Fig 5). This could be leveraged to distinguish between both in RA-gastruloids.

Finally, the application of DNA Typewriter technology to RA-gastruloids sounds very exciting. I am truly looking forward to the future discoveries this will bring!

ADDITIONAL COMMENTS/VIEW ON REMAINING ISSUES BY REVIEWER #2

I was asked to comment on whether the additional response by the authors provided via e-mail addressed points #2 and #3 by reviewer #2.

Point #2: In my opinion, the neural tube staining issue is sufficiently addressed by the SOX1 stainings in the current Fig 2 & 3. One technical point of concern regarding the immunostainings is the staining in Figure 3e. Why is there such a DAPI signal gradient? Especially since the WT1 staining partially follows this gradient. The authors could co-stain with eg FOXC1 (pan-somite) and WT1 to resolve.

Point #3: With their additional response, the authors have now addressed this point, in my opinion.

Finally (regarding point #7), in my view, an embryo model (stembryo) should mimic aspects of *both* embryo cell type *and* patterning and/or architecture. This is *not* the case in the Faustino Martino et al. neuromuscular organoids, which only recapitulate minor aspects of embryo morphogenesis in a very limited temporal window (elongation at day 5). In terms of recapitulating embryonic features (cellular composition concomitant with correct morphogenesis and patterning, it is therefore, in my opinion, a fair claim that "RA-gastruloids progress further than most alternative stembryo models" (side-note: stembryo models is not a correct term: it should be stembryos *or* embryo models; a stembryo model would be a model of a stembryo).

Reviewer #2:

Remarks to the Author:

The study introduces another human gastruloid model (RA-gastruloids) that develop trunk like structures including cell types found in the neural tube and somites. Alongside this a computational

staging method is presented to systematically compare developmental progression across models. The study primarily contains some useful but minor modifications to previously published human gastruloid protocols. I remain of the opinion that the study is better suited to a methods focused journal.

Concerning the specific issues raised in the last round of reviews.

1. The authors have corrected the assertion concerning an active segmentation clock.
2. I remain of the opinion that some of the immunostaining is of poor quality. It is common to have antibody penetration problems resulting in uneven immunofluorescence and false negatives in organoids containing epithelia structures. This to be the case in the current study in some of the images (e.g. Fig 2F).
3. The authors have partially clarified the conclusion that there is dorsal-ventral pattern in the neural tube. The problem is with the word "pattern" as it suggests a consistent and reproducible spatial arrangement of different cell types. The data do not seem to be sufficient to conclude this.
4. The authors now cite Olmstead and Paluh (2021).
5. The authors have corrected their use of "neurogenesis".
6. The authors now comment on the presence of syndetome (SCX+), sclerotome (PAX9+, NKX3-2+) cells. (Note syndetome arises ventrally between sclerotome and dermomyotome.)
7. Unfortunately, the authors continue to overlook the work of Faustino Martins et al, which describes neuromuscular organoids. The authors state that they do not cite this work as neuromuscular organoids are an "an organoid model rather than a gastruloid/stembryo model". In my view, this is a distinction without a difference. NMOs are generated using a protocol very similar to the methods that the authors describe and they contain similar tissue derivatives. Ignoring the work of Faustino Martins et al seems to be an attempt to claim false novelty for their study. Similarly, although Olmstead and Paluh might not have provided single cell transcriptome data for later time points in their study, they offer evidence of advanced differentiation of organoids in their paper. I remain of the opinion that the statement "RA-gastruloids progress further than most alternative mammalian stembryo models" is over reaching.
8. It remains the case that generating chimeric WT - Tbx6-KO gastruloids would be an opportunity of demonstrating the usefulness of gastruloid models to address biological questions.

Reviewer #3:

Remarks to the Author:

The authors have adequately addressed the remaining points I raised.

Moreover, in my opinion, the level of novelty of the embryonic organoid model they describe and the depth of description and analysis of the system across several conditions make this work suitable for publication in NCB.

My final recommendation is to make all the sequencing data they produced fully accessible also in the processed form (i.e., including cell type annotation, etc) to ensure reproducibility and facilitate the use of these beautiful and important datasets by the scientific community.

GUIDELINES FOR SUBMISSION OF NATURE CELL BIOLOGY ARTICLES

ARTICLE FORMAT

ABSTRACT – should not exceed 150 words and should be unreferenced. This paragraph is the most visible part of the paper and should briefly outline the background and rationale for the work, and accurately summarize the main results and conclusions. Key genes, proteins and organisms should be specified to ensure discoverability of the paper in online searches.

TEXT – the main text consists of the Introduction, Results, and Discussion sections and must not exceed 3500 words including the abstract. The Introduction should expand on the background relating to the work. The Results should be divided in subsections with subheadings, and should provide a concise and accurate description of the experimental findings. The Discussion should expand on the findings and their implications. All relevant primary literature should be cited, in particular when discussing the background and specific findings.

FINANCIAL AND NON-FINANCIAL COMPETING INTERESTS – the authors must include one of three declarations: (1) that they have no financial and non-financial competing interests; (2) that they have financial and non-financial competing interests; or (3) that they decline to respond, after the Author

Contributions section. This statement will be published with the article, and in cases where financial and non-financial competing interests are declared, these will be itemized in a web supplement to the article. For further details please see <https://www.nature.com/licenceforms/nrg/competing-interests.pdf>.

REFERENCES – are limited to a total of 70 in the main text and Methods combined,. They must be numbered sequentially as they appear in the main text, tables and figure legends and Methods and must follow the precise style of Nature Cell Biology references. References only cited in the Methods should be numbered consecutively following the last reference cited in the main text. References only associated with Supplementary Information (e.g. in supplementary legends) do not count toward the total reference limit and do not need to be cited in numerical continuity with references in the main text. Only published papers can be cited, and each publication cited should be included in the numbered reference list, which should include the manuscript titles. Footnotes are not permitted.

Methods should be written concisely, but should contain all elements necessary to allow interpretation and replication of the results. As a guideline, Methods sections typically do not exceed 3,000 words. The Methods should be divided into subsections listing reagents and techniques. When citing previous methods, accurate references should be provided and any alterations should be noted. Information must be provided about: antibody dilutions, company names, catalogue numbers and clone numbers for monoclonal antibodies; sequences of RNAi and cDNA probes/primers or company names and catalogue numbers if reagents are commercial; cell line names, sources and information on cell line identity and authentication. Animal studies and experiments involving human subjects must be reported in detail, identifying the committees approving the protocols. For studies involving human subjects/samples, a statement must be included confirming that informed consent was obtained. Statistical analyses and information on the reproducibility of experimental results should be provided in a section titled "Statistics and Reproducibility".

All Nature Cell Biology manuscripts submitted on or after March 21 2016, must include a Data availability statement as a separate section after Methods but before references, under the heading "Data Availability". For Springer Nature policies on data availability see <http://www.nature.com/authors/policies/availability.html>; for more information on this particular policy see <http://www.nature.com/authors/policies/data/data-availability-statements-data-citations.pdf>. The Data availability statement should include:

- Accession codes for primary datasets (generated during the study under consideration and designated as "primary accessions") and secondary datasets (published datasets reanalysed during the study under consideration, designated as "referenced accessions"). For primary accessions data should be made public to coincide with publication of the manuscript. A list of data types for which submission to community-endorsed public repositories is mandated (including sequence, structure, microarray, deep sequencing data) can be found here <http://www.nature.com/authors/policies/availability.html#data>.
- Unique identifiers (accession codes, DOIs or other unique persistent identifier) and hyperlinks for datasets deposited in an approved repository, but for which data deposition is not mandated (see here

for details <http://www.nature.com/sdata/data-policies/repositories>).

- At a minimum, please include a statement confirming that all relevant data are available from the authors, and/or are included with the manuscript (e.g. as source data or supplementary information), listing which data are included (e.g. by figure panels and data types) and mentioning any restrictions on availability.
- If a dataset has a Digital Object Identifier (DOI) as its unique identifier, we strongly encourage including this in the Reference list and citing the dataset in the Methods.

We recommend that you upload the step-by-step protocols used in this manuscript to the Protocol Exchange. More details can found at www.nature.com/protocolexchange/about.

DISPLAY ITEMS – main display items are limited to 6-8 main figures and/or main tables. For Supplementary Information see below.

FIGURES – Colour figure publication costs \$395 per colour figure. All panels of a multi-panel figure must be logically connected and arranged as they would appear in the final version. Unnecessary figures and figure panels should be avoided (e.g. data presented in small tables could be stated briefly in the text instead).

All imaging data should be accompanied by scale bars, which should be defined in the legend. Cropped images of gels/blots are acceptable, but need to be accompanied by size markers, and to retain visible background signal within the linear range (i.e. should not be saturated). The boundaries of panels with low background have to be demarked with black lines. Splicing of panels should only be considered if unavoidable, and must be clearly marked on the figure, and noted in the legend with a statement on whether the samples were obtained and processed simultaneously. Quantitative comparisons between samples on different gels/blots are discouraged; if this is unavoidable, it has to be performed for samples derived from the same experiment with gels/blots were processed in parallel, which needs to be stated in the legend.

- We accept PowerPoint (.PPT) files if they are fully editable. However, please refrain from adding PowerPoint graphical effects to objects, as this results in them outputting poor quality raster art. Text used for PowerPoint figures should be Helvetica (preferred) or Arial.
- We do not recommend using Adobe Photoshop for designing figures, but we can accept Photoshop generated (.PSD or .TIFF) files only if each element included in the figure (text, labels, pictures, graphs, arrows and scale bars) are on separate layers. All text should be editable in 'type layers' and line-art such as graphs and other simple schematics should be preserved and embedded within 'vector smart objects' - not flattened raster/bitmap graphics.
- Some programs can generate Postscript by 'printing to file' (found in the Print dialogue). If using an application not listed above, save the file in PostScript format or email our Art Editor, Allen Beattie for advice (a.beattie@nature.com).

Regardless of format, all figures must be vector graphic compatible files, not supplied in a flattened raster/bitmap graphics format, but should be fully editable, allowing us to highlight/copy/paste all text and move individual parts of the figures (i.e. arrows, lines, x and y axes, graphs, tick marks, scale bars etc). The only parts of the figure that should be in pixel raster/bitmap format are photographic images or 3D rendered graphics/complex technical illustrations.

Unprocessed scans of all key data generated through electrophoretic separation techniques need to be presented in a supplementary figure that should be labeled and numbered as the final supplementary figure, and should be mentioned in every relevant figure legend. This figure does not count towards the total number of figures and is the only figure that can be displayed over multiple pages, but should be provided as a single file, in PDF or TIFF format. Data in this figure can be displayed in a relatively informal style, but size markers and the figures panels corresponding to the presented data must be indicated.

The total number of Supplementary Figures (not including the “unprocessed scans” Supplementary Figure) should not exceed the number of main display items (figures and/or tables (see our Guide to Authors and March 2012 editorial <http://www.nature.com/ncb/authors/submit/index.html#suppinfo>; <http://www.nature.com/ncb/journal/v14/n3/index.html#ed>). No restrictions apply to Supplementary Tables or Videos, but we advise authors to be selective in including supplemental data.

GUIDELINES FOR EXPERIMENTAL AND STATISTICAL REPORTING

REPORTING REQUIREMENTS – We ask authors to complete a Reporting Summary that collects information on experimental design and reagents. We hope this will aid in your evaluation of the paper. The Reporting Summary can be found here <https://www.nature.com/documents/nr-reporting-summary.pdf>) Please note that these forms are dynamic ‘smart pdfs’ and must therefore be downloaded and completed in Adobe Reader. We will then flatten them for ease of use. If you would like to reference the guidance text as you complete the template, please access these flattened versions at <http://www.nature.com/authors/policies/availability.html>.

Author Rebuttal to Initial comments

Response to Reviewers

We thank the three reviewers for their constructive feedback on the revised manuscript. In this point-by-point response that accompanies our further revision, the new reviewer comments are replicated in full in **blue text**, while our responses to each point are in **black text**.

Reviewer #1:

Remarks to the Author:

The authors should be commended for doing a stellar job in re-revising their manuscript, and I'd happily recommend publication in NCB.

Thank you for these kind words and for your recommendation for publication in NCB.

Just three final comments:

1) In the part where the authors discuss their interpretation that having a sufficient number of NMPs, PSMs could be a prerequisite for the subsequent formation of somites upon MG addition, I would recommend to refrain from using the terms "stem cells" and "stem cell pool" and instead use "axial progenitor cells" or "axial progenitor pool" to make it more clear that they do not refer to pluripotent stem cells.

We agree. In the newest revision of the manuscript, we have replaced "stem cells" and "stem cell pool" with "axial progenitor cells" and "axial progenitor pool", respectively. The referenced section now reads as follows:

"Why does a conventional human gastruloid, which possesses both NMPs and presomitic mesoderm (**Fig. 1**), fail to form segmented somites even in Matrigel, while Matrigel alone is sufficient to support somitogenesis in mouse gastruloids^{2,16}? We speculate that this may be because there is a minimum threshold on the size of the **axial progenitor pool** ~~stem cell pool~~ at the onset of somitogenesis induced by mechanical/chemical stimulation by Matrigel. To assess this possibility further, we examined the cell type composition of 48-72 hrs conventional human gastruloids vs. that of axiolooids, somitoids, and segmentoids, based on sc-RNA-seq data¹⁷⁻¹⁹ (**Fig. S32a-c**). Interestingly, at 48 hrs, the axiolooid, somitoid, and segmentoid models were entirely composed of PSMs (91, 92, and 82%, respectively) and NMPs (9, 8, and 18%, respectively), while conventional gastruloids had far fewer PSMs (~9%) or NMPs (~2%) at 48 hrs. These trends were conserved in mouse gastruloid/TLS^{16,87} at 96 hrs, in that these Matrigel-sensitive models also had a higher proportion of NMPs and PSMs than conventional human gastruloids (**Fig. S32a**). It is notable that conventional human gastruloids had a high fraction of cardiac mesoderm-like cells (38%), absent from the other models in this comparison. Mesoderm in conventional human gastruloids expressed several BMPs (**Fig. S32d**), which could negatively impact **axial progenitor** ~~stem cell~~ maintenance, including NMPs^{27,28}, as also observed in the LDN-treated RA-gastruloids (**Fig. 5e**). Together, these observations suggest the hypothesis that having a sufficient number of **axial progenitors** ~~stem cells~~ may be a prerequisite for somite morphogenesis upon the addition of Matrigel."

2) Could the authors please add summary statistics for the RC somite patterning measured in all 18 RA-gastruloids (Fig S20D)? Given that the authors state that 14/18 RA-gastruloids displayed similar (RC) patterning, there really is no reason to not provide the summary statistics instead of just 3 examples with individual quantifications.

Certainly. In the newest revision of the manuscript, we updated **Fig. S20f-h** to show the signal intensities of TBX18 and UNCX staining across all RA-gastruloids examined for this analysis (*i.e.* for full transparency, we included the 4/18 that subjectively did not show similar patterning). We also now summarize the “peak to peak” distances for each staining in boxplots, which were on average just under 100 μm , broadly consistent with the somite lengths reported in **Fig. S14k**.

Figure S20. Markers of spatial patterning and differentiation in the somites of human RA-gastruloids.

f, (left) Representative image of HCR of *UNCX* and *TBX18* in 120 hrs human RA-gastruloids, and quantification of the signal intensity along with the A-P axis. Scale bar, 50 μm . (middle) Signal intensities of *TBX18* (blue) and *UNCX* (red) were measured on the yellow line of the left panel. Data was normalized with the mean values of each signal and processed with LOESS smoothing. (right). Line plot showing the difference between *TBX18* and *UNCX* values. Red and blue dots indicate the peaks for *UNCX*-high (red) and *TBX18*-high (blue), respectively, detected by a findpeaks function of pracma R package. **g**, Similar line plots for 18 RA-gastruloids. **h**, Boxplot showing the distribution of the distances between successive *TBX18*-high (blue) or *UNCX*-high (red) peaks. The upper whisker, upper box edge, bar, and lower box edge represent $1.5 \times \text{IQR}$, third quartile (Q3), median, and first quartile (Q1), respectively.

3) It would be helpful if the authors could add a short note to the Methods where they identify the criteria for "similar morphology" for all occasions where they refer to reproducibility by this terminology.

We certainly see the point, but as these criteria slightly differ in each instance in which we use the broader phrase "similar morphology and patterns of marker gene expression", we now include this information directly in each instance, rather than in the methods. The text has been adjusted in the following figure legends:

2b: "similar morphology (elongated gastruloid with flanking somites) and patterns of marker gene expression (asymmetric, elongated SOX2-mCit+ signal flanked by non-overlapping weak TBXT-mCer signal overlaying somites)"

2e: "similar morphology (elongated gastruloid with flanking somites) and patterns of marker gene expression (asymmetric, elongated, coincident SOX1 and SOX2 staining that did not extend to flanking somites)"

2f: "similar morphology (elongated gastruloid with flanking somites) and patterns of marker gene expression (asymmetric, elongated SOX2 staining flanked by PAX3 staining of flanking somites)"

3e: "similar morphology (elongated gastruloid with flanking somites) and patterns of marker gene expression (punctate, coincident WT1 and PAX8 staining at the lateral border of somites)"

3g: "similar morphology (elongated gastruloid with flanking somites) and expression patterns of marker genes (punctate SOX10 staining of cells appearing on one surface of region exhibiting SOX2 staining)."

S25b: "similar morphology (elongated gastruloid with a high density of somite-like structures appearing along the entire anterior-posterior axis)"

Also, the authors asked for some advice how to distinguish the ectopic NTs from the central NT in future work. My suggestions:

i) the genome engineering strategy employed for mouse Tbx6-KO TLS in Veenvliet et al., 2020. This allows for imaging-based distinction between ectopic and central NTs.

ii) Sampath Kumar et al., Nature Gen 2023 (<https://www.nature.com/articles/s41588-023-01435-6>) performed spatial transcriptomics on WT vs Tbx6-KO embryos & identified distinct molecular profiles for ectopic and central NTs (Fig 5). This could be leveraged to distinguish between both in RA-gastruloids.

Thank you for these terrific suggestions, they are very much appreciated!

Finally, the application of DNA Typewriter technology to RA-gastruloids sounds very exciting. I am truly looking forward to the future discoveries this will bring!

We are excited as well. Stay tuned!

ADDITIONAL COMMENTS/VIEW ON REMAINING ISSUES BY REVIEWER #2

I was asked to comment on whether the additional response by the authors provided via e-mail addressed points #2 and #3 by reviewer #2.

Point #2: In my opinion, the neural tube staining issue is sufficiently addressed by the SOX1 stainings in the current Fig 2 & 3. One technical point of concern regarding the immunostainings is the staining in Figure 3e. Why is there such a DAPI signal gradient? Especially since the WT1 staining partially follows this gradient. The authors could co-stain with eg FOXC1 (pan-somite) and WT1 to resolve.

We are pleased that Reviewer #1 agrees that the questions about neural tube staining raised by Reviewer #2 have been sufficiently addressed. However, they also raise a point about the DAPI staining gradient, together with a suggested experiment. We respond to this by pointing out two aspects of already-performed experiments and also by reporting our results with the suggested experiment.

First, we highlight the results shown in **Fig. S14a-b** (reproduced below) wherein CDH2 accumulation is observed in the internal regions of both the neural tube and somites, as measured with anti-CDH2, whose staining is highly coincident with the pattern of phalloidin accumulation. This result clearly shows that the anti-CDH2 antibody penetrated inside of the epithelialized neural tube and somite.

Figure S14. Morphological properties of human conventional and RA-gastruloids.

a, Immunostaining of N-cadherin (CDH2) and phalloidin in somites in an RA-gastruloid. Phalloidin-stained F-actin and CDH2 were co-localized and highly concentrated at the apical surface of somites. Scale bar, 10 μm . **b**, Immunostaining of N-cadherin (CDH2) and SOX1 in the neural tube in an RA-gastruloid. Scale bar, 100 μm .

Second, following Reviewer #1's suggestion, we purchased two anti-FOXC1 antibodies (PA1-807 and MAB6329-SP) and attempted co-staining of FOXC1 and WT1 in human RA-gastruloids. Unfortunately, these two anti-FOXC1 antibodies did not work at all well (no signal).

However, in this same experiment, we measured DAPI and WT1 intensities, and found no correlation. In particular, although there may be lower DAPI staining at more internal positions, a very sharp dropoff in WT1 staining is observed at depths where DAPI signal is entirely maintained. This result has been added as **Fig. S17c-d**.

Figure S17. Comparison of cell types detected in human conventional vs. RA-gastruloids.

c, Immunostaining of WT1 in 120 hrs RA-gastruloids. Signal intensities of DAPI (blue) and WT1 (red) were measured on the area indicated by the yellow line. Scale bar, 100 μm . **d**, Measured intensities of DAPI (blue) and WT1 (red) signals. Although there may be lower DAPI staining at more internal positions, a very sharp dropoff in WT1 staining is observed at depths where DAPI signal is entirely maintained.

Third, we highlight the results shown in the right panel of **Fig. 3e**, reproduced below, where no DAPI skew is observed, and PAX8 staining appears lateral to somites.

In summary, we understand the concern about WT1 staining partially following the DAPI gradient in **Fig. 3e**. However, the other results highlighted in this response, together with the new experiment added as **Fig. S17c-d**, together clearly show that our finding of lateral localization of WT1 is robust and not an artifact of poor penetration of the anti-WT1 antibody.

Figure 3. Induction of neural crest, intermediate mesoderm and other advanced cell types in human RA-gastruloids. e, (right) Immunostaining of 120 hrs RA-gastruloid with anti-PAX8 antibody (red, renal epithelium), anti-SOX2 antibody (green, neural tube) and DAPI (cyan, nuclear). Scale bar, 100 μm . Arrowheads indicate paired somites. Sm, somite; NT, neural tube.

Point #3: With their additional response, the authors have now addressed this point, in my opinion.

In our further revisions, we have addressed this point as we indicated we would in the planned point-by-point response.

Finally (regarding point #7), in my view, an embryo model (stembryo) should mimic aspects of *both* embryo cell type *and* patterning and/or architecture. This is *not* the case in the Faustino Martino et

al. neuromuscular organoids, which only recapitulate minor aspects of embryo morphogenesis in a very limited temporal window (elongation at day 5). In terms of recapitulating embryonic features (cellular composition concomitant with correct morphogenesis and patterning, it is therefore, in my opinion, a fair claim that "RA-gastruloids progress further than most alternative stembryo models" (side-note: stembryo models is not a correct term: it should be stembryos *or* embryo models; a stembryo model would be a model of a stembryo).

Thank you for this comment, and we agree. We also agree with your last point and have replaced "stembryo models" with "embryo models" in the referenced sentence.

Reviewer #2:

Remarks to the Author:

The study introduces another human gastruloid model (RA-gastruloids) that develop trunk like structures including cell types found in the neural tube and somites. Alongside this a computational staging method is presented to systematically compare developmental progression across models. The study primarily contains some useful but minor modifications to previously published human gastruloid protocols. I remain of the opinion that the study is better suited to a methods focused journal.

Concerning the specific issues raised in the last round of reviews.

1. The authors have corrected the assertion concerning an active segmentation clock.

We are glad to hear that this comment has been adequately addressed.

2. I remain of the opinion that some of the immunostaining is of poor quality. It is common to have antibody penetration problems resulting in uneven immunofluorescence and false negatives in organoids containing epithelia structures. This to be the case in the current study in some of the images (e.g. Fig 2F).

Please see Reviewer #1's adjudication of this comment, together with our response there.

3. The authors have partially clarified the conclusion that there is dorsal-ventral pattern in the neural tube. The problem is with the word "pattern" as it suggests a consistent and reproducible spatial arrangement of different cell types. The data do not seem to be sufficient to conclude this.

Although Reviewer #1 was satisfied that we have addressed this point in their adjudication, it's possible that our understanding of "pattern" in this context may differ from Reviewer #2. We have therefore addressed it by simply striking the word "pattern" from this paragraph, which now reads as follows:

"To investigate the cell types the possibility of dorsal-ventral spatial patterning of the neuroectoderm of human RA-gastruloids more deeply (albeit indirectly), we reanalyzed sc-RNA-seq data from the annotated neural tube, neural crest, and neural progenitor cells on their own, and then examined the expression patterns of dorsal-ventral marker genes in the

resulting embedding (**Fig. S19a-d; Table S4**). A large proportion of neural tube cells expressed dorsal neural tube or roof plate markers, including PAX3. In contrast, ventral neural tube or floor plate markers were not expressed (although SP8, which is present at the boundary of the dorsal and ventral regions of the neural tube in normal development, is detected). Together with the asymmetric appearance of neural crest-like cells on one surface of the RA-gastruloids' neural tubes, these results suggest that the human RA-gastruloids are dorsally biased, similar to mouse TLS and other stembryo models. We speculate that **the enrichment of dorsal cell types among neural tube cells in RA-gastruloids** ~~incomplete establishment of the D-V axis~~ is due to the lack of a Sonic hedgehog (SHH)-secreting notochord."

4. The authors now cite Olmstead and Paluh (2021).

We are glad to hear that this comment has been adequately addressed.

5. The authors have corrected their use of "neurogenesis".

We are glad to hear that this comment has been adequately addressed.

6. The authors now comment on the presence of syndetome (SCX+), sclerotome (PAX9+, NKX3-2+) cells. (Note syndetome arises ventrally between sclerotome and dermomyotome.)

We are glad to hear that this comment has been adequately addressed.

7. Unfortunately, the authors continue to overlook the work of Faustino Martins et al, which describes neuromuscular organoids. The authors state that they do not cite this work as neuromuscular organoids are an "an organoid model rather than a gastruloid/stembryo model". In my view, this is a distinction without a difference. NMOs are generated using a protocol very similar to the methods that the authors describe and they contain similar tissue derivatives. Ignoring the work of Faustino Martins et al seems to be an attempt to claim false novelty for their study. Similarly, although Olmstead and Paluh might not have provided single cell transcriptome data for later time points in their study, they offer evidence of advanced differentiation of organoids in their paper. I remain of the opinion that the statement "RA-gastruloids progress further than most alternative mammalian stembryo models" is over reaching.

Please see Reviewer #1's adjudication of this comment.

8. It remains the case that generating chimeric WT - Tbx6-KO gastruloids would be an opportunity of demonstrating the usefulness of gastruloid models to address biological questions.

We agree, but as per response **R2-8** from the previous round of review, the editor instructed us that this was beyond the scope of this manuscript. We intend to pursue it in future experiments.

Reviewer #3:

Remarks to the Author:

The authors have adequately addressed the remaining points I raised.

Moreover, in my opinion, the level of novelty of the embryonic organoid model they describe and the depth of description and analysis of the system across several conditions make this work suitable for publication in NCB.

We are gratified to hear that the remaining points have been successfully addressed, and appreciate the positive comments regarding the level of novelty, the depth of description and analysis, and the appropriateness of the paper for NCB.

My final recommendation is to make all the sequencing data they produced fully accessible also in the processed form (i.e., including cell type annotation, etc) to ensure reproducibility and facilitate the use of these beautiful and important datasets by the scientific community.

Thank you for this comment and encouragement. The raw and processed data, including cell x gene matrices with cell type annotations, etc., is GEO under accession GSE208369 (accessible with reviewer token "qtoncgmkftynxoz"). We will move it from private to public status after acceptance and prior to publication. Both the GEO accession and reviewer token are now listed in the **Data Availability** section.

Other editorial requests

(G) Finally please pay close attention to our guidelines on statistical and methodological reporting (listed below) as failure to do so may delay the reconsideration of the revised manuscript. In particular please provide:

A source data file corresponding to Fig. S28b, which is the only gel, is now included.

We now provide this as the 'Source_data_table.xlsx' file, submitted with the manuscript.

We therefore invite you to take these points into account when revising the manuscript. In addition, when preparing the revision please:

- ensure that it conforms to our format instructions and publication policies (see below and <https://www.nature.com/nature/for-authors>).

We have confirmed that the paper conforms to your format instructions and publication policies.

- provide a point-by-point rebuttal to the full referee reports verbatim, as provided at the end of this letter.

The point-by-point response is enclosed with the revision.

- provide the completed Reporting Summary (found here www.nature.com/documents/nr-reporting-summary.pdf). This is essential for reconsideration of the manuscript and will be available to editors and referees in the event of peer review. For more information see www.nature.com/authors/policies/availability.html or contact me.

The Reporting Summary is enclosed with the revision.

Decision Letter, first revision:

21st May 2024

Dear Jay,

Thank you for submitting your revised manuscript "Induction and *in silico* staging of human gastruloids with neural tube, segmented somites & advanced cell types" (NCB-A52363A). It has now been seen by the original referee #1 (who was asked to go through your responses to all the remaining reviewer points) and their comments are below. The reviewer finds that the paper has improved in revision, and therefore we'll be happy in principle to publish it in Nature Cell Biology as a Technical Report, pending minor revisions to comply with our editorial and formatting guidelines.

If the current version of your manuscript is in a PDF format, please email us a copy of the file in an editable format (Microsoft Word or LaTeX)-- we cannot proceed with PDFs at this stage.

We are now performing detailed checks on your paper and will send you a checklist detailing our editorial and formatting requirements in about 10 days. Please do not upload the final materials and make any revisions until you receive this additional information from us.

Thank you again for your interest in Nature Cell Biology. Please do not hesitate to contact me if you have any questions.

Best regards,
Stelios

Stylianos Lefkopoulos, PhD
He/him/his
Senior Editor, Nature Cell Biology
Springer Nature
Heidelberger Platz 3, 14197 Berlin, Germany

E-mail: stylianos.lefkopoulos@springernature.com
Twitter: @s_lefkopoulos
LinkedIn: [linkedin.com/in/stylianos-lefkopoulos-81b007a0](https://www.linkedin.com/in/stylianos-lefkopoulos-81b007a0)

Reviewer #1 (Remarks to the Author):

The authors have addressed all remaining comments. In my opinion, the manuscript is now ready for publication in NCB.

Decision Letter, final checks:

Our ref: NCB-A52363A

5th June 2024

Dear Dr. Shendure,

Thank you for your patience as we've prepared the guidelines for final submission of your Nature Cell Biology manuscript, "Induction and *in silico* staging of human gastruloids with neural tube, segmented somites & advanced cell types" (NCB-A52363A). Please carefully follow the step-by-step instructions provided in the attached file, and add a response in each row of the table to indicate the changes that you have made. Ensuring that each point is addressed will help to ensure that your revised manuscript can be swiftly handed over to our production team.

In recognition of the time and expertise our reviewers provide to Nature Cell Biology's editorial process, we would like to formally acknowledge their contribution to the external peer review of your manuscript entitled "Induction and *in silico* staging of human gastruloids with neural tube, segmented somites & advanced cell types". For those reviewers who give their assent, we will be publishing their names alongside the published article.

Nature Cell Biology offers a Transparent Peer Review option for new original research manuscripts submitted after December 1st, 2019. As part of this initiative, we encourage our authors to support increased transparency into the peer review process by agreeing to have the reviewer comments, author rebuttal letters, and editorial decision letters published as a Supplementary item. When you submit your final files please clearly state in your cover letter whether or not you would like to participate in this initiative. Please note that failure to state your preference will result in delays in accepting your manuscript for publication.

Cover suggestions

COVER ARTWORK: We welcome submissions of artwork for consideration for our cover. For more information, please see our guide for cover artwork.

Nature Cell Biology has now transitioned to a unified Rights Collection system which will allow our Author Services team to quickly and easily collect the rights and permissions required to publish your work. Approximately 10 days after your paper is formally accepted, you will receive an email in providing you with a link to complete the grant of rights. If your paper is eligible for Open Access, our Author Services team will also be in touch regarding any additional information that may be required to arrange payment for your article.

Please note that *Nature Cell Biology* is a Transformative Journal (TJ). Authors may publish their research with us through the traditional subscription access route or make their paper immediately open access through payment of an article-processing charge (APC). Authors will not be required to make a final decision about access to their article until it has been accepted. Find out more about Transformative Journals

Please use the following link for uploading these materials:
[Redacted]

Best regards,

Kendra Donahue
Staff
Nature Cell Biology

On behalf of

Stylianos Lefkopoulos, PhD
He/him/his
Senior Editor, Nature Cell Biology

Springer Nature
Heidelberger Platz 3, 14197 Berlin, Germany

E-mail: stylianos.lefkopoulos@springernature.com
Twitter: @s_lefkopoulos
LinkedIn: [linkedin.com/in/stylianos-lefkopoulos-81b007a0](https://www.linkedin.com/in/stylianos-lefkopoulos-81b007a0)

Reviewer #1:

Remarks to the Author:

The authors have addressed all remaining comments. In my opinion, the manuscript is now ready for publication in NCB.

Final Decision Letter:

Dear Jay,

I am pleased to inform you that your manuscript, "Retinoic acid induces human gastruloids with posterior embryo-like structures", has now been accepted for publication in Nature Cell Biology. Congratulations to you and the whole team!

Once your paper has been scheduled for online publication, the Nature press office will be in touch to

confirm the details. An online order form for reprints of your paper is available at <https://www.nature.com/reprints/author-reprints.html>. All co-authors, authors' institutions and authors' funding agencies can order reprints using the form appropriate to their geographical region.

Please note that *Nature Cell Biology* is a Transformative Journal (TJ). Authors may publish their research with us through the traditional subscription access route or make their paper immediately open access through payment of an article-processing charge (APC). Authors will not be required to make a final decision about access to their article until it has been accepted. Find out more about Transformative Journals

If you have not already done so, we strongly recommend that you upload the step-by-step protocols used in this manuscript to protocols.io (<https://protocols.io>), an open online resource that allows researchers to share their detailed experimental know-how. All uploaded protocols are made freely available and are assigned DOIs for ease of citation. Protocols and Nature Portfolio journal papers in which they are used can be linked to one another, and this link is clearly and prominently visible in the online versions of both. Authors who performed the specific experiments can act as primary authors for the Protocol as they will be best placed to share the methodology details, but the Corresponding Author of the present research paper should be included as one of the authors. By uploading your

Protocols onto protocols.io, you are enabling researchers to more readily reproduce or adapt the methodology you use, as well as increasing the visibility of your protocols and papers. You can also establish a dedicated workspace to collect your lab Protocols. Further information can be found at <https://www.protocols.io/help/publish-articles>.

With kind regards,
Stelios

Stylianos Lefkopoulos, PhD
He/him/his
Senior Editor, Nature Cell Biology
Springer Nature
Heidelberger Platz 3, 14197 Berlin, Germany

E-mail: stylianos.lefkopoulos@springernature.com
Twitter: [@s_lefkopoulos](https://twitter.com/s_lefkopoulos)
LinkedIn: [linkedin.com/in/stylianos-lefkopoulos-81b007a0](https://www.linkedin.com/in/stylianos-lefkopoulos-81b007a0)